# Capillary constrictions prime cancer cell tumorigenicity through PIEZO1

Giulia Silvani [1], Chantal Kopecky[2], Sara Romanazzo [2], Vanina Rodríguez [3], Ayan Das [4], Elvis Pandzic[5], John G. Lock [4], Christine L. Chaffer [3,6], Kate Poole [4] & Kristopher A. Kilian [1,2] ✉

Metastasis is responsible for most cancer-related deaths. However, only a fraction of circulating cancer cells succeed in forming secondary tumours, indicating that adaptive mechanisms during circulation play a part in dissemination. Here, we report that constriction during microcapillary transit triggers reprogramming of melanoma cells to a tumorigenic cancer stem cell-like state. Using a microfluidic device mimicking physiological flow rates and gradual capillary narrowing, we show that compression through narrow channels causes cell and nuclear deformation, rapid chromatin remodelling and increased calcium signalling via mechanosensor PIEZO1. Within minutes, cells upregulate transcripts associated with metabolic reprogramming and metastatic processes. Over time, this results in the stable adoption of a cancer stem cell-like state. Squeezed cells express elevated melanoma stem cell markers, exhibit increased trans-endothelium invasion and display enhanced tumorigenicity in vitro and in vivo. Pharmacological inhibition of PIEZO1 blocks this transition, while activation with Yoda1 induces the stem cell-like state irrespective of constriction. Deletion of PIEZO1 completely abolishes the constriction-induced phenotype. Together, these findings demonstrate that compressive forces during circulation reprogram circulating cancer cells into tumorigenic, stem cell-like states, primed for extravasation and metastatic colonization.

The primary cause of cancer-related deaths is by metastatic spread of cells from the primary tumour site to distant organs through the blood or lymphatic system[1]. During metastasis, cancer cells travel through the vascular system as circulating tumour cells (CTCs) until they encounter capillary beds and initiate extravasation[2]. Even a small vascularized primary tumour can shed millions of tumour cells daily into the bloodstream[3]. However, a vast majority of these circulating cells die due to the inhospitable environment within the vascular system[4]. Once in the bloodstream, CTCs must accommodate a lack of nutrients, oxidative stress, immune surveillance and anoikis, which is an apoptotic pathway that occurs when anchorage-dependent cells detach from the extracellular matrix. Additionally, CTCs must overcome intense mechanical forces arising from blood vessel flow, such as compression, friction and shear stress[5,6]. Despite these circumstances, a small fraction of CTCs manages to survive and reach capillary constrictions where they interact with the endothelial layer and extravasate to colonize a new environment. Survival during this process requires phenotypic plasticity with changes in cell state to effectively adapt to stresses and dynamic environmental conditions[7,8].

[1]School of Materials Science and Engineering, UNSW, Sydney, NSW, Australia. [2]School of Chemistry, Australian Centre for NanoMedicine (ACN), UNSW, Sydney, NSW, Australia. [3]St. Vincent's Clinical School, UNSW Medicine, UNSW Sydney, Darlinghurst NSW, Sydney, Australia. [4]School of Biomedical Sciences, Faculty of Medicine and Health, UNSW, Sydney, NSW, Australia. [5]Katharina Gaus Light Microscopy Facility, Mark Wainwright Analytical Centre, UNSW, Sydney, NSW, Australia. [6]The Kinghorn Cancer Centre, Garvan Institute of Medical Research, Darlinghurst, NSW, Australia. ✉e-mail: k.kilian@unsw.edu.au

During CTC residence time within microcapillaries, the narrowest of blood vessels, the cell and nuclear membrane can undergo large degrees of deformation, inducing damage or even triggering cell death[5,9–13]. However, several recent studies have shown how cell deformation can also activate pathways that elicit changes in metastatic potential. Hsia et al. demonstrated that cancer cells squeezing through small constrictions undergo nuclear deformation, which induces chromatin accessibility at promoter regions of genes linked to chromatin silencing, tumour invasion, and DNA damage response[14]. Intriguingly, Fanfone et al. showed that migrating breast cancer cells developed resistance to anoikis following their passage through confined spaces, which enhanced cell motility, evasion from immune surveillance and advantage to form lung metastatic lesions[15]. Using an in vivo model, Furlow et al. uncovered a molecular basis for prosurvival mechanisms of cancer cells upon microvasculature-induced biomechanical trauma through mechanosensitive pannexin-1 channels[16]. Together, these studies demonstrate how microvascular confinement can have a broad impact on cell state and cancer progression.

The difficulty in monitoring single cells transiting microcapillaries in vivo has led to the development of microfluidic-based models, where architecture and flow rates can be precisely varied with integrated imaging systems at single-cell resolution[17–22]. By mimicking constricted microchannels, flow-induced deformation and other mechanical stimuli arising from circulatory conditions, such as shear stress and friction, have been investigated and found to influence cancer cell transformations. For instance, Cognart et al. showed that combinations of shear flow and mechanical constraints differentially influenced cell phenotype, with significant changes in morphology and gene expression upon circulation[18]. Using a complex microfluidic design mimicking several blockages and constricted microchannels, Nath et al. showed how biophysical stimuli could drive mesenchymal-to-epithelial transition (MET), a necessary transformation after extravasation to form new tumours[20]. Estrada et al., engineered a closed-loop fluidic system to analyse molecular changes induced by variations in flow rate and pressure on lung adenocarcinoma cells[23]. They found that surviving cancer cell populations that express a specific circulatory transition phenotype are enriched in stem-like cells that overexpress MET markers, which led to increased metastatic disease and shorter survival in rodents.

In this article, we report the discovery that capillary-like constrictions catalyse phenotype changes in melanoma cells to a tumorigenic stem cell-like state through PIEZO1-mediated mechanotransduction. Custom microfluidic models were designed to mimic the gradual narrowing of the microcapillary bed, thereby approximating the mechanical forces encountered in vivo. After transiting the model microcapillaries, cells show increased expression of molecular markers of stemness, enhanced interactions with endothelial cells, and evidence for increased metastatic potency and tumorigenicity. Using pharmacological manipulators and gene editing approaches, we determined that PIEZO1 activity is responsible for this change in phenotype. Conventional models of metastasis suggest that only the stem cell fraction from a primary tumour can disseminate and establish colonies at distant sites, yet these models fail to account for the high rates of metastasis observed in patients. Our study challenges this view by uncovering an emerging mechanism for dissemination, where mechanical forces activating PIEZO1 during constriction in the microvasculature play a crucial role in enabling a broader population of tumour cells to successfully extravasate, survive and grow at metastatic sites.

## Results

### Microcapillary-like constrictions induce nuclear deformation and chromatin remodelling

To investigate how melanoma cells might respond to compressive forces during circulation, we designed and implemented a custom microfluidic device that replicates the progressive constrictions found in microcapillary networks (Fig. 1A). The device consists of a series of parallel constrictive channels systematically decreasing in diameter: 30 μm (i), 20 μm (ii), 10 μm (iii), 5 μm (iv). Each region is spaced apart by 150 μm to allow cells to relax after membrane deformation (Fig. 1B). These dimensions were chosen to reflect the range of vessel sizes encountered during microvascular transit, with the final 5 μm constrictions approximating capillary diameters observed in vivo[24–26]. Injecting cells at a physiological flow rate of 6 dyne/cm² relative to the narrowest channel, we observed a nearly 40% reduction in cell viability (Fig. 1C), as anticipated due to the intense mechanical forces involved. Furthermore, cells exhibited varying degrees of accumulated cell membrane deformation within the channel, as indicated by the deformation index (DI) in Fig. 1D, that correlated inversely with the channel diameter (Fig. 1E). Specifically, circulating melanoma cells accumulated significant membrane deformation in the 10 μm channel and 5 μm channel, with an increase of median deformation equal to 37% and 240%, respectively (highlighted in the cyan box plot in Fig. 1F). Upon passage into the relaxation chamber at the outlet of the corresponding channels, cells retained median deformation equal to 29% and 52%, from the 10 μm and 5 μm channels respectively (Fig. 1F).

The significant membrane deformation retained after passage through the 5 μm channel, smaller than the nucleus, suggests that the cells experienced substantial mechanical stress, which we hypothesized might lead to alterations in nuclear morphology and structural integrity of the chromatin, potentially influencing downstream cell behaviour. We first used Hoechst dye to inspect the nucleus before and after constriction (Fig. 1G). Squeezed melanoma cells show a 33% reduction in nuclear area along with a 66% increase in Hoechst intensity, indicating a likely change in chromatin condensation and accessibility compared to control cells (Fig. 1H, I). Since mechanical stress on the nucleus can weaken the nuclear envelope, we assessed Lamin A, a key nuclear membrane protein responsible for maintaining nuclear shape and stability, to determine whether nuclear structure was compromised. Immunolabelling showed no significant change in Lamin A expression between squeezed and control cells (Supplementary Fig. 1A). We further investigated whether these alterations were indicative of DNA damage by analysing Histone γH2A.X expression, a well-known marker for DNA double-strand breaks. However, we did not observe any significant changes in phospho-histone γH2A.X expression in the squeezed cells compared to the control (Supplementary Fig. 1B and C), suggesting that the mechanical stress applied during constriction likely results in chromatin remodelling rather than DNA damage.

To evaluate the chromatin state after constriction, we probed specific histone amino acid residues, namely histone H3 at lysine 27 (H3K27) and lysine 9 (H3K9). We chose to evaluate the acetylation states, associated with a relaxed and transcriptionally active euchromatin, and the trimethylation states, associated with the formation of transcriptionally inactive heterochromatin. H3K27 did not show significant differences between control and squeezed conditions for either acetylation or trimethylation states (Supplementary Fig. 2). In contrast, we observed a decrease of H3K9ac (Fig. 1L, M) and a significant increase of H3K9me3 (Fig. 1N, O) in squeezed melanoma cells compared to control. These histone modifications occurred rapidly, within 15 minutes of mechanical deformation, and reflect a shift from transcriptionally active to repressive chromatin states. To assess the stability of these changes, we performed additional time-course experiments, which revealed that H3K9ac reduction was transient and largely restored by 24 hours, while H3K9me3 remained stably elevated (Supplementary Fig. 3). This suggests that mechanical stress induces chromatin condensation and transcriptional silencing in specific genomic regions, priming cells for downstream transcriptional reprogramming and adaptive phenotypic changes.

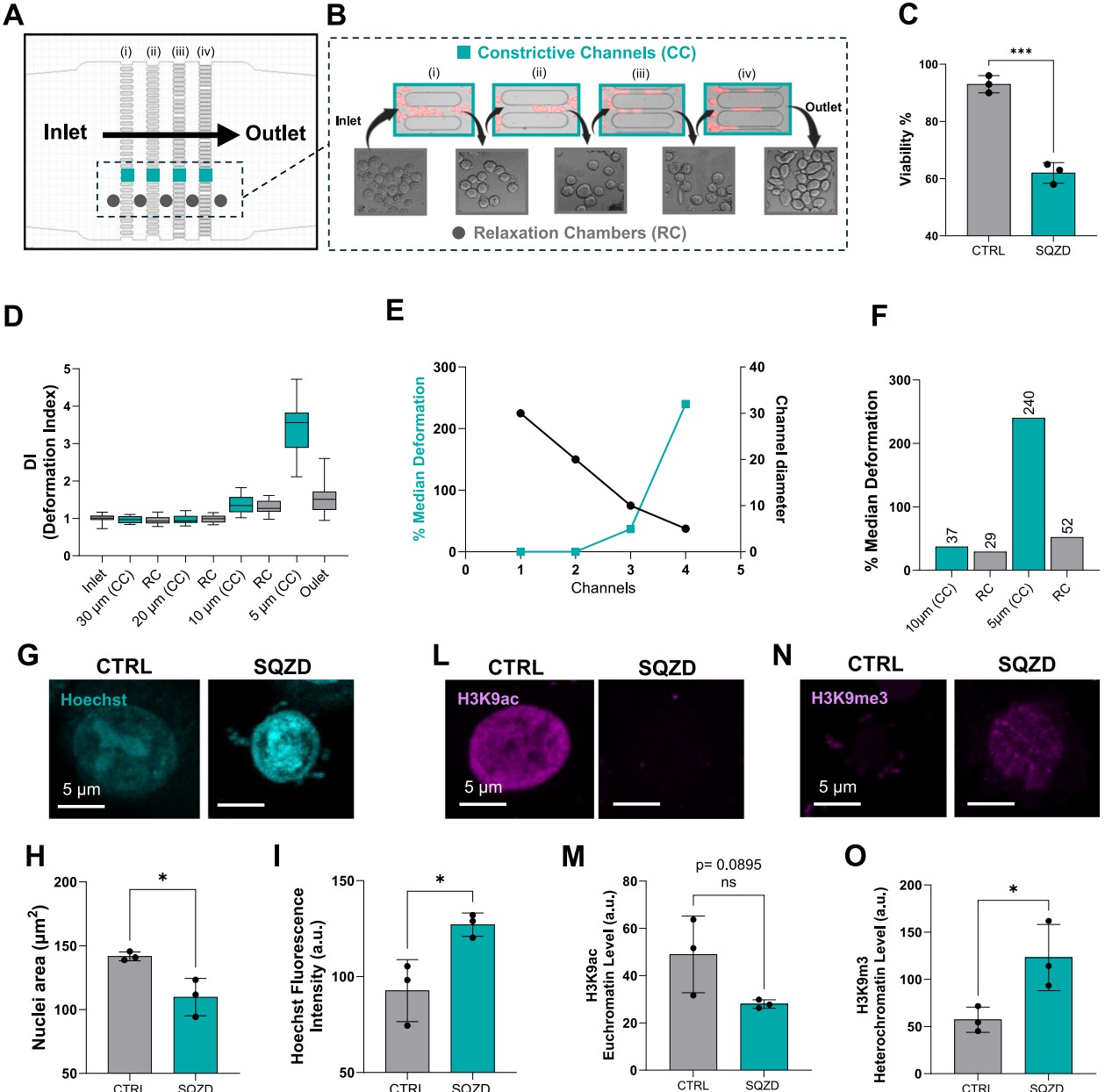

**Fig. 1 | Compressive forces induce melanoma cell state changes in a model microcapillary device. A** Device schematic consisting of a series of parallel constrictive channels of 30 μm (i), 20 μm (ii), 10 μm (iii), 5 μm (iv) diameter. **B** Images of melanoma cells passing through the micro constrictions and relaxation chambers. (**C**) Quantification of viable cells in control (CTRL) and squeezed (SQZD) groups. The results are expressed as the mean ± SEM from three independent experiments. Statistical significance assessed with two-sided unpaired t-test: *p* = 0.003 (***), 95% CI: [−38.52, −23.48]. **D-F** Plot of deformation index (DI) of melanoma cells transiting the microfluidic device, demonstration of inverse relationship between channel diameter and median deformation, and quantification of the % median deformation. *n* = 20 cells for each plotted condition. The observed trend was confirmed with 3x biological repeat experiments performed on different days. **G** Representative fluorescence images of nuclei (Hoechst) for CTRL and SQZD cells.

**H** Graph of nuclear area, expressed as mean ± SEM from three independent experiments. Statistical significance assessed with two-sided unpaired t-test: *p* = 0.0212 (*), 95% CI: [−56.12, −7.86]. **I** Graph of nuclear intensity (Hoechst), expressed as mean ± SEM from three independent experiments. Statistical significance assessed with two-sided unpaired t-test: *p* = 0.0261(*), 95% CI: [6.688, 62.11]. **L** and **M** Immunofluorescence images and quantification of H3K9ac in melanoma cells, expressed as mean ± SEM from three independent experiments. Statistical significance assessed with two-sided unpaired t-test: *p* = 0.00895 (ns), 95% CI: [−47.12, −5.131]. **N** and **O** Immunofluorescence images and quantification of H3K9me3 in melanoma cells, expressed as mean ± SEM from three independent experiments. Statistical significance assessed with two-sided unpaired t-test: *p* = 0.0382 (**), 95% CI: [5.805, 126.2]. Source data are provided as a Source Data file.

## Narrow constrictions induce phenotype changes associated with a melanoma stem cell-like state

Considering the changes in chromatin state, we next sought to explore whether transiting through a microcapillary would influence gene expression. Cells were collected immediately after exiting the microfluidic channel and subjected to whole-genome RNA sequencing.

Multidimensional scaling analysis showed separation of clusters associated with control and squeezed conditions (Fig. 2A), with 94% overlap in gene expression (Fig. 2B). The distribution of squeezed replicates is broader than control, which we attribute to heterogeneity in biological response. Nevertheless, all squeezed samples exhibit statistically significant differential gene expression compared to the

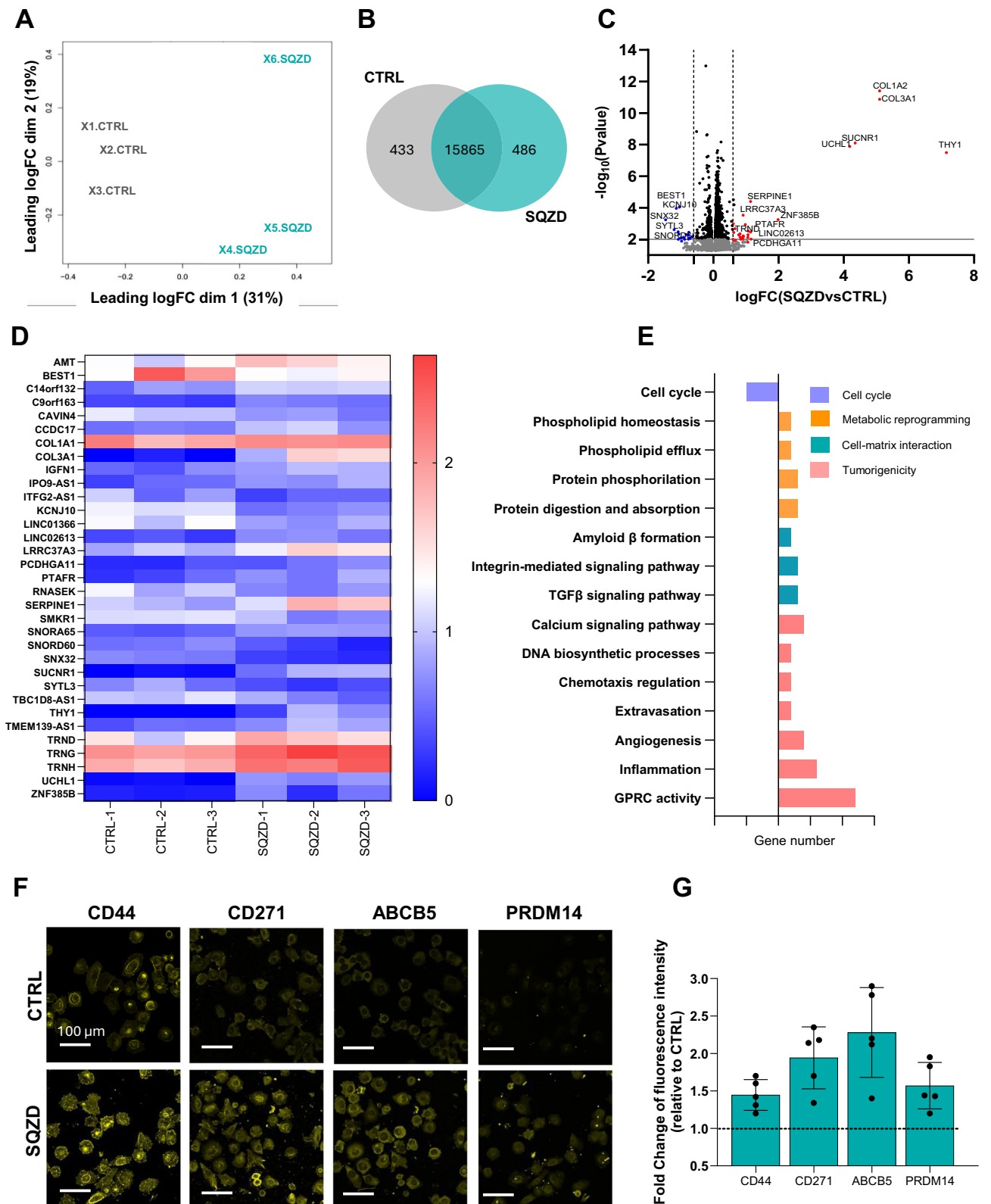

control group. Expression analysis revealed differential regulation of transcripts in the squeezed condition, as shown in the volcano plot and heatmap (Fig. 2C, D; log$_2$FC > 0.6, 1.5 fold change). Top hits include changes in transcriptional regulation (ZNF385B, SNORA65), tumour microenvironment (COL1A2, COL3A1), and metastatic processes (UCHL1, SUCNR1, THY1) (Fig. 2D). THY1, also known as CD90, was highly upregulated in the squeezed group, a protein associated with

calcium dependent integrin interactions between cancer cells and the endothelium[27,28]. SUCNR1, often found expressed in many types of cancers, was found upregulated in the squeezed cells compared to control and has been related to metastatic behaviour[29]. Moreover, pathway analysis performed on the top 100 genes with a *$p < 0.01$ indicated an upregulation of pathways associated with membrane transport, TGFβ signalling, extravasation, angiogenesis and

**Fig. 2 | Transcriptomics and immunofluorescence after constriction indicate increased invasiveness and evidence for stem cell-like states. A** MDS of the two groups control (X1, X2, X3 CTRL) and squeezed (X4, X5, X3 SQZD) analysed in triplicates, where X and Y axes represent dimensions that capture the largest differences between samples based on their gene expression profiles.
**B** Corresponding Venn diagram showing unique and overlapping genes between the two groups. **C** Volcano plot indicating significantly differentially expressed mRNAs between SQZD and CTRL. The grey horizontal line shows *P*-value cut-off (*$p < 0.01$) and the vertical dashed lines indicate up/down-regulated genes (<−1.5 and >1.5fold change). Statistical significance was determined using the edgeR exact test (two-sided), with Benjamini-Hochberg correction for multiple comparisons.

**D** Heatmap of the differentially expressed genes. **E** Bar graph displaying the number of genes involved in the GO biological processes and KEGG pathways associated with metabolic reprogramming (orange), tumorigenicity and metastasis (pink), cell-matrix interactions (green) and cell cycle (purple). Ticks in the gene number x axis are every 5 genes. **F** Representative fluorescence images of melanoma cancer cells, seeded on glass before (CTRL) and 4 hours after been squeezed (SQZD) into the microfluidic device, for biomarkers relative to stemness characteristic, e.g., CD44, CD271, ABCB5, PRDM14. **G** Bar graph showing the fold change of fluorescence signal of stemness-related biomarkers for SQZD cells compared to CTRL cells. The results are expressed as the mean ± SEM from five independent experiments. Source data are provided as a Source Data file.

metabolism, amongst others, and downregulation of pathways associated with cell cycle (Fig. 2E). Gene ontology enrichment analysis revealed transcripts associated with cancer progression and metastasis, cell growth and proliferation, including the negative regulation of the apoptotic process and various protein kinase activities (Supplementary Fig. 4), as well as networks associated with integrin signalling, cancer cell–endothelia cell interactions and angiogenesis (Supplementary Fig. 5).

With evidence for transcript networks corresponding to increased cancer cell invasive processes, we evaluated changes at the protein level through immunofluorescence analysis of markers associated with tumorigenicity and metastatic potential. After passing through the microfluidic channel, cells were collected and seeded on glass to allow adhesion prior to fixation and immunostaining. Remarkably, we observed an increase in expression of molecular markers associated with tumorigenicity, chemoresistance and melanoma cancer stem cell-like states within hours: specifically, CD44, CD271, ABCB5, and PRDM14 (Fig. 2F, G). To clarify the timing of these phenotypic changes, we assessed marker expression at multiple time points ranging from 15 minutes to 72 hours post-constriction. Differences in protein expression for stemness-associated markers like CD44 and ABCB5 began to emerge around 3–4 hours, peaked at 24 hours, and diminished by 72 hours (Supplementary Fig. 6). Although the passage through the microfluidic constriction occurs within milliseconds under physiologically relevant flow conditions, the resulting changes in protein expression emerged over a longer timescale, indicating that the initial mechanical stimulus triggers sustained downstream signalling activity. To confirm that deformation, rather than flow alone, drives the observed effects, we conducted control experiments using a non-constrictive device where cells experienced identical flow rates without confinement. No changes in stemness markers were observed under these conditions, supporting that mechanical constriction is the critical factor driving phenotypic reprogramming (Supplementary Fig. 7A–D).

As a functional test of stem cell-like characteristics, we collected cells at the outlet of the microcapillary device and grew them in equal number in a tumorsphere assay for a week (Fig. 3A). Cells that experience microcapillary constriction displayed a greater tendency for tumorsphere formation, with a 2-fold increase in size compared to control at the end of the assay, e.g., day 7 (Fig. 3B, C). Notably, no tumorsphere formation was observed in control conditions lacking mechanical deformation, underscoring the role of constriction-induced reprogramming (Supplementary Fig. 7E, F).

This significant enhancement in growth is evidence for a higher fraction of stem cell-like progenitors[30]. However, while in vitro tumorsphere assays provide strong evidence in support of tumorigenicity, experimental metastasis in animal models is necessary to confirm pro-metastatic activity. As an initial proof-of-concept, we conducted a trial metastasis experiment using tail vein injection of 5 × $10^5$ A375-luciferase melanoma cells into 7-week-old female Balb/c Nude mice. This model allowed us to examine early tumour cell seeding and outgrowth in the lungs under controlled conditions. To confirm

comparability, we first validated that the squeezed cell population exhibited elevated expression of stemness and tumorigenicity markers prior to injection (Supplementary Fig. 8). At 30 days post-injection, IVIS imaging revealed increased pulmonary metastases in animals injected with squeezed melanoma cells (Fig. 3D). This was confirmed by histological analysis of lung sections using Hematoxylin and Eosin (H&E) staining, which showed a greater number of metastatic nodules compared to controls (Fig. 3E, F).

Building upon this preliminary finding, we next expanded and validated our observations using a more sensitive and systemic intracardiac injection model to assess global metastatic dissemination and organotropic colonisation. Following the injection of $1 \times 10^5$ A375-Luc2 melanoma cells into the left ventricle, mice receiving squeezed cells exhibited a striking increase in total metastatic burden and distribution. Longitudinal bioluminescence imaging demonstrated elevated whole-body photon flux in the squeezed group by week 4 (Fig. 3G), including increased median signals in the lung (Fig. 3H) and significantly higher signals in bone and brain (Fig. 3I, L), indicating elevated organ-specific homing. Importantly, survival analysis indicated that animals injected with squeezed cells had reduced overall survival, with multiple animals reaching ethical endpoints before day 50 (Fig. 3M). In fact, 9/10 animals in the squeezed group reached the ethical endpoint by day 52, compared to 1/8 animals in the control group. Collectively, these results demonstrate that microconstriction enhances the in vivo metastatic potency and lethality of melanoma cells in a systemic dissemination model.

**Constriction-induced conversion to a melanoma stem cell-like phenotype is governed by PIEZO1 mechanosensory activity**
Cells interpret mechanical stimuli through specialized mechanosensors, including mechanically gated ion channels. Among these, PIEZO1 has emerged as a critical regulator of mechanotransduction in cancer, previously implicated in responses to shear stress, membrane stretch, and compression[31]. Given that our microcapillary model imposes abrupt mechanical deformation and elicits downstream changes associated with calcium handling, we hypothesized that PIEZO1 could play a key role in sensing and transducing these mechanical cues. To test this, we investigated PIEZO1 expression and localization at the plasma membrane using immunofluorescence, noting an increase in PIEZO1 signal intensity in squeezed melanoma cells when compared to unsqueezed controls (Fig. 4A, B). This suggests that mechanical deformation promotes membrane reorganization and clustering of PIEZO1, positioning the channel for enhanced mechanosensitive activation under capillary-like stress and facilitating downstream calcium influx. To verify this result, we conducted calcium imaging of cells transiting the microcapillary. Using the calcium-sensing dye Calbryte 520 AM, we observed a rapid influx of calcium ions coinciding with cell deformation during live cell imaging (Fig. 4C). Notably, this calcium influx was primarily observed during transit through the narrower constrictions (10 μm and 5 μm), where significant deformation was induced. To evaluate signal transduction downstream of PIEZO1, we performed phosphoproteomic profiling at early

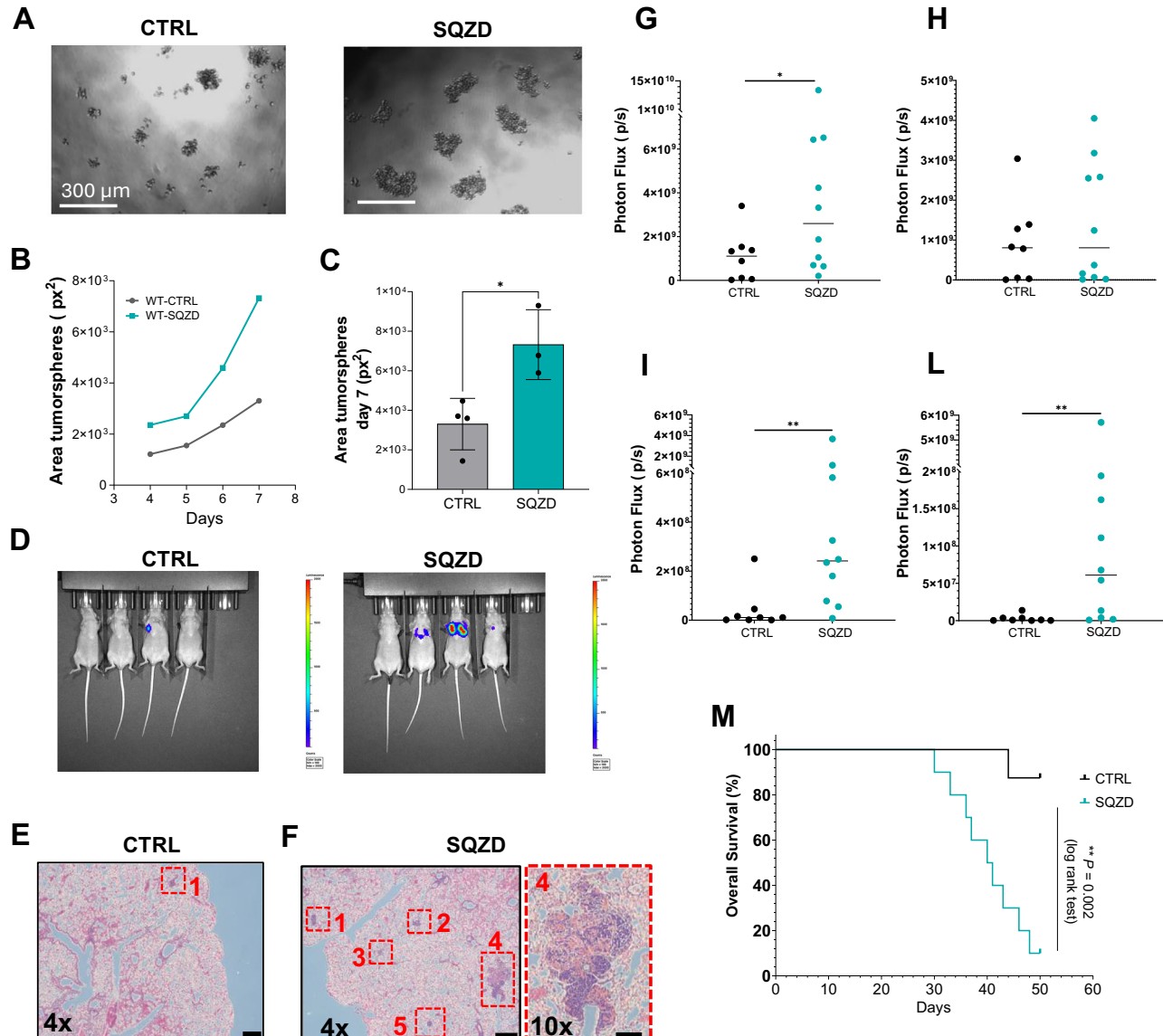

**Fig. 3 | Microcapillary-like constrictions enhance melanoma cell tumor-igenicity.** **A** Representative images of control (CTRL) and squeezed (SQZD) mel-anoma cells forming a tumour sphere in serum-free media after 7 days of culture. **B** and **C** Graph representative of the mean tumour sphere area increasing over time for different conditions and the quantification of tumour sphere area at day 7. The result is expressed as the mean ± SEM from $n = 4$ CTRL and $n = 3$ SQZD independent experiments. Statistical significance was assessed using a two-sided unpaired t-test. Asterisks indicate p-values, $p = 0.0174$ (*), 95% CI: [1060, 6975]. **D** IVIS Imaging of lung metastases that developed after 30 days from injecting CTRL and SQZD luciferase melanoma cells into the lateral tail vein of 7-week-old female Balb/c Nude mice. Representative H&E-stained lung sections from mice injected with CTRL cells (**E**) and SQZD cells (**F**) harvested 30 days post tail vein injection. These samples correspond to the mice with the highest luciferase signal in each group ($n = 4$ mice per group). Histological analysis was performed on these individual animals only. **G** Whole body luciferase signal of mice injected intra-cardically at week 4, the final timepoint at which all mice in both groups were accounted for (CTR, $n = 8$ mice; SQZ, $n = 10$ mice). Statistical significance was assessed using a two-sided unpaired t-test. Asterisks indicate p-values, $p = 0.0195$ (*), 95% CI: [182243370, 5419316630]. Photon flux from individual metastatic sites revealed increased median signals in the **H** lung and significantly higher signals in (**I**) bone and **L** brain in the SQZ group compared to controls. Statistical significance was assessed using a two-sided unpaired t-test. Asterisks indicate p-values, Bone $p = 0.0031$ (**), Brain $p = 0.0022$ (**). Source data are provided as a Source Data file. Each dot represents an indivi-dual animal. **M** Metastatic burden and Kaplan–Meier survival curves following IC injection with control (grey) and squeezed (cyan) cells.

time points post-constriction using multiplexed immunolabeling[32,33]. The analysis revealed dynamic changes in key mechanosensitive pathways including NF-κB, PKC, YAP1, mTOR, and ERK, consistent with their established roles downstream of PIEZO1 activation (Supplementary Fig. 9). Together, these findings provide molecular evidence that PIEZO1 is activated in response to capillary-scale mechanical com-pression and likely plays a key role in initiating the downstream sig-nalling cascades observed in this model.

Based on these observations, we next sought to functionally assess whether PIEZO1 activation is directly involved in regulating the cancer stem cell-like phenotype. To this end, we pharmacologically manipulated PIEZO1 activity using selective activators and inhibitors of mechanically gated ion channels. First, we treated the cells with the selective PIEZO1 activator Yoda1, which alters the molecular structure, thereby lowering the activation threshold for the influx of calcium. Strikingly, treating the cells with Yoda1 without any mechanical sti-mulation resulted in the increased expression of molecular markers associated with the melanoma stem cell-like phenotype (Fig. 4D, E). We also used the mechanically gated ion channel inhibitor Ruthenium Red (RR), which broadly blocks calcium channels and reduces ion flow

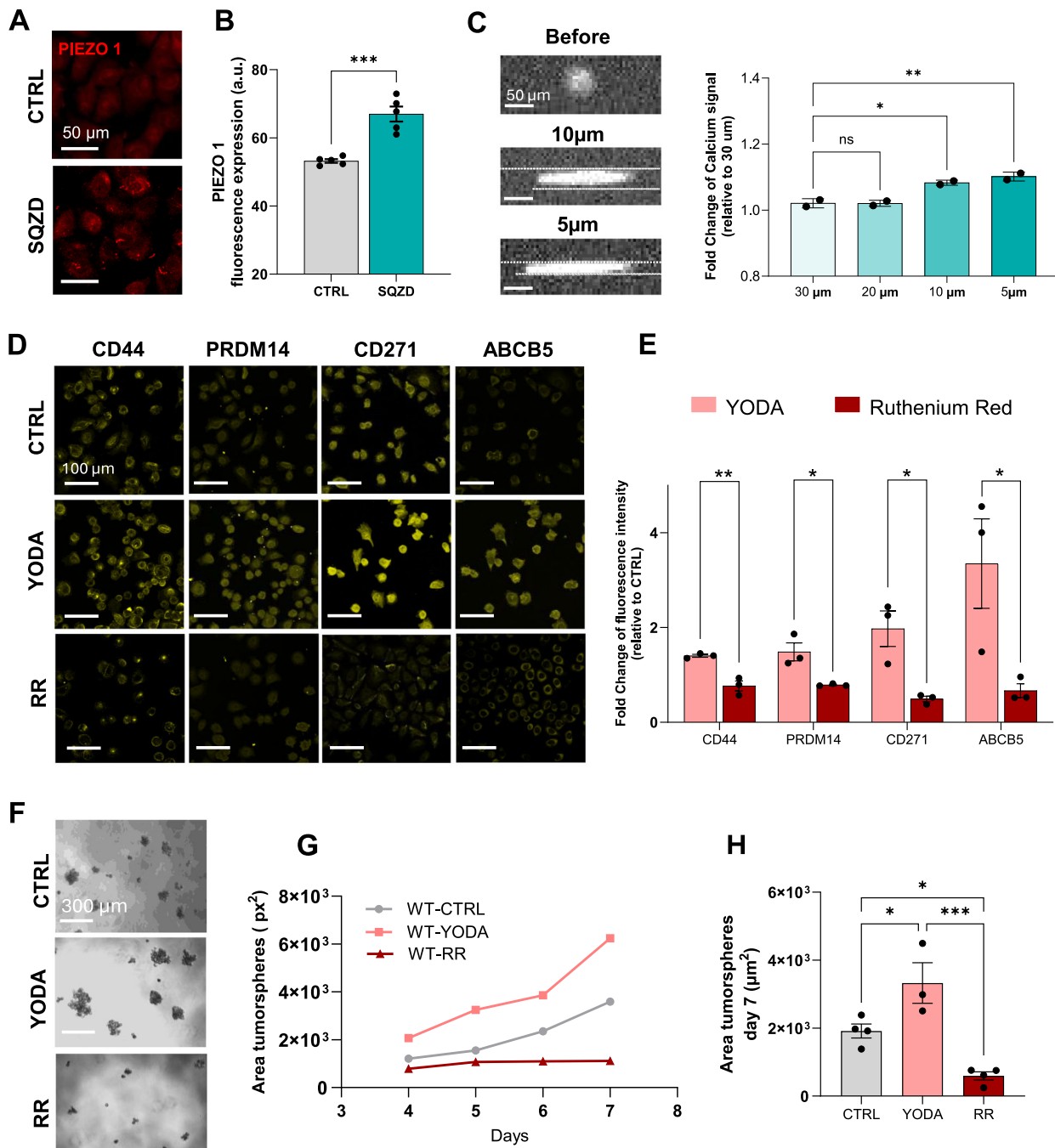

**Fig. 4 | PIEOZ1 activity directs stem cell-like states and tumorigenicity after microconstriction. A** Representative fluorescence images of PIEZO1 expression for CTRL and SQZD cells. **B** Quantification of PIEZO1 immunofluorescence intensity expressed as the mean ± SEM from five independent experiments. Statistical significance assessed using a two-sided unpaired t-test: $p = 0.0003$ (***).
$p = 0.0003$(***), 95% CI: [8.521, 18.99]. **C** Representative fluorescence images showing melanoma cells as they squeeze through the constrictive channels with quantification of fold change intracellular calcium concentration using Calbryte 520 AM. The results are expressed as the mean ± SEM from 2 independent experiments. Statistical significance was assessed using one-way ANOVA (two-sided): 30 μm vs. 10 μm (CC) $p = 0.0185$ (*) 95% CI: [−0.1084 to −0.01587]; 30 μm vs. 5 μm (CC) $p = 0.0071$ (**) 95% CI: [−0.1271 to −0.03463]. **D** Representative fluorescence images of cancer cells showing stemness biomarkers under CTRL condition and after treatment with 20 μM Yoda1 and 30 μM Ruthenium Red (RR). **E** Bar graph showing the fold change of fluorescence signal of stemness-related biomarkers for cells under different treatments. The result is expressed as the mean ± SEM from three independent experiments. Statistical significance was assessed using a two-sided unpaired t-test. Asterisks indicate p-values as follows: CD44 $p = 0.0044$ (**) 95% CI: [−0.9418, −0.3318], PRDM14 $p = 0.0212$ (*) 95% CI: [−1.228, −0.1722], CD271 $p = 0.0175$ (*) 95% CI: [−2.537, −0.4282], ABCB5 $p = 0.048$ (*) 95% CI: [−5.333, −0.02600]. **F-H** Images of CTRL and treated melanoma cells, forming tumorsphere in serum-free media after 7 days of culture, demonstration of increased mean tumorsphere area over time and quantification of tumorsphere area expressed as the mean ± SEM from four independent experiments, except for YODA ($n = 3$). Statistical significance was assessed using two-way ANOVA (two-sided). Asterisks indicate p-values as follows: CTRL vs. YODA $p = 0.0346$ (*) 95% CI: [−2712, −114.6]; CTRL vs. RR $p = 0.0333$ (*) 95% CI: [117.0, 2522]; YODA vs. RR $p = 0.0008$ (***) 95% CI: [1434, 4032].

across the membrane. Treating the cells with RR led to a reduction in expression of molecular markers of the melanoma stem cell-like state after constriction (Fig. 4D, E).

In serum-free culture conditions, melanoma cells exhibited consistent responses to drugs manipulating the PIEZO1 channel (Fig. 4F). Specifically, cells treated with Yoda1 demonstrated enhanced tumorsphere growth compared to controls (1.34-fold increased growth rate), whereas treatment with RR abrogated growth, resulting in small clusters of melanoma cells (Fig. 4G). At day 7, cells subjected to Yoda1 treatment showed a 70% increase in tumorsphere area, compared to non-treated cells, whereas treatment with RR reduced tumorsphere size by 97%, compared to Yoda1 treatment (Fig. 4H). These results raise the provocative possibility that PIEZO1 stimulation alone is sufficient to regulate the melanoma stem cell-like state and cancer tumorigenicity.

To further examine the specificity of mechanosensitive channel involvement, we repeated the experiments using GsMTx4, a more selective blocker of stretch-activated ion channels. Consistent with the results obtained using Ruthenium Red, treatment with GsMTx4 suppressed the upregulation of CD44 and ABCB5 following mechanical squeezing and significantly reduced tumorsphere formation capacity (Supplementary Fig. 10). These findings strengthen the evidence for a mechanosensitive ion channel-dependent mechanism underlying the observed phenotypic changes. Finally, to directly assess the requirement of extracellular calcium influx, we performed squeezing experiments under calcium-depleted conditions. In calcium-free media, mechanical squeezing failed to induce upregulation of stemness markers CD44 and ABCB5 (Supplementary Fig. 11), indicating that PIEZO1-mediated calcium entry is necessary to initiate the stem cell-like phenotype. Together, these results support a model in which mechanical stress activates PIEZO1-dependent calcium signalling to drive melanoma stem cell-like reprogramming.

To validate the involvement of the PIEZO1 molecule constriction-induced stem cell-like phenotypes, we utilized CRISPR technology to knock out the mechanosensor. PIEZO1 deletion led to different membrane deformation characteristics, as reflected by the DI measured for cells after transiting into the microconstriction (Fig. 5A). Notably, cells lacking PIEZO1 showed minimal retained membrane deformation compared to wild-type cells (WT), with Knockout (KO) cells returning to their original spherical shape after transiting into the $10\,\mu m$ and $5\,\mu m$ channels. Viability analysis showed that KO cells experienced a similar degree of cell death upon deformation compared to WT cells, indicating that PIEZO1 deletion does not impact survival after constriction (Fig. 5B). To further investigate this difference in mechanical response, we performed real-time deformability cytometry to quantify cell mechanical properties before and after squeezing. WT melanoma cells showed a significant increase in stiffness following constriction, consistent with an adaptive mechanical response. In contrast, KO cells did not exhibit any appreciable change in stiffness or viscoelastic properties, suggesting that PIEZO1 is required for this mechano-adaptive behaviour (Supplementary Fig. 12). Simultaneous calcium imaging of cells squeezing into the microcapillary shows decreased calcium uptake for KO cells within the narrowest capillaries compared to control (Fig. 5C, D), indicating that calcium influx during microcapillary-induced deformation is, in part, dependent on PIEZO1. Additionally, in contrast to WT cells, the nuclear area in KO cells did not show a significant difference after constriction (Fig. 5E, F), along with no significant change in the H3K9ac state (Supplementary Fig. 13A, B). However, constriction showed a modest decrease in the H3K9me3 mark in the KO cells, instead of an increase in this mark observed in WT cells, suggesting alternative epigenetic regulation during constriction in the absence of PIEZO1 (Supplementary Fig. 13C). Immunofluorescence analysis of KO cells after constriction revealed no change in molecular markers of a stem cell-like state when compared to control cells, confirming the direct role of PIEZO1 in mechano-regulation of this state (Fig. 5G, H). As a functional test of stemness and tumorigenicity, we repeated the tumorsphere assay using KO cells (Fig. 5I). PIEZO1 deletion from squeezed melanoma cells results in a significant reduction in tumorsphere formation and growth over time (Fig. 5L), with an 80% decrease in size compared to squeezed WT cells (Fig. 5M). Notably, the decrease in tumorsphere formation capacity from PIEZO1 deletion is comparable to WT cells treated with RR, confirming the direct role of PIEZO1 in regulating tumorigenicity (Fig. 5M).

## Constriction-induced phenotypes show enhanced interactions with the endothelium

In addition to increased tumorigenicity, the transcriptomics and experimental metastasis assay suggest increased invasiveness and metastatic potency after microconstriction. However, quantifying direct engagement of CTCs with the endothelium is challenging in vivo. Therefore, to assess the invasiveness of constriction-induced stem cell-like states, we subjected the cells to a selection of in vitro assays for measuring extravasation and invasiveness. First, we conducted a transwell assay to monitor cell migration through a porous membrane as a model of invasion (Fig. 6A). Squeezed cells showed increased migration through the transwell membrane, whereas the PIEZO1 KO cells performed similarly to untreated WT cells (Fig. 6B). Cells that migrated through and adhered to the bottom of the insert were stained for CD44, revealing higher expression of this marker in squeezed cells compared to both WT controls and KO cells (Supplementary Fig. 14), supporting the link between increased invasiveness and acquisition of stem cell-like traits.

An essential phase in the metastatic cascade, crucial for the successful extravasation and colonization of distant organs, involves the interaction between circulating cancer cells and endothelial cells lining the blood vessel walls. To investigate whether constriction might augment this process, we conducted an in vitro co-culture of melanoma cancer cells with endothelial cells (Fig. 6C). Specifically, we utilized a perfusable PDMS chip to facilitate the formation of a mature endothelial layer (Fig. 6D, **Endothelium**). After culturing endothelial cells for 3 days, we verified endothelial barrier integrity through immunostaining for VE-Cadherin. Once maturation was confirmed, melanoma cell suspensions (WT and KO, squeezed and untreated) were centrifuged and only the viable cells were introduced into the PDMS platform and co-cultured overnight under mild flow conditions. Interestingly, the squeezed WT melanoma cells induced an inflammatory response in the endothelium as evidenced by increased expression of endothelial vascular cell adhesion molecule (*VCAM-1*), which was not observed in the control conditions (Fig. 6D). KO cells did not exhibit the same inflammatory response under both experimental conditions. Crucially, squeezed WT melanoma cells caused remodelling of VE-Cadherin at endothelial cell-cell junctions, indicating that trans-endothelial migration involves the disruption of junctional proteins during extravasation. In contrast, this mechanism was absent in squeezed KO cells, where VE-Cadherin staining remained linear and localized at junctions (Fig. 6E).

Observing extravasation of cancer cells from vasculature in vivo is challenging and currently impossible to probe beyond single isolated events. To evaluate the potential for extravasation, we adopted a previously established microfluidic chip-based model[19], where a perfusable blood vessel is juxtaposed to a chamber to monitor invasion events (Fig. 7A). Endothelial cells were seeded into the vascular channel and perfused at physiological flow rates until a mature and functional blood vessel formed within the microfluidic chip, as indicated by the signal of VE-Cadherin junction at cell-cell contact (Fig. 7B). Subsequently, melanoma cancer cells were injected into the circulation and their penetration through the modelled vessel barrier and into the tissue compartment was monitored overnight. The squeezed WT melanoma cells exhibited an increased propensity for extravasation

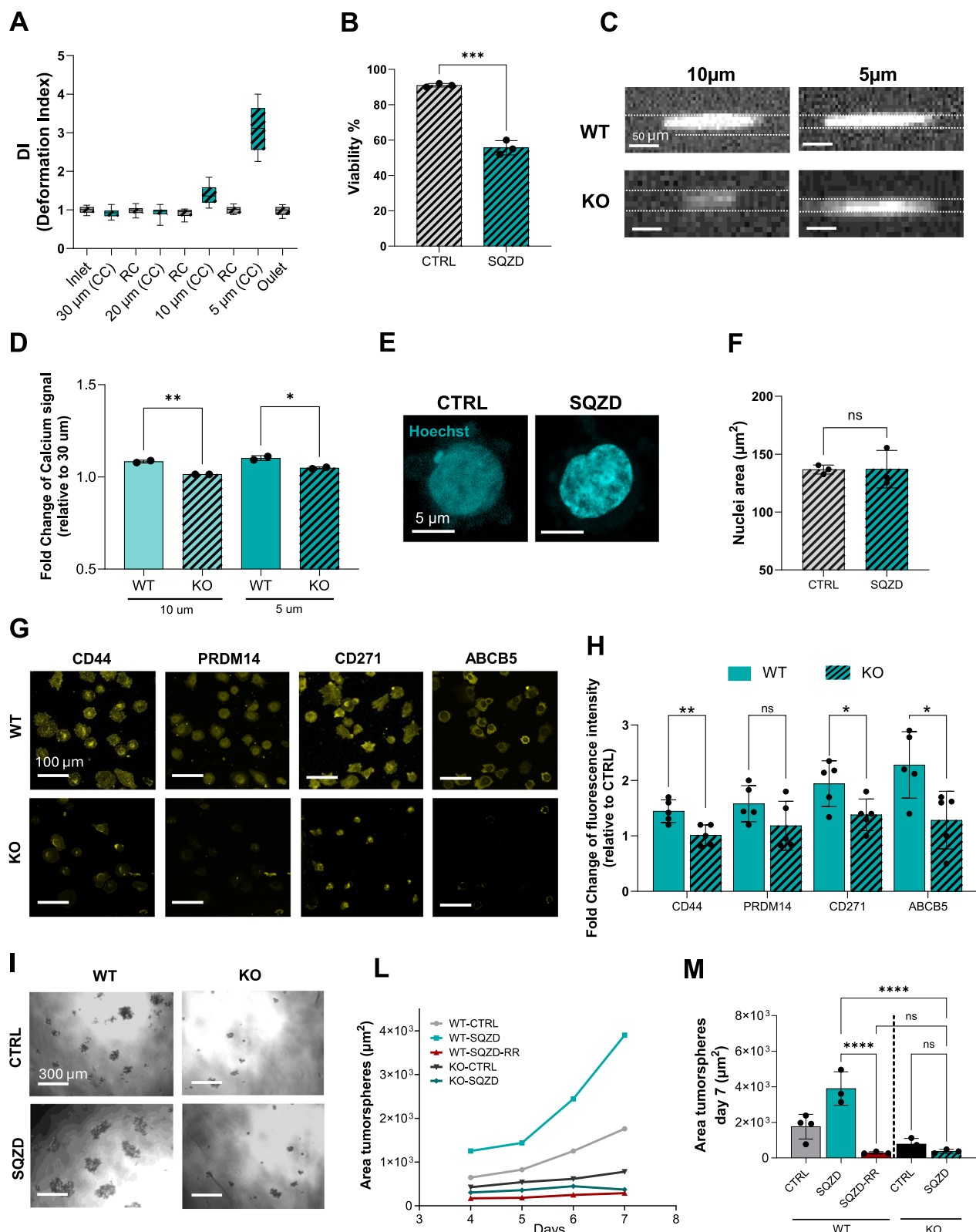

into the chamber compared to controls (Fig. 7C), which was not observed in the squeezed PIEZO1 KO population. To evaluate how constricted cells may remodel endothelial cell-cell junctions to support invasion, we assessed blood vessel permeability immediately following the overnight exposure of melanoma cells to microcirculation (Fig. 7D). First, we assessed blood vessel permeability (Vessel) by measuring the diffusion of dextran nanoparticle over time into the

tissue compartment. As expected, we found that the vessel's permeability coefficient was two orders of magnitude lower than a cell-free device (CTRL), confirming the presence of a functional biological barrier hampering dye diffusion. Strikingly, we found that the permeability of the blood vessel exposed to squeezed cells was 20-fold increased compared to untreated and squeezed PIEZO1 KO cells. These results indicate that the tumorigenic and pro-metastatic state induced

**Fig. 5 | Loss of PIEZO1 obviates emergence of stem cell-like phenotypes after microcapillary-induced deformation. A** Plot of deformation index (DI) for melanoma cells transiting the microfluidic device ($n = 100–150$ cells per condition). c **B** Cell viability expressed as the mean ± SEM from three independent experiments. Statistical significance assessed with two-sided unpaired t-test: $p = 0.0001$ (***) 95% CI: [-42.01, -28.66]. **C, D** Immunofluorescence images of melanoma cells transiting constrictive channels and quantification of fold change intracellular calcium concentration ($n = 7–40$ cells per condition). The results are expressed as the mean ± SEM from 2 independent experiments. Statistical significance assessed using two-sided unpaired t-test: 10 µm $p = 0.0038$ (**) 95% CI: [0.03575 to 0.1028]; 5 µm $p = 0.0459$ (*) 95% CI: [0.0008904 to 0.06797]. **E, F** Immunofluorescence images and quantification of nuclei with results expressed as the mean ± SEM from three independent experiments. Statistical significance assessed using a two-sided unpaired t-test: $p = 0.97$ (ns), 95% CI: [-26.46, 27.12. **G** Immunofluorescence images of stemness markers in melanoma cells and **H** quantification of fold change relative to control, expressed as the mean ± SEM from five independent experiments. Statistical significance assessed using two-sided unpaired t-tests: CD44 $p = 0.0079$ (**), 95% CI: [-0.7189, -0.1491]; CD271 $p = 0.037$ (*), 95% CI: [-1.081, -0.04346]; ABCB5 $p = 0.0135$(*), 95% CI: [-1.709, -0.2568]. **I, L, M** Representative images of melanoma cells forming tumorsphere after 7 days of culture, demonstration of change in tumorsphere area over time, and quantification of tumorsphere area at day 7, expressed as the mean ± SEM from four independent experiments. Statistical significance assessed using two-sided unpaired t-test: WT SQZD vs. WT SQZD_RR $p < 0.0001$ (****), 95% CI: [2130, 093]; WT SQZD vs. KO SQZD_RR $p < 0.0001$ (****), 95% CI: [2051, 5013].

by microconstriction predisposes cells to disrupt blood vessel integrity to enhance extravasation and metastatic spread.

## Discussion

Metastasis, the leading cause of cancer-related deaths, is a highly selective process in which only a small fraction of cancer cells can survive the challenging conditions of the microcapillary bed and successfully extravasate to form secondary tumours. Despite its critical role in cancer progression, the factors influencing this small subset of cells remain poorly understood and require further investigation. Utilizing in vitro models that replicate mechanical forces, such as fluid dynamics and physical constrictions, is essential for advancing our understanding of how these cells adapt and succeed in the metastatic process. Here, we utilised a microfluidic model of microcapillary constriction to study the effect on circulating melanoma cells, which revealed distinct relationships between the forces in microcapillary constriction and regulation of a melanoma stem cell-like state, conveyed through the action of the mechanosensor PIEZO1.

We introduced a design that simulates the gradual narrowing of a microcapillary bed using a sequence of four channel widths: 30 µm, 20 µm, 10 µm, and 5 µm, closely approximating the physiological range of microcapillary diameters, typically between 5 and 10 µm, and capturing the dynamic constriction that circulating cells encounter in vivo[24]. By incorporating a series of progressively narrower constrictions, this approach closely mimics the complex architecture of physiological microcapillary beds. The wider upstream channel sections facilitate smooth cell entry and proper flow alignment, enabling more realistic simulation of how cells interact with and navigate through confined vascular environments. Although microfluidic technology does not fully replicate the mechanical properties of in vivo capillaries, particularly their deformability, which may lead to an overestimation of the role of capillary constriction in cancer progression, it has nonetheless proven invaluable for modelling flow in confined spaces, such as microcapillary beds, offering broad design versatility and precise control over dimensions and dynamic experimental conditions[18,20,34–37].

Melanoma cells injected into the microfluidic chip at physiological flow rates exhibited morphological remodelling that was dependent on channel dimensions. Specifically, the extent of cell deformation after passing through the constrictions varied with channel width, with smaller widths inducing greater deformation transmitted at the nuclear level. Interestingly, after constriction <10 µm, there was a retained change in cell shape, suggesting intracellular maintenance of the deformed phenotype. The mechanics and functional consequences of cell and nuclear deformations have been extensively investigated in the context of cancer migration. Although it is well recognized that confined migration can cause DNA damage[13,19,38], evidence suggests that mechanical constriction forces acting on a cell's nuclei can also promote signalling underlying metastatic potential[5,39,40]. In line with a recent study, where the authors found an altered chromatin accessibility after confined

migration[14,41–43], we observed rapid changes in nuclear shape and chromatin compaction after constriction. Constriction caused a decrease in nuclear size with evidence of chromatin compaction through visualising the nucleus using Hoechst stains. Chromatin compaction has been observed previously on account of the mechanoregulation of invasive processes like EMT[44]. Interestingly, constriction led to > 4-fold decrease in the H3K9 acetylation state with a concurrent > 2-fold increase in the H3K9me3 trimethylation state. Previous work has shown how a decrease in H3K9ac corresponds to silencing of somatic cell transcriptional programs and regulation of a slow cycling stem cell-like population[45,46]. Furthermore, a corresponding increase in H3K9me3 has been shown to correspond to increased gene expression, with examples of this histone mark regulating metastatic programs[45,47]. Together, these rapid changes in nuclear architecture and histone marks, due to constriction, demonstrate the plasticity of melanoma cells and the potential for mechanical induction of cell state changes.

Changes in chromatin state invariably correspond to changes in gene expression. Transcriptomics analysis demonstrates immediate regulation of gene expression after constriction, with upregulation of programs associated with invasive behaviour. This is consistent with previous studies that demonstrated cell deformation can augment a cancer cell's migratory characteristics and metastatic potency[48,49]. While our transcriptomic and histone mark analyses support a constriction-induced phenotypic shift, future work incorporating genome-wide profiling of H3K9me3 and H3K27ac will be necessary to precisely map chromatin remodelling and its influence on gene regulatory networks.

At the protein level, we see an increase in the expression of molecular markers associated with a melanoma stem cell-like state. This is interesting because cancer stem cells display remarkable plasticity and are well known to show increased invasive character, along with survival advantages to resist chemotherapy and form new tumours after dissemination[50,51]. Fanfone et al. have studied how breast cancer cells can become more aggressive after nuclear deformation with increased metastatic potency[15]. Additionally, several studies have demonstrated how increased metastatic potency is a hallmark of stem cell-like populations in ovarian cancer[52], pancreatic cancer[53,54], and melanoma[55]. However, a relationship between microcapillary constriction and stem cell-like states had not been previously reported.

To support our hypothesis that microcapillary constriction induces a stem cell-like state, we conducted in vitro and in vivo assays of tumorigenicity. Cells that underwent microcapillary constriction demonstrated a 2-fold increase in tumorsphere growth. This assay has been widely employed to classify the stem fraction in a population of cancer cells, with increased tumorsphere formation corresponding to increased stemness[30]. Experimental metastasis in Balb/c mice using a luciferase reporter cell line further verified the increase in tumorigenicity, with more metastatic lesions occurring faster in animals inoculated with squeezed cells and evidence of melanoma-specific organotropism. This experiment suggests an increase in both

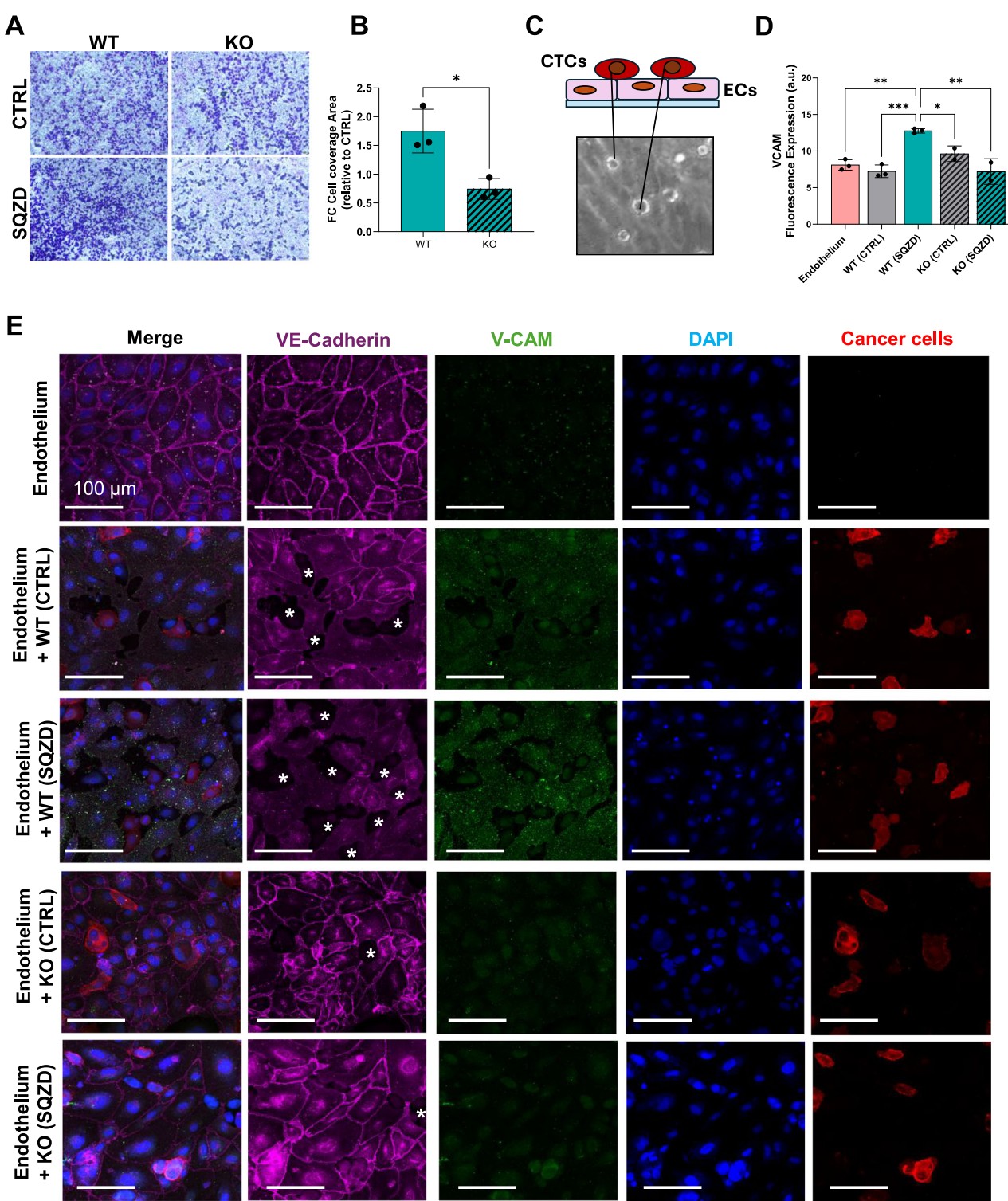

metastatic potency and tumorigenicity, aligned with the emergence of a metastatic stem cell-like population after microcapillary transit. Future studies may explore complementary approaches, such as subcutaneous or orthotopic implantation models to assess tumour growth in vivo, along with longitudinal cell tracking and single-cell multi-omics to better understand how constriction shapes cell behaviour over time.

Mechanistically, the physical deformation experienced during constriction is expected to stretch the cellular membrane, activating mechanosensitive ion channels and associated signalling pathways[56].

Our transcriptomics analysis revealed activation of calcium signalling and multiple transcripts associated with membrane-bound G-protein coupled receptor pathways on account of cell deformation. Using the calcium-sensitive dye Calbryte 520 AM, we saw increased calcium handling during cell constriction, with the degree of deformation correlating with calcium influx. Intracellular calcium levels control integral cellular processes, such as survival, differentiation, and motile activities under physiological or pathological conditions[57]. Abnormal expression and function of calcium channels have been linked to stemness and tumorigenesis[58]. Dysregulated calcium signalling can

**Fig. 6 | Microconstrictions facilitate increased engagement of melanoma stem cell-like cells with endothelial cells. A** Representative image showing Crystal Violet dye signal from WT and KO melanoma cells migrating through the transwell membrane before and after been squeezed into the microfluidic chip. The image highlights the cells that have successfully migrated through the membrane pores and adhered to the bottom surface of the transwell insert. **B** Bar graph showing the fold change of WT and KO cell coverage area, compared to their relative controls. Results represent the mean ± SEM from 3 independent experiments. Statistical significance was assessed using two-sided unpaired t-tests. Asterisks indicate p-values, $p = 0.0142$ (*) 95% CI: [−1.682, −0.3339]. **C** A simplified schematic illustrating the co-culture of melanoma cancer cells with a functional monolayer of endothelial cells. **D** Bar graph displaying the fluorescence expression of VCAM protein from endothelial cells cultured under CTRL condition and with the presence of WT and KO cells, before or after being squeezed into the microfluidic device. The results are expressed as the mean ± SEM from 3 (Endothelium, WT_CTRL, WT_SQZD) and $n = 2$ (KO_CTRL and KO_SQZD) independent experiments. Statistical significance was assessed using one-way ANOVA (two-sided). Asterisks indicate p-values, as follow, Endothelium vs. WT (SQZD) $p = 0.0019$ (**) 95% CI: [−7.288, −2.037]; WT (CTRL) vs. WT(SQZD) $p = 0.0006$ (***) 95% CI: [−8.149, −2.898]; WT (SQZD) vs. KO (CTRL) $p = 0.036$ (*) 95% CI: [0.2070, 6.077]; WT (SQZD) vs. KO (SQZD) $p = 0.0012$ (**) 95% CI: [−2.628, 8.499]. Source data are provided as a Source Data file. **E** Representative fluorescence images of vascular endothelial proteins, e.g., Vascular endothelial cadherin junction (VE-Cadherin) and Vascular Cell Adhesion Protein (VCAM), when the endothelial monolayer is co-cultured with melanoma cells. Experimental numbers and associated statistical analysis for the full data set are shown in panel D. Asterisks are reported in monolayer's area, where VE-Cadherin remodel and leave gap in between cells.

contribute to the aggressive properties of cancer stem cells, including their ability to self-renew, resist treatments, and drive tumour progression[59,60]. Blocking calcium release from the endoplasmic reticulum has been shown to diminish cancer stem cell-like populations in breast tumours[60]. Furthermore, inhibition of calcium influx was shown to decrease the stemness of oral cancer cells[61]. The increased calcium signalling observed in our microcapillary constriction model suggests a relationship between calcium dynamics and the emergence of a stem cell-like state. Consistent with this, when cells were squeezed in calcium-free media, we observed no upregulation of stemness markers, indicating that calcium signalling is required to mediate the phenotypic response to mechanical stress.

At the forefront of calcium influx regulation during membrane deformation is the mechanosensitive ion channel PIEZO1, which serves as a key transducer of mechanical forces into biochemical signals. Activating PIEZO1 leads to an influx of $Ca^{++}$ and an increase in intracellular calcium levels[62,63]. Following microcapillary constriction, we observed an increase in PIEZO1 expression coupled with an increase in its membrane localisation, suggesting that mechanical deformation induces membrane reorganization and PIEZO1 clustering, thereby priming the channel for mechanosensitive activation and facilitating downstream calcium signalling. This mechanotransductive role of PIEZO1 aligns with previous findings in breast cancer cells, where mechanical compression activates PIEZO1 and triggers cytoskeletal reinforcement, ultimately enhancing invasive behaviour[64]. Similarly, our data indicate that melanoma cells respond to confinement by increasing stiffness in a PIEZO1-dependent manner, pointing to a conserved adaptive mechanism. As cells migrate through narrow constrictions, increased tension in the lipid bilayer and actomyosin cortex likely serves as a trigger for PIEZO1 activation[65–67]. This activation initiates calcium-dependent pathways that remodel cellular mechanics, enabling survival and migration under physical stress. Together, these findings suggest that PIEZO1 acts not only as a sensor but also as a regulator of adaptive responses in mechanically challenging environments.

In line with PIEZO1-mediated calcium influx, we observed distinct activation patterns across several mechanosensitive signalling pathways following microcapillary constriction. Specifically, the phosphorylated state of NF-κB and YAP exhibited increased nuclear localisation, consistent with their established roles in transcriptional regulation of plasticity and survival-associated gene programs[68–72]. Nuclear enrichment of mTOR was also observed, supporting its involvement in epigenetic regulation following mechanical stress[73,74]. In contrast, pERK showed a brief increase in nuclear localization, which then decreased, suggesting a rapid transcriptional response that is tightly controlled to prevent prolonged ERK activation, which could otherwise disrupt cellular homoeostasis[75]. Finally, PKC activity exhibited a rapid cytoplasmic spike immediately after constriction, likely driven by calcium influx, followed by downregulation over time as homoeostasis was re-established. Collectively, these early phosphorylation events downstream of PIEZO1

likely converge to promote transcriptional plasticity and support the emergence of a melanoma stem cell-like state. While further studies investigating intermediate regulators, such as pathway-specific inhibitors, may shed light on the broader mechanotransduction network at play, we next sought to functionally validate the role of PIEZO1 itself in orchestrating these downstream responses. To this end, we pharmacologically modulated PIEZO1 activation during constriction. Blocking channel activity with RR during constriction abolished the emergence of the stem cell-like state with diminished capacity to form a tumorsphere in culture. Treatment with GsMTx4, a more selective inhibitor of stretch-activated ion channels, produced similar effects, further supporting the role of mechanically activated calcium influx in driving this phenotype. Conversely, activation of PIEZO1 using Yoda1, even in the absence of deformation, was sufficient to induce stem cell-like marker expression and enhance tumorsphere growth. Previously, activation of PIEZO through treatment with Yoda1 has been shown to enhance the invasiveness in several cancers, including breast and ovarian cancers[76–78]. Recently, PIEZO1 was found to be highly expressed in the CD133 + CD44+ colon cancer tissue and associated with patients in the advanced clinical stage. Knocking down PIEZO1 attenuated these populations tumorigenicity and self-renewal capacity[79]. Our findings extend these observations to melanoma, demonstrating that PIEZO1 activation alone is sufficient to initiate a stem cell-like state and increase tumorigenicity.

To support these results, we used a previously established CRISPR knockout cell line of PIEZO1; similarly, deletion of PIEZO1 led to decreased expression of molecular markers of the stem cell-like state after constriction. Viability analysis showed that KO cells experienced a similar degree of cell death upon deformation as WT cells, indicating that PIEZO1 is not essential for acute survival following mechanical stress. However, in the surviving population, only WT cells exhibited the adaptive acquisition of stemness features and increased tumorsphere formation, while PIEZO1 KO cells failed to upregulate these markers or form tumorspheres, behaving similarly to RR-treated cells. Furthermore, the PIEZO1-KO cells did not show a change in calcium influx during deformation; they returned to a spherical state after transit, and there was no observed decrease in the H3K9ac mark after constriction. Together, these findings demonstrate that PIEZO1 functions as the central mechanosensor coupling transient mechanical deformation to the acquisition of the tumorigenic stem cell-like phenotype in melanoma cells.

Metastasis is mediated by interactions of tumour cells with the endothelium, first during intravasation into circulation, followed by extravasation to a metastatic site to form new tumours, which requires direct engagement via endothelial cell surface receptors. While E-selectin primarily mediates initial tethering and rolling of circulating melanoma cells on the endothelium, especially under inflammatory conditions[80], later stages of extravasation depend on firm adhesion and integrin-dependent signalling involving molecules such as VCAM-1[81]. Adhesion to the endothelium through CD44 has been shown to involve VCAM-1, a transmembrane glycoprotein

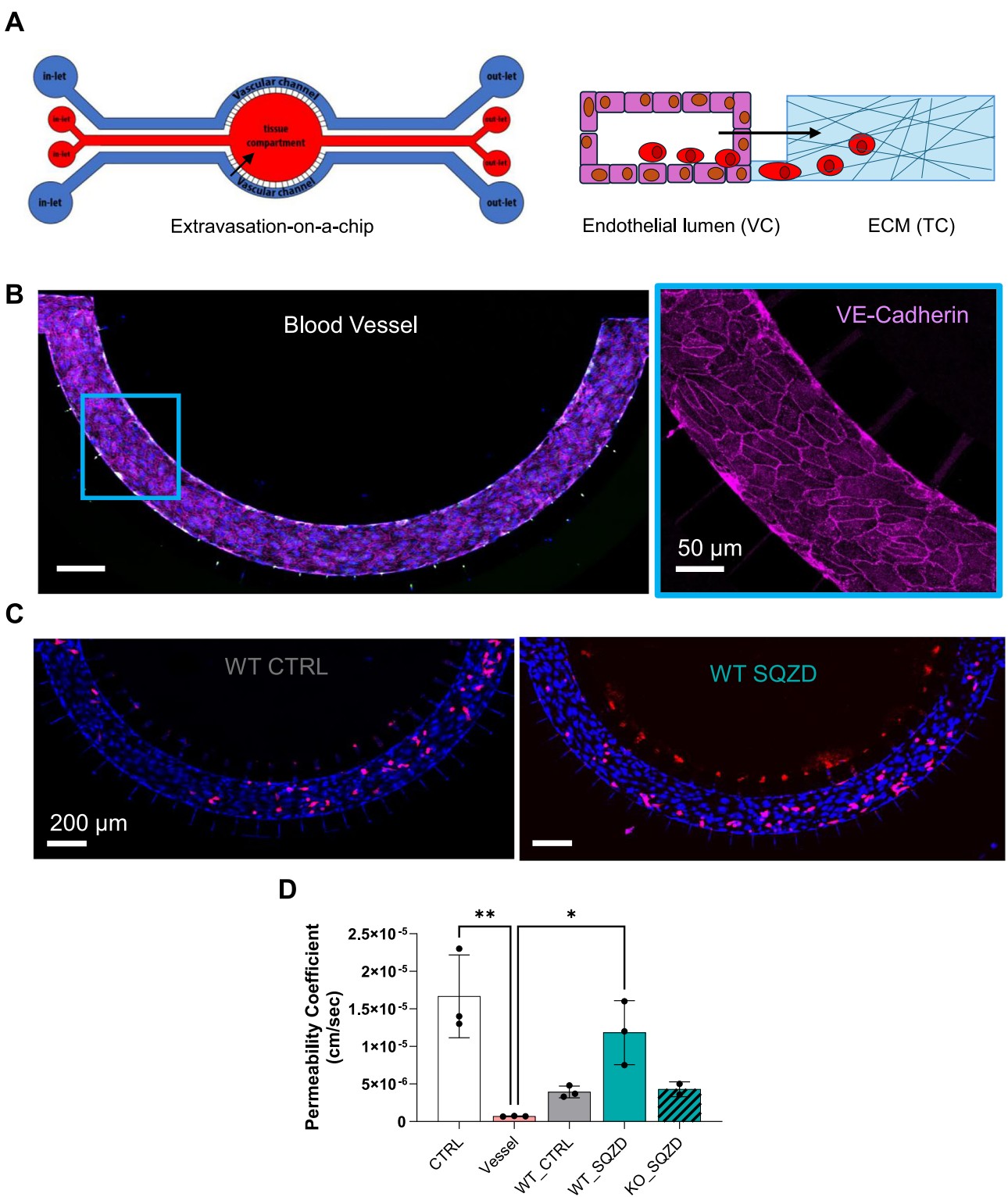

**Fig. 7 | Microconstriction primes melanoma cells to compromise endothelial barrier function. A** Schematic illustrating the microfluidic chip used for melanoma extravasation assay and composed of two independent vascular channels and a central tissue chamber, communicating with a membrane of pores. After a dynamic culture protocol of 3 days to obtain the functional biological barrier into the vascular channel, melanoma cells are injected and allowed to extravasate into the tissue chamber overnight. **B** Representative fluorescence images of the mature blood vessel. Inset showing the expression of VE-Cadherin at the cell-cell border, indicating a functional biological barrier. Images are representative of at least three independent experiments with similar results. **C** Representative fluorescence images of WT melanoma cells (red) after 24 hours of residence in the blood vessel under dynamic conditions. These images illustrate the heightened invasiveness of SQZD WT cells compared to CTRL cells, with most SQZD WT cells having penetrated through the membrane pores into the tissue chamber. **D** Bar graph showing permeability coefficient comparison for CTRL (cell-free device), Vessel, WT-CTRL (Vessel + WT melanoma cells non-treated), WT-SQZD (Vessel + WT squeezed melanoma cells), KO-SQZD (Vessel + KO squeezed melanoma cells). Results are mean ± SEM from 3 independent experiments, except for vessel and KO-SQZD ($n = 2$). Statistical significance was assessed using one-way ANOVA (two-sided). Asterisks indicate p-values as follow, CTRL Vs Vessel $p = 0.0016$ (**) 95% CI: [6.863e−006, 2.508e−005]; Vessel Vs WT_SQZD $p = 0.0169$ (*) 95% CI: [−2.024e−005, −2.030e−006]. Source data are provided as a Source Data file.

implicated in cancer progression and metastatic dissemination in several malignancies, including breast cancer[82,83]. Based on the enhanced metastatic potential observed in our experimental metastasis model, we hypothesized that mechanical constriction might promote tumour-endothelial interactions by enhancing this adhesion pathway. Using custom microfluidic assays, we observed that squeezed melanoma cells exhibited increased adhesion to endothelial monolayers. This was accompanied by upregulation of VCAM-1 expression and disruption of junctional VE-cadherin in endothelial cells, indicating endothelial activation and loss of barrier integrity. Deletion of PIEZO1 abolished these effects, supporting a role for PIEZO1-dependent mechanotransduction in driving both the tumour cell pro-adhesive phenotype and the endothelial response. Importantly, to specifically assess tumour-intrinsic mechanisms, we deliberately avoided pre-stimulation of endothelial cells with TNF-α or other cytokines, which are often used to mimic inflammatory conditions in extravasation models. Remarkably, squeezed melanoma cells alone were sufficient to initiate this endothelial activation, even in the absence of exogenous inflammatory cues. This finding suggests a tumour-intrinsic capacity to modulate the endothelial environment post-constriction, enhancing metastatic potential in a physiologically relevant manner.

While a relationship between PIEZO1 and melanoma trans-endothelium intravasation has not previously been reported, Wang et al. demonstrated that PIEZO1 is a primary driver of trans-endothelial migration of neutrophils[84] and PIEZO1 activity has been shown to coordinate collective migration in keratinocytes[85]. Furthermore, PIEZO1 has been implicated in downstream signalling events that open the endothelial barrier in leucocyte extravasation[84,86]. In line with these findings, our results suggest that mechanical deformation during microconstriction activates PIEZO1-dependent pathways that not only promote a stem cell-like, tumorigenic phenotype but also enhance trans-endothelial invasion in melanoma cells following colonization.

While we primarily focused on the adaptive, tumorigenic phenotype that emerges from cells surviving mechanical deformation, it is important to recognize that PIEZO1 activation may also sensitize circulating tumour cells to immune-mediated apoptosis, as highlighted in recent studies involving TRAIL exposure under fluid shear stress conditions[87,88]. Indeed, in our system, we observed a measurable loss in cell viability immediately after constriction (Fig. 1C), suggesting that mechanical stress may trigger cell death in a subset of cells. However, the long-term transcriptional and phenotypic reprogramming observed in the surviving population underscores the capacity of certain cells to resist apoptosis and adapt, acquiring stem-like and pro-invasive features. Thus, PIEZO1 may represent a double-edged sword in the metastatic cascade, promoting adaptation and metastasis in some cells, while increasing apoptotic vulnerability in others when exposed to immune effectors such as TRAIL-expressing NK cells, macrophages, or dendritic cells.

The present study provides evidence for a relationship between microconstriction, stemness, and extravasation, fostering an emerging mechanism for metastatic spread that supports cell state plasticity as a central hallmark of cancer progression. Conventional models of a cancer stem cell hierarchy do not coincide with the observed frequency and irregularity of metastasis, e.g., there are too few cancer stem cells in a primary tumour or CTC populations to explain disseminated disease, thereby requiring dynamic changes in cell state to occur during intravasation, circulation, and extravasation. Here we show that microconstriction activates signal transduction cascades through PIEZO1, priming cells to adopt an invasive cancer stem cell-like state, that is poised to engage with the endothelium, extravasate and colonise distal tissue sites.

Beyond this extravasation-specific mechanism, mechanical deformation has more broadly been shown to drive nuclear remodelling, chromatin changes, and invasive behaviour; however, these models have largely focused on interstitial matrix confinement rather than the mechanical challenges encountered by circulating tumour cells during microcapillary transit. Our study addresses this gap by modelling the transient, extreme deformations experienced in capillary-scale constrictions, revealing that such deformation alone is sufficient to induce a stem cell-like reprogramming in melanoma cells. Furthermore, preliminary experiments in breast cancer and osteosarcoma cell lines show similar upregulation of stem cell-like features following microcapillary constriction (Supplementary Figs. 15 and 16), albeit with variation in the timing and magnitude of response. These findings suggest that mechanosensitive reprogramming may be a conserved phenomenon across multiple solid tumours, though its kinetics may differ depending on cancer cell type and microenvironment context.

Since PIEZO1 activation alone is sufficient to nurture this phenotype, it is likely that other biophysical cues in the microenvironment could elicit the appearance of similar cell states; for instance, tissue tension and compression, fluid flow, changing matrix mechanics, etc. Considering how PIEZO1 has been found to influence invasive behaviour in other cancers like breast and prostate, it is imperative that this putative mechanotransduction mechanism in regulating stem cell-like phenotype transitions is evaluated in other solid tumours. An important next step will be to examine whether similar mechanisms operate in patient-derived cancer cells and tissues by assessing PIEZO1 expression and its potential correlation with stemness markers and epigenetic changes.

Tracking CTC dissemination in vivo, particularly in tumour regions subject to mechanical compression or in proximity to blood vessels, could offer further insight into the relevance of PIEZO1-mediated signalling under physiologic conditions. Future studies examining PIEZO1 activation and associated signalling in compressed tumour regions or perivascular niches within clinical specimens will be critical to validate the physiological relevance of this mechanism and assess its translational potential in metastatic disease. Furthermore, targeting PIEZO1 or biophysical mechanisms that precede PIEZO1 activation (e.g., capillary-like constrictions) could prove a therapeutic avenue in combating metastatic disease.

## Methods

### Microfluidic device fabrication

The microfluidic device was manufactured by replica moulding, blending polydimethylsiloxane (PDMS) and cross-linker at a ratio of 10:1 and curing the degassed mixture in a SU-8 master mould at 70 °C for a minimum of 2 h. The SU-8 mould was specifically designed for the experiment using CAD software and realized with a soft lithography process. The device features four sets of microchannels arranged in parallel, each set sequentially decreasing in width (30 μm, 20 μm, 10 μm, 5 μm). Each microchannel is 150 μm in length and 15 μm in height, spaced 150 μm apart. After PDMS curing, individual devices were separated by cutting. Inlet and outlet holes were punched using a 1.5 mm puncher before the device was plasma bonded to a microscope glass slide to allow connection with Tygon tubing. Prior to cell injection, the device underwent UV exposure for 30 minutes and was coated with 1% BSA for 30 minutes in an incubator.

### Cell culture

A375-MA2 were obtained from ATCC (Cat. Number CRL-3223). PIEZO1 knockout cells were generated using CRISPR/Cas9 editing, as described[26]. The culture medium was the Dulbecco's Modified Eagle Medium (DMEM, Gibco, Life Technologies, Cat. Number 11965092) high glucose that was supplemented with 10% Foetal Bovine Serum (FBS, Sigma-Aldrich Cat. Number F2442) and 100U/ml aqueous penicillin, 100 mg/ml streptomycin (Sigma-Aldrich, Cat. Number P0781). A375-Luc2 human melanoma cells used for the in vivo model were obtained from ATCC (Cat. Number CRL-1619-LUC2) and cultured as

mentioned above. HUVECs were purchased from Lonza (Cat. Number C2519A). The culture medium was the endothelial basal medium (EBM-2), supplemented with endothelial growth medium (EGM-2) BulletKit from Lonza (Cat. No. CC-3162). To ensure the expression of key endothelial proteins, all experiments used HUVECs between passage numbers 5–7. Both cell types were maintained at 37 °C in a humidified atmosphere containing 5% $CO^2$ until confluency.

## mScarlet-CAAX membrane labelling of A375-MA2 melanoma cells

To visualise A375-MA2 cells with live-cell microscopy, a construct was designed to label the plasma membrane via the expression of mScarlet with the CAAX motif, as previously explained in ref. 89.

## Microcapillary-like constriction experiment

Culture flasks with 80–90% confluence of wildtype and PIEZO1-knockout A375-MA2 melanoma cells were washed with Phosphate Buffered Saline (PBS, Sigma Aldrich), detached using Trypsin-EDTA (0.05%, Gibco, Life Technologies, Cat. Number 25300054) for 5 minutes at 37 °C in 5% $CO_2$ and blocked with DMEM. Cell suspension was collected and centrifuged at 220 x $g$ for 3 minutes at room temperature (RT) and the supernatant was discarded. Cells were then resuspended and counted to prepare a final solution of $10^6$ cells in 3 mL of DMEM that was loaded into a 10 mL syringe. Using Tygon tubing (John Morris Scientific Pty Ltd, Cat. Number 1012028), cells were injected into the microfluidic chip at a physiological flow rate of 260 µl/min, which corresponds to a shear stress of 6 dyne/cm² in the 5 µm microchannel. Squeezed cells were then collected at the outlet using a Tygon tube connected to a 1 mL Eppendorf and centrifuged again before being processed for further analysis.

## Cell viability

Cell viability was assessed using the trypan blue exclusion method (0.4%, Gibco, Life Technologies, Cat. Number 15250061). Following treatment or mechanical constriction, cells were collected, resuspended in PBS, and mixed 1:1 with 0.4% trypan blue solution (Thermo Fisher Scientific). The cell suspension was incubated at room temperature for 2–3 minutes and then immediately loaded into a hemocytometer.

Viable (unstained) and non-viable (blue-stained) cells were manually counted under a light microscope. Total cell number and percentage viability were calculated from at least three independent fields per sample. All samples were processed in triplicate and analysed promptly to avoid staining artifacts.

## Cell morphology analysis

To assess cell membrane deformation after transiting into the microcapillary-like constrictions, images were taken using the Zeiss LSM 800 microscope with the Plan-Apochromat 10× magnification objective (NA 0.45). Specifically, mScarlet-labelled wildtype and PIEZO1-knockout cells were injected and the flow was interrupted to allow imaging of cells at each specific region of the microfluidic device, namely, the Constrictive Channel (CC) and the corresponding Relaxation Chamber (RC). Images were captured using both transmitted light and a 555 nm laser to assess cell position within the constrictive channels and appreciate the corresponding membrane deformation. The deformation index (DI) was used to quantify cell membrane deformation using the following equation:

$$DI = \frac{D_x}{D_0} \qquad (1)$$

Where $D_x$ is the major axis at any x location and $D_0$ is the major axis of the initial cell shape prior to any deformation. The % Median

Deformation was calculated by dividing the median deformation index (DI) of at least 20 cells at various locations within the microfluidic device by the median DI of 20 cells before deformation, then multiplying by 100.

## Real-time deformation cytometry (RTDC)

To assess single cell deformability, experiments were performed using a microscope equipped with AcCellerator (Zellemechanik) module as described previously[90]. Briefly, cells before and after constriction were stained with 2 µL Calcein-AM (ThermoFisher L3224) and incubated for 15 minutes. Next, cells were centrifuged and resuspended in CellCarrier (Zellmechanik, Dresden) medium composed of PBS solution supplemented with <1% methylcellulose to adjust the viscosity (pH = 7.4, Osmolarity = 310–330 mOsm/kg). Cells were then loaded in 1 ml Syringe while another syringe was loaded with 1 mL of fresh CellCarrier medium. Both syringes were connected to tubing and placed on Cetoni pumps (Low-Pressure Syringe Pump neMESYS 290 N with Contiflow Valve). Tubing connected to the plain CellCarier syringe was attached to the sheath inlet of the 30 µm channel FlicXX chip (Zellmechanick, Dresden), and the outlet tubing was connected to the exit line of the chip. Sheath flow speed was adjusted to 0.12 ul/s and sample speed was set to 0.04 µL/s. Sheath to sample flows were set at 3:1 ratio. Both pumps were started, and the fluidic chip was monitored for leaks. If no visible leaks were present, the hip was carefully placed on the Zeiss AxioObserver microscope stage of the system, and orientation and position were adjusted so that 40X/0.65 is just under the narrowest part of the chip, where cells are supposed to be squeezed through. Fluorescent dyes were excited with 488 and 561 nm lasers and fluorescence emissions were collected through BP 525/50 and 593/46 nm, respectively. While there are many hard thresholds that can be imposed on detections, based on cellular morphology, size and fluorescence intensity, we only cut out cells that were smaller than 100 and larger than 1500 micrometres squared, which ensured small cellular debris and large cellular clusters are not collected. A 250 by 100 pixels (0.339 micrometres/pixel) ROI images were collected using a high-speed sCMOS camera (Mikrotron EoSens CL), at 100 fps, while the sample was illuminated with a high-power LED AcCellerator L1 (Zellmechanik, Dresden). More than 10000 cells were collected for each sample and the data stored into.rtdc files, which were further processed in ShapeOut (Zellmechanik) software. Properties, such as Deformation vs Cellular Area, were calculated from collected images and plotted in ShapeOut, where no extra processing, gating filters were imposed. Axes were adjusted to show most of the cellular detection and adjusted to a logarithmic. Young's modulus was estimated based on cell deformation and projected area using a calibrated viscoelastic model, as previously described[91].

## Calcium Imaging

Cells were loaded with 5 µM Calbryte 520 AM dye for 30 min in serum-free media, supplemented with 0.04% Pluronic F-127. Cell suspension was then collected and centrifuged at 220 x $g$ for 3 minutes at room temperature (RT) and the supernatant was discarded. Cells were then resuspended and counted to prepare a final solution of $5 \times 10^5$ cells in 3 mL of DMEM that was loaded into a 10 mL syringe. Using the same protocol of microcapillary-like constriction experiment, cells were injected into the microfluidic chip and were imaged using a CoolLED pE-4000 to illuminate the sample (488 nm illumination) on an Olympus IX70 inverted light microscope fitted with an Andor iXon EM-CCD camera and a 4X objective. Recorded videos were analysed using ImageJ software. A region of interest (ROI) was selected corresponding to the position of the cells within the microcapillary-like constrictions. The maximum fluorescence intensity was then extracted and

normalized to the baseline fluorescence intensity of the cells prior to entering the microconstriction.

## Fluorescence Staining Protocols

To explore the expression of relevant biomarker before and after exposure to the microcapillary-induced cell membrane deformation, wildtype and PIEZO1-knockout A375-MA2 melanoma cells were seeded on a cover glass for 4 hours, before and after being exposed to the membrane deformation within the microcapillary-like constrictions. After this time, cells were fixed in 4% paraformaldehyde (PFA) for 20 min at RT and permeabilized for 10 min in 0.1% Triton X-100. Depending on the specific assay, cells were then incubated overnight at 4 °C with primary antibodies diluted 1:200 in PBS. Specifically, the primary antibodies were against CD44 (ThermoFisher Scientific Australia, Cat. Number MA5-13890), p75 NGF Receptor antibody [8J2] CD271(abcam, Cat. Number AB245134), ABCB5 (Novus Biologicals, Cat. Number NBP1-77687), PRDM14 (Abclonal, Cat. Number A5543), PIEZO1 (ThermoFisher Scientific Australia, Cat. Number MA5-32876) H3K9ac (Cell Signalling, Cat. Number C5B11), H3K9m3 (Cell Signalling, Cat. Number D4W1U). After washing, fluorescent secondary antibodies at a concentration of 2 µg/ml (Goat anti-Mouse IgG (H + L) Secondary Antibody, DyLight™ 488 from Thermo Fischer Scientific, Cat. Number 35502, and Anti-Rabbit IgG (H + L), CF™ 647 antibody produced in goat from Sigma-Aldrich, Cat number SAB4600184) were added for 2 hours, still in the dark at RT. Nuclei were stained with Hoechst (Thermo Fischer Scientific 1:1,000). Images of cells were taken using the confocal Zeiss LSM 800 microscope with the Plan-Apochromat 20× magnification objective (NA 0.8) and Plan-Apochromat oil immersion 40X magnification objective (NA 1.3). A customised MATLAB code (Mathworks, available at https://github.com/ElvPan/VoronoiAndWatershedBased2DSegmentationWithCurvatureAnalysis.) was then used to post-process the images and quantify the protein expression under different experimental conditions.

## Indirect immunofluorescence imaging

For immunofluorescence experiments, 96-well #1.5 glass-bottomed plates (Cellvis, CA) were pre-coated per well with 40 µL of Cell-Tak at 3.5 µg/cm$_2$ in NaHCO3 (pH 8) for two hours. Following incubation, excess Cell-Tak was removed, wells were air-dried, washed twice with 200 µL of MiliQ-H$_2$O and dried completely before use. 48,000 A375-MA2 cells were then seeded into 48 wells of 96-well glass bottom plates, with each well having a total volume of 200 µL. Plates were then spun at 300 $g$ for 10 mins. Following centrifuging, wells were aspirated, and cells were fixed with 4% paraformaldehyde (PFA, Electron Microscopy Sciences) in PBS for 20 min and subsequently permeabilised with 0.1% Triton X-100 (Sigma-Aldrich) for 10 min, followed by second fixation step with 4% paraformaldehyde for 20 mins. Wells were then washed 3 times with phosphate-buffered saline (PBS) prior to initiating immunofluorescence (IF) labelling. IF begins with cell blocking (1 hr in 4% bovine serum albumin (BSA) in PBS (PBS/BSA)) prior to primary antibody incubation (overnight at 4 °C, diluted in 4% PBS/BSA) (all labelling reagents presented in Supplementary Table 1). 4 times washing with 4% PBS/BSA preceded secondary antibody labelling together with 6-Diamidino-2-phenylindole (DAPI, Sigma-Aldrich) and Phalloidin-488 (BioLegend) incubation (45 mins at room temperature, diluted in 4% BSA). 4 times PBS washing preceded confocal imaging. All washing and aspiration steps were performed using a semi-automated 50 TS microplate washer (BioTek). **Confocal Microscopy**. Imaging was conducted on a Nikon AXR confocal microscope using Nikon Jobs software for automation. Images were captured using the 20 x PLAN APO 0.75 NA air immersion objective, with a high-speed resonance scanner at a resolution of 2048 × 2048 with 8x line averaging. Channels were sequentially scanned for minimal crosstalk, with excitations of 405 nm, 488 nm, 555 nm and 647 nm that were detected with 429-

474 nm, 499–530 nm, 571–625 nm and 662–737 nm, respectively. Differential interference contrast (DIC) images were also captured with 647 nm excitation. For each well, 20 images were acquired using the Nikon Perfect Focus System (PFS) to maintain optimal focal stability. A z-stack was initially acquired using DAPI and phalloidin staining to identify the best focal plane with the strongest signal. Subsequently, a single z-slice corresponding to the optimal focal plane was selected for 4-channel image acquisition. Image data was output as 16 bit.ND2 files per well. **Image analysis and downstream quantitative analyses**. ND2 images were converted to single-channel 16 bit.tiff images using nd2 handler, a Python-based script. Illumination correction was performed using a Python implementation of the BaSiC algorithm {https://github.com/linum-uqam/PyBaSiC, https://doi.org/10.1038/ncomms14836} with default parameters. This process estimated flatfield and darkfield images for each fluorophore from the image stacks, which were then used to correct the raw single-channel.tiff files. These corrected raw single-channel.tiff files were further processed using using customised CellProfiler workflows (version 4.2.8, Broad Institute, MIT) (https://doi.org/10.1186/gb-2006-7-10-r100). Nuclei were segmented using minimum cross-entropy thresholding (MCET) of DAPI signals, with nuclear masks seeding propagation-based thresholding of cell bodies using MCET on phalloidin signals. Cytoplasm domains were calculated by subtracting nuclei from cell bodies. Quantitative features were measured per cell, capturing information on per-marker intensity and subcellular localisation. Per-cell nuclear-to-cytoplasmic (N: C) ratios were calculated by dividing the mean nuclear intensity by the mean cytoplasmic intensity for each marker, prior to log10 transformation. Single-cell data was parsed and analysed using a custom Knime workflow (KNIME AG, v4.5, Zurich). First, quantitative outlier cells were detected and removed based on the interquartile range of each feature, R = [Q1 - k(IQR), Q3 + k(IQR)] with IQR = Q3 - Q1 and k = 1.5. Second, hypothesis-driven parsing used linked quantitative and image datasets, allowing for identification of segmentation artefacts based on extreme value combinations (e.g., low cell area with high cell perimeter values; typical of noisy segmentation). Third, data-driven parsing used robust Z-normalized data, prior to principal components analysis (PCA; retaining 97% of total variance), prior to UMAP[92] manifold embedding. This highlighted outlier cells and cell clusters in UMAP space, followed by observation of matched image data to determine outlier status. After data parsing, feature values were normalized by centering each value around the control group median (i.e., norm value = raw value − control-median), computed per marker and time point in the line plot. Mann-Whitney U tests were performed to compare feature values between control and squeeze groups, with $p$-values adjusted for multiple testing across timepoints using the Benjamini-Hochberg false discovery rate correction method to control the expected proportion of false positives. Results are presented in box-plots, line plots or heatmaps. Heatmaps represent median log10-transformed fold-changes (relative to control medians) in nuclear intensity, cytoplasmic intensity, and nuclei-to-cytoplasmic (N/C) ratios across timepoints and treatments for each marker, with values cantered around the control group median for each marker and timepoint, visualized using a colour gradient to indicate the magnitude and direction of changes in cell signalling response to mechanical squeeze.

## Tumorsphere assay

To assess the self-renewal and stem-like properties of wildtype and knockout A375-MA2 melanoma cancer cells under control and squeezed experimental conditions, we employed a tumorsphere assay using ultra-low attachment culture plates. Melanoma cells were cultured in standard conditions as explained above. Prior to seeding them into the serum-free media (R&D Systems™ StemXVivo Serum-Free

Tumorsphere Media, Cat. Number CCM012), cells were counted and exposed to membrane deformation within the microcapillary-like device, following the protocol explained above. Control and squeezed cells were then seeded in 96-well ultra-low attachment plates at a concentration of approximately $3 \times 10^3$ cells per well. The plates were incubated at 37 °C with 5% $CO_2$, and tumorsphere formation was monitored over 7 days, changing media each 48 hours. Tumorspheres were observed using bright-field microscopy and the size of spheres was quantified using ImageJ software.

For experiments involving pharmacological manipulation of PIEZO1, cells were plated in 96-well plates as detailed in the "Drug Treatment" section for 4 hours and then detached using Trypsin-EDTA for 5 minutes at 37 °C in 5% $CO_2$ and blocked with DMEM. Cell suspension was collected and centrifuged at $220 \times g$ for 3 minutes at RT and the supernatant was discarded. Cells were then seeded in 96-well ultra-low attachment plates at a concentration of approximately $3 \times 10^3$ cells per well in serum-free media, following the culture protocol explained above.

## Drug Treatment

To assess the role of PIEZO1 in stem-like and tumorigenicity state of A375-MA2 melanoma cancer cells, pharmacological manipulation using activator Yoda1 (Sigma Aldrich, Cat. Number SML1558 reconstituted in DMSO) and blocker Ruthenium Red (abcam, Cat. Number ab120264) was performed. Specifically, cells were seeded in 96-well plate at a concentration of $20 \times 10^3$ cells in 100 μL per well. Cells were allowed to attach, and media was replaced with 20 μM of Yoda1 to activate PIEZO1, or 30 μM of RR to inhibit PIEZO1 activity, both in DMEM. Cells were maintained at 37 °C in a humidified atmosphere containing 5% $CO^2$ for 4 hours and then fixed and stained for immunofluorescence staining as explained in the following section "Fluorescence Staining Protocols". For functional assessment of PIEZO1 activation under mechanical stress, blockers were also applied during the constriction experiments. Specifically, for microcapillary constriction assays, cells were prepared and suspended in DMEM according to the protocol described in the "Microcapillary-like constriction experiment" section. Immediately prior to injection into the microfluidic device, either Ruthenium Red (30 μM) or GsMTx4 (10 μM) was added directly to the cell suspension, allowing continuous exposure to the inhibitor during the constriction procedure. Following transit through the microcapillary device, cells were immediately collected and seeded into 96-well plates under normal growth conditions without additional inhibitor treatment.

For calcium-deprivation experiments, cells were suspended in calcium-free DMEM (Gibco, Cat. Number 21068-028) throughout the constriction procedure to assess the role of extracellular calcium influx during mechanical stimulation. After constriction in calcium-free media, cells were similarly seeded for downstream analysis.

## RNA sequencing

Samples were collected immediately after each experimental condition and total RNA was isolated using Qiagen RNeasy mini kit following manufacturer's instructions. Bulk RNA sequencing and subsequent bioinformatic analyses were conducted at the Australian Genome Research Facility. RNA integrity was evaluated using the Agilent TapeStation system with RNA ScreenTape. Library preparation was carried out using 500 ng of total RNA per sample, following the TruSeq Stranded RNA protocol. Quantification of libraries was performed via qPCR, and sequencing was executed on the Illumina NovaSeq X Plus platform, generating 150 bp paired-end reads. FASTQ files were produced after adapter trimming, and reads were aligned to the human reference genome (GRCh38) using STAR (v2.3.5a). Transcript quantification was achieved with StringTie (v2.1.4), and differential gene expression analysis was conducted using edgeR (v3.38.4). Genes with pvalue ≥ 0.01 and with logFC(SQZD-CTRL)

above 0.6 or below −0.6 were noted as differentially expressed genes (DEG). Pathway analysis, gene ontology (GO) enrichment analysis for biological processes was performed with David Functional tool on top 100 genes with $p$-value ≥ 0.01, and clustering analysis of upregulated genes was done with Cytoscape.

## Transwell migration assay

The transwell migration assay was used to assess the ability of WT and KO A375 MA2 melanoma cancer cells, under control and squeezed conditions, to migrate through an 8 μm pore size membrane, simulating their potential for invasion and metastasis. Melanoma cells were cultured in standard conditions as explained above. Prior seeding them into the transwell chambers insert into a 24-well plate, cells were counted and exposed to membrane deformation within the microcapillary-like device, following the protocol explained above. Control and squeezed cells were then resuspended at a concentration of 80,000 cells in 300 μL of serum-free media and seeded into the upper chamber of the transwell insert. The lower chamber was filled with complete medium to attract the cells, and the plate was incubated overnight at 37 °C with 5% $CO_2$. After incubation, the transwell inserts were removed and cells in the upper chamber were gently washed with PBS. Cells were then fixed with 4% PFA for 20 minutes and permeabilized using methanol for 1 hour at RT. Following a PBS washing, cells were then stained with crystal violet (Sigma-Aldrich, Cat. Number C0775) for 1 hour at RT. To visualize migrated cells, the inserts were placed back into a 24-well place and observed with a brightfield microscope. To assess migration under different experimental conditions, a custom MATLAB code was then used to post-process the images and quantify the cell's coverage area from the insert.

## In vitro co-culture experiment

To investigate the metastatic potential of circulating melanoma tumour cells experiencing membrane deformation, an in vitro co-culture of melanoma cancer cells with endothelial cells was specifically designed. A perfusable platform was realized by replica moulding blending PDMS and cross-linker at a ratio of 10:1 and curing the degassed mixture on a 3D printed mould with a rectangular geometry (height of 1 mm, width of 1 cm and length of 2.5 cm) at 70 °C for a minimum of 2 h. After PDMS curing, individual devices were separated by cutting. Inlet and outlet holes were punched using 1.5 mm puncher before the device was plasma bonded to a microscope glass slide to allow connection with Tygon tubing. Prior to cell injection, the device underwent UV exposure for 30 minutes and was coated with 100 μg mL−1 fibronectin (ThermoFisher Scientific Australia, Cat. Number 33016015) for 1 hour at 37 °C.

Culture flasks with 80-90% confluence of HUVEC cells were washed with Dulbecco Phosphate Buffered Saline (PBS, Sigma Aldrich), detached using Trypsin-EDTA (0.05%, Gibco, Life Technologies) for 5 minutes at 37 °C in 5% $CO_2$ and blocked with EGM-2. Cell suspension was collected and centrifuged at 220 x $g$ for 3 minutes at RT and the supernatant was discarded. Cells were then resuspended and counted to prepare a final solution of $10^6$ cells in 1 mL of EGM-2 that was loaded into the PDMS platform using Tygon tubing and placed into the incubator. After 4 hours, the media was replaced to washed cells that didn't bind the fibronectin and the platforms were kept at 37 °C in 5% $CO_2$ for three days, changing the media each day and allowing the maturation of a functional endothelial monolayer inside the PDMS platform. After 3 days, culture flasks with 80-90% confluence of wild-type and knockout A375-MA2 melanoma cells were washed with Dulbecco Phosphate Buffered Saline (PBS, Sigma Aldrich), detached using Trypsin-EDTA (0.05%, Gibco, Life Technologies) for 5 minutes at 37 °C in 5% $CO_2$ and blocked with DMEM. Cell suspension was collected and centrifuged at $220 \times g$ for 3 minutes at room temperature (RT) and the supernatant was discarded. Cells were then resuspended and counted to prepare a final solution of $5 \times 10^5$ cells in 1 mL of DMEM that was

loaded into the PDMS platform and kept at 37 °C in 5% $CO_2$. After 1 hours, to allow cancer cells interact with the endothelial monolayer, a 5 mL syringe of EGM-2 was connected to the system and placed on a syringe pump (New Era NE-4000 Double syringe pump, Adelab Scientific) inside the incubator. A gentle flow rate of 1 mL/min was perfused overnight to prevent detachment of cancer cells. After this time, cells were fixed and stained following the Fluorescence Staining Protocols explained above.

### Development of an artificial blood vessel

The procedure to form an artificial blood vessel into the microfluidic chip was adapted from ref. 93 and here described briefly. A mixture of PDMS and curing agent (10:1 w/w) was degassed and poured over a custom-designed SU-8 mould created using CAD software. The mould design featured a central tissue region (1575 μm width × 100 μm height) flanked by two parallel vascular channels (each 200 μm wide × 100 μm high), separated by an array of pores (25 μm × 25 μm, spaced every 50 μm) that enabled diffusion between compartments. After curing at 70 °C for 2 hours, the PDMS structures were removed, trimmed, and bonded to glass coverslips via oxygen plasma treatment, following 30 minutes of UV-sterilization.

Endothelial cells (HUVECs) were expanded to 80–90% confluence, harvested using 0.05% trypsin-EDTA, centrifuged at $220 \times g$ for 3 minutes, and resuspended in EGM-2 media at a density of $10^8$ cells/mL. Devices were first primed with PBS and coated with fibronectin (200 μg/mL) at 37 °C for 2 hours to promote cell adhesion. HUVEC suspensions were then loaded into the vascular channels using Tygon tubing connected to a syringe pump (NE-4000, Adelab Scientific). Following injection, channels were sealed by clamping the inlet and outlet, and devices were incubated for 4 hours under static conditions to allow cell attachment. To support endothelial monolayer formation and the maturation of junctional proteins, a dynamic flow regime was applied continuously over 72 hours, with average wall shear stress calculated according to the method described in ref. 93.

For functional validation, endothelial junctions were examined by immunostaining for VE-cadherin and VCAM-1. Devices were perfused with PBS at 0.5 μL/min, fixed with 4% PFA, permeabilized with 0.2% Triton X-100, and stained overnight at 4 °C using primary antibodies R&D Systems Human VE-Cadherin Affinity Purified Polyclonal Ab (In Vitro Technologies Pty Ltd, Cat. Number RDSAF938) and VCAM-1 Recombinant Rabbit Monoclonal Antibody (ThermoFisher Scientific Australia, Cat. Number MA531965) at 5 μg/mL. Fluorescent secondary antibodies (Donkey Anti-Goat IgG H&L (Alexa Fluor® 647) (abcam, Cat Number ab150131) and Goat anti-Mouse IgG (H + L) Secondary Antibody, DyLight™ 488 (Thermo Fischer Scientific, Cat. Number 35502), 2 μg/mL) and Hoechst (Thermo Fischer Scientific 1:1,000) were then injected for 2 hours at RT, and images were acquired using a Zeiss LSM 800 confocal microscope with 20× objective (NA 0.8). Z-stacks were stitched to reconstruct the full channel geometry.

### Extravasation-on-a-chip assay

To explore the metastatic progression of circulating melanoma tumour cells experiencing membrane deformation, an extravasation-on-a-chip model was specifically designed. After completing maturation of the artificial blood vessels, WT and KO A375-MA2 melanoma cells under different experimental conditions (CTRL and SQZD) were resuspended to a concentration of $5 \times 10^6$/mL in EGM-2 media and injected into one of the vascular channels, functionalized with endothelial cells to form the blood vessel, using Tygon tubing connected to a 1 mL syringe. The system was left under static conditions for 30 minutes to allow cells to adapt and interact with the endothelium at 37 °C and 5% CO². After this time, the syringes were placed on a syringe pump inside the incubator, and a gentle flow rate of 0.5 μL/min was pulled overnight to prevent detachment of cancer cells. To create a gradient of nutrients and attract melanoma cancer cells to migrate

across the endothelial barrier, the parallel empty vascular channel was perfused with DMEM with 4 ng/ml of transforming growth factor beta (TGF-β) at a flow rate of 0.5 μL/min overnight. After 24 hours, the blood vessel was either used for a vascular permeability assay or fixed and stained following the previously explained protocol for quantifying extravasation of mScarlet-labelled melanoma cancer cells through the biological barrier. Confocal images of different portions of the endothelium were acquired using confocal Zeiss LSM 800 microscope and a Plan-Apochromat 20× magnification objective (NA 0.8) and successively stitched together to reconstruct the whole channel.

### Vascular Permeability assay

To evaluate endothelial barrier function under various experimental conditions, a vascular permeability test was conducted using fluorescent tracer diffusion, based on method described in ref. 93. Upon completion of the extravasation-on-a-chip assay, the microfluidic device was detached from the syringe pump. A 1 mL syringe containing Texas Red–dextran (40 kDa, Thermo Fisher Scientific, Cat. No. D1829) diluted to 10% in culture medium (from a 10 mg/mL stock) was connected to the vascular channel inlet. The tracer was perfused through the vascular network at a flow rate of 0.5 μL/min while live imaging was performed using a Leica TCS SP8 DLS confocal microscope housed in a temperature- and $CO_2$-controlled incubator (37 °C, 5% $CO_2$). To monitor tracer extravasation, time-lapse imaging was acquired every 2 minutes over a 90-minute period. Signal accumulation in the adjacent tissue compartment was captured using a 10× objective (HC PL Apo L, NA 0.3), scanning the x–y plane to cover the entire device area. Post-processing and quantification were carried out in ImageJ. The permeability (P) was then quantified based on method described in ref. 93.

### In vivo metastasis model and optical imaging of bioluminescent metastasis

All experiments were approved by and conducted in accordance with the UNSW Sydney Animal Care and Ethics Committee (approval: ACEC 22/85B). Seven-week-old female athymic nude mice (Ozgene ARC Australia) were housed in groups of five per cage under specific pathogen-free (SPF) conditions, maintained at 21–23 °C with 40–60% relative humidity and a 12-hour light/dark cycle. Food and water were provided ad libitum. For the experimental model of metastasis, A375-Luc2 human melanoma cells (ATCC CRL-1619-LUC2), subjected or not to the mechanical deformation within the microcapillary-like constriction device, were injected at a density of $5 \times 10^5$ cells/animal into the lateral tail vein of 7-week-old female BALB/c nude mice. For bioluminescence imaging, mice were injected intraperitoneally with 15 mg/ml PierceTM D-Luciferin (Thermo Fisher Scientific) at 10 μl/g at indicated timepoints and placed in a light tight chamber. A cooled CCD camera apparatus (IVIS, Xenogen) was used to detect photon emissions with an acquisition time of 2 min. Analyses of the images were performed with Living Imaging software (Xenogen). Thirty days after iv tail injection, mice were euthanized, lungs were collected and fixed in 4% paraformaldehyde for 24 h before storage in 80% ethanol at 4C until further processing. Fixed lungs were embedded in paraffin and 5 μm slices were prepared at the Biological Specimen Preparation Service at the Kataharina Gaus Light Microscopy Facility, Mark Wainwright Analytical Centre, UNSW Sydney. Samples were stained with Haematoxylin and Eosin (H&E) for visualisation of metastatic nodules and lung architecture.

### Intracardiac injection of A375 melanoma cells in athymic nude mice

To establish a metastatic xenograft model, intracardiac injections were performed using A375 Luc2 human melanoma cells. A single-cell suspension containing $1 \times 10^5$ cells in 100 μL of sterile 1X PBS, subjected or not to the mechanical deformation within the microcapillary-like

constriction device, was prepared for each mouse. Seven-week-old female athymic nude mice (ABR, Australia) were anesthetized with 3% isoflurane in 98% oxygen, administered at a flow rate of 2 L/min. Thermal support was provided throughout the procedure to maintain normothermia, and animals were monitored until full recovery from anaesthesia. All experiments were approved by and conducted in accordance with the National Health and Medical Research Council Statement on Animal Experimentation, the requirements of New South Wales State Government legislation, and the rules for animal experimentation of the Biological Testing Facility of the Garvan Institute and the Victor Chang Cardiac Research Institute (protocol #24_25). **Injection Procedure**. Mice were positioned supine, and both forelimbs were gently extended and immobilized perpendicular to the midline. The thoracic area was disinfected using 70% ethanol swab. Anatomical landmarks, specifically the xiphoid process and the jugular notch, were identified and marked using a surgical pen. A third mark was placed midway between these two landmarks, slightly to the animal's left (ride side when facing the animal), at the level of the third intercostal space. This location, approximately 1-2 mm left of the midline and between the third and fourth ribs, served as the injection site. A 0.3 mL syringe fitted with a 29-gauge needle was used to load the 100 μL cell suspension. A small air bubble was included at the end of the syringe to assist in verifying correct needle placement via visual confirmation of blood pulsation. The needle was inserted vertically at the marked injection site. Successful entry into the left ventricle was indicated by the appearance of bright-red, pulsatile blood within the needle hub or cell suspension. If no pulsation was observed, the needle was repositioned. The cell suspension was injected slowly over a period of 40-60 seconds. Following the injection, mice were returned to clean pre-warmed cages and observed continuously until they had fully recovered from anaesthesia. One hour post-injection, mice were imaged using the IVIS Spectrum imaging system to confirm the successful delivery of luciferase-labelled cells into circulation. **Bioluminescence Imaging for Metastatic Tracking**. In vivo tracking of metastatic progression was performed using the IVIS Spectrum imaging system (PerkinElmer) on luciferase-expression A375 xenografts. Mice were imaged weekly and at the terminal endpoint. Prior to each imaging session, mice were weighed and injected intraperitoneally with D-luciferin potassium salt (10 μL/g body weight; 15 mg/mL stock solution, Sigma-Aldrich, #LUCK-2G). Imaging was conducted 10 minutes post-injection under isoflurane anaesthesia, using auto-exposure settings and D magnification. During weekly imaging, mice were scanned from both dorsal (whole body including head) and ventral (whole body including legs) views to assess metastatic spread. At the endpoint, whole-body imaging was repeated using the same views. Additionally, ex vivo bioluminescence imaging was performed on isolated organs, specifically the brain, skull, lungs, heart, and legs, to confirm and localize metastatic lesions. **Assessment of Disease Progression and Humane Endpoints**. Following the intracardiac injection of cells, animals were monitored closely for the first three days post-injection to ensure proper recovery and to detect any immediate adverse effects. After this initial period, mice were monitored twice weekly for changes in body weight and body condition score (BCS). To assess disease progression, bioluminescent imaging using the IVIS system was performed weekly to track the development and spread of metastases. As clinical signs became apparent, monitoring frequency increased to daily, with assessments including weight, BCS, and detailed clinical observations. This study was designed as a global survival experiment. Therefore, the end of the experiment for each animal was determined based on pre-established humane endpoint criteria. Humane endpoints were defined by the presence of clinical, behavioural, or physiological signs indicating pain, distress, or significant deterioration in well-being. Specific parameters included: Hind limb paralysis or partial paralysis, lethargy or reduced responsiveness, weight loss of ≥20% of initial body weight, low BCS (≤2/5) indicating marked loss of muscle

and fat mass, respiratory distress (e.g., laboured or irregular breathing), and pain-associated behaviours such as hunched posture, vocalization, or guarding behaviour. Animals that met any of these criteria were deemed to have reached the humane endpoint and were humanely euthanized in accordance with institutional animal care guidelines.

## Quantification and Statistical Analysis

Unless otherwise noted, all experimental results are from at least three independent experiments. Error bars represent the standard error of the mean (SEM). For comparisons between two groups, a two-sided unpaired t-test was used. For comparisons involving two factors, two-way ANOVA with Tukey's post hoc test was used. All analyses were performed using GraphPad Prism v8.2.0 (GraphPad Software). Statistical details are provided in the figure legends. $P$-value is reported for statistical significance. Comparisons between samples were considered to be statistically significant if the $p$-value was $*p < 0.05$, $**p < 0.01$, $***p < 0.001$, $***p < 0.0001$. No statistical method was used to predetermine sample size. Sample sizes were chosen based on standard practice in the field and observed effect sizes. No data were excluded from the analyses. The experiments were not randomized.

## Reporting summary

Further information on research design is available in the Nature Portfolio Reporting Summary linked to this article.

## Data availability

The datasets generated and/or analysed in this study are available in the Genome Sequence Archive (Genomics, Proteomics & Bioinformatics 2021) in the National Genomics Data Centre (Nucleic Acids Res 2022), under accession code HRA009642. Source data are provided with this paper.

## Code availability

The custom code used for the analysis of fluorescence marker signals from microscopy images is available on GitHub at https://github.com/ElvPan/VoronoiAndWatershedBased2DSegmentationWithCurvatureAnalysis.

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

## Acknowledgements

This work was supported through funding from the National Health and Medical Research Council Grant APP1185021 (K.A.K.) and the National Cancer Institute of the National Institutes of Health Grant R01CA251443 (K.A.K.). The authors acknowledge the help and support of staff at the Katharina Gaus Light Imaging Facility (KGLMF) of the UNSW Mark Wainwright Analytical Centre. The authors also acknowledge the facilities and the scientific and technical assistance of the UNSW node of the National Imaging Facility and the UNSW Node of the Australian National Fabrication Facility. The authors would like to thank Prof. J. Justin Gooding, Prof. Maria Kavallaris and Prof. Martina Stenzel for helpful discussions over the course of this work.

## Author contributions

G.S. and K.A.K. conceived the ideas and initiated the study. G.S. designed and carried out experiments across the entire range of approaches, analysed data, prepared figures, and organized the preparation of the manuscript. C.K., V.R. and C.L.C. designed and performed the animal experiments and analysed the animal data.

S.R. contributed to the RNA sequencing analysis and figure preparation. E.P. designed and performed the real-time deformation cytometry (RTDC), A.D. and J.G.L. designed and performed the phosphoproteomic profiling. K.P. provided CRISPR-edited cells and contributed reagents for mechanistic studies. K.A.K. supervised all aspects of the work and contributed to experimental design and manuscript organisation. All authors contributed to the writing of the manuscript.

## Competing interests

The authors declare no competing interests.
