## [Transparent Peer Review file · Nature Communications]

Capillary constrictions prime cancer cell tumorigenicity through PIEZO1

Corresponding Author: Professor Kristopher Kilian

Version 0:

Reviewer comments:

Reviewer #1

(Remarks to the Author)

This manuscript addresses the effect that compressive squeezing forces (such as experienced in capillary transport) have on tumorigenicity and other functions of melanoma cells. There are some interesting observations and implications presented here. The authors address the role of Piezo1 using knockout cells and also using the chemical agonist Yoda1. My specific comments are given below:

1. I am very interested to know how the time between mechanical stimulus and analysis will affect the results, as this has been shown to be a dynamic process in previous studies where even a relatively short mechanical deformation stimulus can continue to have a varying effect on cells for hours or even days afterwards.
2. Related to that point, The authors should measure the mechanical properties of the cells before and after squeezing stimulus, as it has been previously shown that cancer cells change their stiffness and viscoelasticity in response to mechanical stress.
3. The authors should discuss recent studies showing that fluid shear stress activation of Piezo1 in various types of cancer cells, designed to be a model of circulating cancer cells such as (presumably) in this current manuscript and its context of capillary transport, of how Piezo1 activation is also known to sensitize cancer cells to undergo apoptosis via TRAIL exposure. Since squeezed cancer cells will naturally encounter TRAIL on the surface of immune cells such as natural killer cells, macrophages and dendritic cells, might an increasing propensity to undergo apoptosis counteract the enhanced tumorigenicity shown here?
4. Does the capillary squeezing also phosphorylate NF-kB and Zap70 in a Piezo1-dependent manner as found in other mechanical activation of circulating cancer cells?
5. It would be interesting to see the outcome of introducing the squeezed melanoma cells in a cardiac injection model of metastasis rather than tail vein, to see if this model recreates any of the organ-specific metastasis sites of melanoma. If it doesn't, what would that imply for the proposed importance of capillary squeezing?
6. I did not see much discussion of a major limitation of the in vitro capillary squeezing system, namely that the microfluidic chip is not deformable like an in vivo capillary. The implication is that this study might overestimate the importance of this type of stimulus in the overall cancer development.
7. Has anyone ever documented calcium influx in response to cancer cells caught or passing through mouse capillaries, via intravital microscopy? It seems that if this was a predominant, first-order effect, that it would have been found by others examining the passage of CTCs.
8. Might have missed this, but if cells can be directly observed in the squeezing chip, it should be possible to observe cells "mid-squeeze" instead of just analyzing the whole population afterwards. In this manner, the degree of cell polarization (more calcium etc. in the leading edge vs. the trailing edge, or vice versa). It would be quite interesting to determine to what

degree such polarization occurs, and how important polarization might be for the downstream processes. Also interesting would be to try and modulate polarization by changing the direction of flow in the chip...

9. The quantification of nuclear deformation is interesting... others have shown that nuclear structural proteins such as Lamin A/B are important for conferring mechanical survival to circulating breast cancer cells when subjected to pulses of high shear stress.

10. The effect of squeezing force on endothelial cell interactions is also interesting. To what degree is this dependent on E-selectin? Others have shown a distinct E-selectin dependence of melanoma adhesion to stimulated endothelium. Speaking of, I don't see that the endothelium used in these studies have been pre-stimulated into an inflammatory state (believed to be important for blood borne metastasis) via TNF-alpha or other signal.

11. The stem cell marker emergence is notable, and is consistent with research showing that stem cell markers emerge in breast cancer cells in conjunction with up regulation of cell:cell adhesion proteins and glycoproteins such as E-selectin ligand-1 and E-cadherin.

Reviewer #2

(Remarks to the Author)

Revision Silvani et al., Nat comm

Silvani and colleagues show that capillary-like constriction triggers phenotypic changes in melanoma cells toward a tumorigenic stem cell-like state through PIEZO1 mechanosensation. Authors show that, upon constriction melanoma cells display membrane deformation, which is partially recovered, chromatin condensation, and changes in chromatin marks, with reduced H3K9Ac and increased H3K9me3. Moreover, after crossing the microcapillaries, cells show increased expression of molecular markers of stemness, enhanced interactions with endothelial cells, and increased tumor sphere formation capacity. Using pharmacological agents and gene editing approaches, they conclude that PIEZO1 is responsible for these phenotypic changes.

Findings are intriguing and in line with recent studies showing mechanical compression as a driver of phenotypic changes in cancer cells including fostering metastatic potential. However, most of the conclusions are only backed by experiments in a single cell line, which is only representing a subtype of melanoma, and the in vivo studies are weak and not sufficiently powered to make any valid conclusions. Also, PIEZO modifying cancer phenotype is not fully novel.

Major issues:

1. Unclear if findings are generalizable across cancer types. Authors make general statements, most of the study was conducted on a single cell line (A375-M2). There is insufficient justification for using melanoma as a model and how much the findings apply to other cancer types. Therefore, authors should conduct studies in other melanoma cell lines, and in other cancer types if they want to make general claims beyond melanoma. Otherwise, the conclusions should be tempered, and the word 'melanoma' added to the title.
2. The metastasis experiments presented are inadequate. Tail vein injection is not a true model of metastasis; cancer cells are trapped in the lung capillaries and therefore this assay is more indicative of tumor cell seeding. Intracardiac injection or intradermal injection followed by survival surgery would truly recapitulate the metastatic process. Also, the number of mice (n= 4 per group) is clearly insufficiently powered for this type of assays. Differences in Fig 3F are not statistically significant. A true quantitation of the number of metastatic foci and the size of those mets (tumor burden) is missing.
3. Mechanistically, the study lacks important aspects: How PIEZO protein levels or subcellular localization change in response to microcapillary 'squeezing'? Is the mechanism Ca²⁺-dependent or PIEZO1 conformation dependent. How does PIEZO1 change results in H3K9me3 increase and expression of stem-like genes?
4. Limited evidence of clinical relevance. Authors should provide patient-derived data supporting their findings.

Other specific points to be addressed:

Figure 1

- 1.1. It is not clear why authors decided to use A375-MA2 melanoma cells which are an increased metastatic version of the parental A375. If the scope of the study is to prove that compression increases metastatic potential, authors should have used the parental A375 which are less metastatic. Also, in line with my comment above, confirming some key findings in additional melanoma cell lines and at least other 2 cancer models (i.e., breast, lung, ovarian) it is necessary to make general conclusions.
- 1.2. No justification is provided for choosing gradient 30-5um channels; are these sizes based on physiological observations? How long time is required for the cells to pass through the whole device?
- 1.3. The viability was measured exclusively by Trypan blue (according to material and methods) which can create bias in immunofluorescence staining. In addition, what is the control? Are cells subjected to the same flow rate but in a channel-free device or cells at the inlet? This can create a strong biological bias between control and squeezed cells. Please provide a clear description of controls.
- 1.4. Authors should perform H2AX to evaluate the activation of DNA damage, total H2AX is insufficient.
- 1.5. How long after deformation histone modifications are maintained?
- 1.6. In line with my previous concern, immunofluorescence should be repeated in the presence of phospho-Caspase3 to

quantify the increase in Hoechst and H3K9me3 within alive cells.

1.7. Control and SQZD cells should be assessed by H3K9me3 ChIP-seq or CUT&Tag/Run to determine if/how H3K9me distribution across the genome changes upon microcapillary induced constriction. These analyses would reveal if the observed changes in histone marks are (at least partially) responsible for the observed differentially expressed genes.

Figs. 2 and 3

2.1. Related to the previous question on the amount of time required for the cells to pass through the device, the time point in which the RNAseq was performed is not clear. If the time window is too narrow, the true effects in gene expression may be lost due to the high stability of many RNAs. Also, author should consider again the concern in viability which may bias the overall differential expression analysis. Interestingly, PIEZO1 is linked to a mechano-dependent apoptosis pathway (Hope et al., Cell Death Dis 2019; Song et al, iScience 2022) which the authors should mention.

2.2. Authors performed sphere assays which is a good proxy for clonogenicity in 3D justified by the increase in stemness-like genes. However, I find essential to complement this study with an in vivo tumor formation by intradermal/subQ injection followed by survival surgery, a true model of metastasis. As mentioned above, the number of animals presented in the tail vein model was insufficient (n=4). Power calculations should be conducted and provided, and the corresponding n of mice should be used.

2.3. It is counterintuitive that there are almost no metastases by IVIS in the control conditions since A375-MA2 are highly metastatic. Is it possible that cells lost expression of the luciferase reporter? Quantitation of tumor burden by histological analyses could be more reliable. Indeed, in FIG3g, one can see micro-mets in the lungs H&E control although the bioluminescence shows no signal.

Figure 4

3.1 PIEZO1 comes out of nowhere. Is this gene among the differentially expressed between control and squeezed cells? The choice of this gene should be justified.

3.2 PIEZO1 translocation to the cytoplasmic membrane is not well supported. A PIEZO1 Western blot of subcellular protein fractionations should be provided.

3.3 To further support the specificity of their findings, authors could include a cell line which does not express PIEZO1. If not available, they could try overexpressing PIEZO1 and test if cells become stem-like without compression.

3.4 The observation of PIEZO1 pathway in modulating stem-like genes in a Yoda1-dependent manner is interesting; the use of Yoda1 to evaluate the epistasis makes sense, however ruthenium red (RR) inhibits all Ca²⁺ channels. It'd be more direct to inhibit PIEZO1, by transducing cells siRNA or sgRNA. In addition, the authors should address whether is the actual influx of Ca²⁺ and/or the conformational change of PIEZO1 to modulate the pathway using Ca²⁺ chelators or reproduce the experiment in Ca²⁺-free media.

3.5 There is no western blot nor qPCR showing the actual efficacy of PIEZO1 KO, please provide it. Is this a clone KO or a pool? In Material & Methods for this, the wrong reference (ref #24) seems to be cited. Specify, if it's a single clone and not pooled cells, authors should repeat the experiments in additional 2 clones. Also, is the viability of KO cells affected? To really prove the specificity of PIEZO1 KO effects, two independent sgRNAs or an orthogonal approach (e.g. shRNA) should be used. Alternatively, to prove the specificity of the KO experiments with Yoda1, RR, or Ca²⁺ depletion should be repeated in KO cells (or cells lacking PIEZO1 levels).

3.6 Fig5N-P cell viability of all conditions is missing- 2D proliferation assays should be provided, to demonstrate if the effects are specific for tumor sphere formation or simply affect cell viability. Why is KO-SQZD-RR missing?

Figs 5, 6

4.1. Authors conclude that squeezing increases the ability of cancer cells to disrupt blood vessel integrity to enhance extravasation and metastatic spread. However, the in vivo assay that they use is not conducive to test this hypothesis because it doesn't require extravasation. In the tail injection assay cancer cells are just adhered to the lung microcapillaries and start growing there. A true metastasis assay from a flank tumor or intracardiac injection requires extravasation and would represent a better model to assess effects of compression/squeezing on extravasation. This part of the study should be complemented by proper in vivo metastasis studies.

4.2. Optimally, the authors should evaluate the relevance of their findings to patient tissues. Authors should analyze FFPE of primary invasive melanomas (in vertical growth phase) for PIEZO1, looking for correlations with stem cells markers (CD44, ABCB5, PRDM14, CD271), or histone markers (H3K9me3, H3K9ac), to characterize whether their observations are likely to happen in human samples. Based on their findings, they would expect PIEZO1 cytoplasmic membrane localization in cells close to the epidermal to dermal boundary, and to endothelial/pericyte cells of the vessels. PIEZO1 positive cells should also express more stem-cell markers. Do PIEZO1 levels correlate with patients' outcomes?

Reviewer #3

(Remarks to the Author)

The manuscript by Silvani et al studies the effect of mechanical constriction on the transition of cells into a cancer-like state. The study mainly utilized a microfluidic-based squeezing assay, accompanied with a series of cell/molecular biology experiments. The authors identified Piezo1 as a key molecular factor in this cancer-like transition.

Overall, I found this topic interesting, potentially to a broad readership. The manuscript is, for the most part, well written and easy to follow. Most of the data presented in the manuscript are convincing. However, the reviewer found key experimental conditions that are missing. Additionally, several errors/typos throughout the main text and figure captions significantly reduced the readability of the manuscript.

My main concerns are:

- 1) There is very little information about the time scale of the experiments. Since key conclusions of the paper are based on measurements of protein expression. The durations of applied stimulation/waiting time are critical.
- 2) Figure 1B, the change in DI (deformation index) seems to depend on cell-cell (or potentially cell-substrate) adhesion (cells are narrow constrictions appear to cluster/adhere to each other, making them less spherical). Additionally, the inverse correlation between DI and channel diameter should be clearly plotted.
- 3) Please clearly state the biological meaning of the observed changes in H3K9.
- 4) Figure 2A, what is 'X' and wouldn't the large variation between replicas significantly reduce the strength of the conclusion?
- 5) Figure 2D and 2E appear mis-labelled.. Why are there many more genes presented in 2E than in 2D?
- 6) Consider the large amount of cell death after constriction (Figure 1C), how do the authors control the total number of cells when carrying out experiments in Figure 3? More importantly, would signals released during cell death potentially affect the interpretation of the results. Notably, an inflammatory response was observed in figure 6.
- 7) Figure 3B 3E, error bars are missing. 3F, statistical test missing. 3C, how do the authors know that the increase in area is not due to flattening of the cell cluster?
- 8) Figure 4, the reviewer didn't find data that support the increase in 'Piezo1 translocation to the membrane'. Also, please provide the dynamic traces of Ca²⁺. Please provide scale bars in 4C.
- 9) Caption for figure 5I is missing. What is the meaning of each data point in 4D?
- 10) The effect of Piezo1 on the recovery of DI after constriction is somewhat frustrating. What are the potential mechanisms where a membrane protein could change the deformability of the entire cell?
- 11) Figure 6D, the integrity of the endothelium seems to be damaged in all cases with cancer cells. 6J, what are the meaning of control and vessel?
- 12) The author should comment on whether the identified Piezo1 effects are purely Ca²⁺-based, or may have to do with more unique molecular features of Piezo1.
- 13) The use of Ruthenium Red as a Piezo1 blocker is not well justified. How specific is the blocker? How does it compare to potentially more broadly used blockers such as GxMT4?

Minor suggests:

- 1) It would be helpful to more clearly discuss the novelty of the current findings with respect to previous studies of mechanically induced metastatic transition and of Piezo-mediated cell mechanics.
- 2) The (i) to (iv) label in figure 1A are wrong. And several labels of figure axis are too small to read.
- 3) The concentration of Yoda1 (20 μ M) is somewhat high, have the authors tested the effect of lower Yoda1 concentration?
- 4) Please be more specific on what CAAX motif was used for membrane lable.
- 5) 'u' was used instead of ' μ ' in several units.
- 6) Please be more specific about how total cell number (and dead cells) were determined in viability assay? Did the author consider the possibility of trypan blue staining live cells?
- 7) While it's fine to not provide the matlab codes at the current stage, please specify what principles were used to determine expression level.
- 8) Page 9, line 21. 108 cells per 1 ml seem very low.

Reviewer #4

(Remarks to the Author)

Version 1:

Reviewer comments:

Reviewer #1

(Remarks to the Author)

The authors have done a good job in responding to my prior review comments. Especially the addition of new data to explore the points raised. The Lamin A experiment probably needs some context for the reader, because in the current manuscript it sort of comes out of nowhere.

It's an interesting study and I don't have any major remaining areas of concern.

Reviewer #2

(Remarks to the Author)

The authors have addressed most of the previously raised concerns, and this reviewer appreciates their efforts to better dissect the PIEZO1-dependent compression mechanism in the context of melanoma and additional cancer cell models (triple-negative breast cancer and osteosarcoma). Particularly noteworthy is the addition of the intracardiac metastasis model, which demonstrates not only an increase in metastasis of cancer cells upon compression but also a preferential tropism toward certain organs such as the brain and bone. This aspect could be further pursued in future studies. However, some of my previous comments remain unaddressed, which are crucial for publication in Nature Communications.

Outstanding Points

1. How do PIEZO protein levels or subcellular localization change in response to microcapillary 'squeezing'? The authors performed phosphoproteomics after squeezing and identified significant increases in pERK, NFkB, YAP1/TEAD, PKC, and mTOR. Using inhibitors against these pathways (e.g., Trametinib, Rapamycin) could provide insights into the mechanisms of PIEZO1 increased expression and activity.
2. How does PIEZO1 alteration result in H3K9me3 increase and expression of stem-like genes? A genome-wide characterization of H3K9me3 and H3K27ac deposition would help determine if differences in intensity are accompanied by de novo deposition of histone modifications at specific genes and regulatory elements, which is more compatible with a phenotypic change.
3. Control and squeezed cells should be assessed by H3K9me3 ChIP-seq or CUT&Tag/Run to determine if/how H3K9me genome distribution changes upon microcapillary-induced constriction. These analyses would reveal if the observed changes in histone marks are (at least partially) responsible for the differentially expressed genes observed. The authors have already performed RNA-seq in control and SQZD conditions. Integrating transcriptional changes with ChIP-seq or Cut&Tag/Run for H3K9me3 and H3K27ac would provide a more complete epigenetic characterization of whether there is a switch to repressive or active chromatin at specific loci. These experiments would also allow for DNA binding motif analysis at the differential peaks to uncover potential transcription factors involved in the direct activation of PIEZO1 and the differentially expressed genes (e.g., NFkB, JUN, TEAD, BHLH, known to be downstream of the pathways identified by phosphoproteomics).
4. In vivo/clinical relevance: The authors could inspect cancer cells in proximity to blood vessels or in tumor-dense regions where mechanical compression may elicit a similar activation of PIEZO1 and its targets. An in vivo confirmation would provide further support for the physiological relevance of the reported in vitro effects.
5. Injecting control vs. squeezed cells subcutaneously or intradermally and monitoring tumor growth would be important to assess if the increase in metastasis following compression is truly due to enhanced metastatic capacity and/or proliferation. Faster-growing cells are more likely to cause a higher metastatic burden.

Minor Point

Is the mechanism Ca²⁺-dependent or PIEZO1 conformation-dependent? The authors performed part of the requested experiments; however, it is unclear why they did not include the complete media experiment in parallel to have all conditions side by side. Please clarify.

Reviewer #3

(Remarks to the Author)

The authors have satisfactorily addressed my concerns. Just a final suggestion, I noticed the added Method section is written quite carelessly. For example on page 5: what is ISM?, ul?, ship? Also, why does a "viscosity model" give Young's modulus?

Reviewer #4

(Remarks to the Author)

Reviewer's Comments:

Reviewer #1 (Remarks to the Author):

This manuscript addresses the effect that compressive squeezing forces (such as experienced in capillary transport) have on tumorigenicity and other functions of melanoma cells. There are some interesting observations and implications presented here. The authors address the role of Piezo1 using knockout cells and also using the chemical agonist Yoda1. My specific comments are given below:

1. I am very interested to know how the time between mechanical stimulus and analysis will affect the results, as this has been shown to be a dynamic process in previous studies where even a relatively short mechanical deformation stimulus can continue to have a varying effect on cells for hours or even days afterwards.

We thank the reviewer for highlighting the dynamic nature of cellular responses to mechanical stimuli. Although the mechanical deformation experienced by cells during microcapillary transit lasts milliseconds, due to the physiological flow rates we employed to simulate *in vivo* capillary dynamics, we found that this brief stimulus is sufficient to induce lasting transcriptional and phenotypic changes. As a follow up to the reviewer's point, we analysed the expression of stemness markers CD44 and ABCB5 at multiple time points after squeezing. Both markers exhibited a significant increase compared to control, with a peak at 24 hours, followed by a decline by 72 hours (New Fig. S6). We also investigated phosphosignalling pathways post constriction which confirms some changes in key activities downstream of Ca²⁺/PIEZO1 and regulation of "stemness" signalling (New Fig. S9). These findings indicates that even a short-lived mechanical input can have a transient reprogramming window, during which the mechanically stressed cells adopt a stem-like phenotype before gradually returning toward baseline.

We have added new figure in supplementary material (Fig. S6) and clarified these time points and findings in the revised manuscript (p.21, lines 15-22):

"To clarify the timing of these phenotypic changes, we assessed marker expression at multiple time points ranging from 15 minutes to 72 hours post-constriction. Differences in protein expression for stemness-associated markers like CD44 and ABCB5 began to emerge around 3–4 hours, peaked at 24 hours, and diminished by 72 hours (Fig. S6). Although the passage through

the microfluidic constriction occurs within milliseconds under physiologically relevant flow conditions, the resulting changes in protein expression emerged over a longer timescale, indicating that the initial mechanical stimulus triggers sustained downstream signalling activity.”

Fig. S6 Temporal dynamics of phenotypic marker expression following transient mechanical constriction. (A) Bar graph displaying the CD44 levels in CTRL and SQZD cells at multiple time points (15 min to 3 hrs). (B) Bar graphs displaying the CD44 levels in CTRL and SQZD cells at multiple time points (4 hrs to 72 hrs). (C) Bar graph displaying the ABCB5 levels in CTRL and SQZD cells at multiple time points (15 min to 3 hrs). (D) Bar graphs displaying the ABCB5 levels in CTRL and SQZD cells at multiple time points (4 hrs to 72 hrs).

Fig. S9. Phosphoprotein imaging reveals time-resolved activation of mechanosensitive signalling pathways. (A) Heatmap representing log₁₀ fold change in the activation of selected phospho-markers (NF-κB, PKC, YAP1, mTOR, and p-ERK) at 7-, 15-, and 50-minutes post-constriction. Line plot showing quantification of N/C ratios for YAP1 (B), p-ERK (C), and NF-κB (D) over time. Box plot representing mTOR nuclear intensity (E) and PKC cytoplasmic intensity (F) over time. Error bars indicate the Standard Error of the Mean (SEM).

2. Related to that point, The authors should measure the mechanical properties of the cells before and after squeezing stimulus, as it has been previously shown that cancer cells change their stiffness and viscoelasticity in response to mechanical stress.

As requested, we have conducted experiments using real-time deformability cytometry (AcCellerator, Zellemechanik) to measure both cell deformation and apparent Young's modulus in wild-type (WT) and PIEZO1 knockout (KO) melanoma cells. This data revealed that WT cells exhibit a significant change in mechanical properties following passage through the microfluidic constriction device, specifically an increase in stiffness, indicating an adaptive response to mechanical deformation. In contrast, PIEZO1 KO cells did not exhibit a comparable change in mechanical properties, suggesting that PIEZO1 is required for this mechano-adaptive response.

This observation aligns with previous studies in other cancer models, such as MDA-MB-231 breast cancer cells, where mechanical compression activates PIEZO1, leading to cytoskeletal reinforcement and increased invasiveness ¹. Our findings suggest that melanoma cells similarly engage a PIEZO1-dependent program to structurally reinforce themselves after constriction, enhancing their resilience in confined microenvironments.

While questions remain with respect to the mechanism of this apparent change in cell mechanics, the results strengthen the central premise of our study by demonstrating that PIEZO1 mediates an active remodelling of cell mechanics in response to external mechanical stress. The relevant data have now been included in the revised manuscript (new Fig. S12), Methods section (Real-time deformation cytometry (RTDC)), and Result section (p.25, lines 22-28).

“To further investigate this difference in mechanical response, we performed real-time deformability cytometry to quantify cell mechanical properties before and after squeezing. WT melanoma cells showed a significant increase in stiffness following constriction, consistent with an adaptive mechanical response. In contrast, KO cells did not exhibit any appreciable change in stiffness or viscoelastic properties, suggesting that PIEZO1 is required for this mechano-adaptive behaviour (Fig. S12)”

And discussion section (p.34, lines 10-19):

“This mechanotransductive role of PIEZO1 aligns with previous findings in breast cancer cells, where mechanical compression activates PIEZO1 and triggers cytoskeletal reinforcement, ultimately enhancing invasive behaviour ¹. Similarly, our data indicate that melanoma cells respond to confinement by increasing stiffness in a PIEZO1-dependent manner, pointing to a conserved adaptive mechanism. As cells migrate through narrow constrictions, increased tension in the lipid bilayer and actomyosin cortex likely serves as a trigger for PIEZO1 activation ²⁻⁴. This activation initiates calcium-dependent pathways that remodel cellular mechanics, enabling survival and migration under physical stress. Together, these findings suggest that PIEZO1 acts not only as a sensor but also as a regulator of adaptive responses in mechanically challenging environments.”

Fig. S12 PIEZO1 is required for post-constriction mechanical adaptation in melanoma cells. (A–B) Contour density plots showing the deformation (A) and apparent Young's modulus (B) of wild-type (WT) melanoma cells before (WT_CTRL, blue) and after (WT_SQZD, red) passing through a microfluidic constriction. (C–D) The same analysis for PIEZO1 knockout (KO) melanoma cells, comparing control (KO_CTRL, light blue) and constricted (KO_SQZD, orange) populations. Each contour plot represents the distribution of single-cell measurements ($n > 1,000$ cells per condition), with contour lines corresponding to 50% and 90% quantile density levels.

3. The authors should discuss recent studies showing that fluid shear stress activation of Piezo1 in various types of cancer cells, designed to be a model of circulating cancer cells such as (presumably) in this current manuscript and its context of capillary transport, of how Piezo1 activation is also known to sensitize cancer cells to undergo apoptosis via TRAIL exposure. Since squeezed cancer cells will naturally encounter TRAIL on the surface of immune cells such as natural killer cells, macrophages and dendritic cells, might an increasing propensity to undergo apoptosis counteract the enhanced tumorigenicity shown here?

We thank the reviewer for this important point and for highlighting the relevance of recent studies showing that PIEZO1 activation by fluid shear stress can sensitize cancer cells to TRAIL-induced apoptosis in the context of circulation. Indeed, we acknowledge that circulating tumour cells are exposed to both mechanical forces and immune surveillance, including interactions with TRAIL-expressing immune cells such as natural killer cells, macrophages, and dendritic cells.

In our study, we observed reduced viability in a subset of squeezed melanoma cells (Fig. 1C), suggesting that mechanical deformation may indeed increase apoptotic sensitivity. While this aligns with literature showing PIEZO1-dependent apoptosis under shear stress and TRAIL exposure, our study was primarily focused on the adaptive, tumorigenic phenotype that emerges from the surviving cell population. Therefore, while we did not directly assess apoptotic sensitivity or immune cell interactions in our model, we agree that PIEZO1-mediated responses may represent a double-edged sword, promoting stemness and metastasis in some contexts while potentially sensitizing cells to immune clearance in others.

We now mention this in the revised Discussion (p. 36, lines 22-32):

“While we primarily focused on the adaptive, tumorigenic phenotype that emerges from cells surviving mechanical deformation, it is important to recognize that PIEZO1 activation may also sensitize circulating tumour cells to immune-mediated apoptosis, as highlighted in recent studies involving TRAIL exposure under fluid shear stress conditions ^{5, 6}. Indeed, in our system, we observed a measurable loss in cell viability immediately after constriction (Fig. 1C), suggesting that mechanical stress may trigger cell death in a subset of cells. However, the longterm transcriptional and phenotypic reprogramming observed in the surviving population underscores the capacity of certain cells to resist apoptosis and adapt, acquiring stem-like and pro-invasive features. Thus, PIEZO1 may represent a double-edged sword in the metastatic cascade, promoting adaptation and metastasis in some cells, while increasing apoptotic vulnerability in others when exposed to immune effectors such as TRAIL-expressing NK cells, macrophages, or dendritic cells.”

4. Does the capillary squeezing also phosphorylate NF-κB and Zap70 in a Piezo1-dependent manner as found in other mechanical activation of circulating cancer cells?

We thank the reviewer for this insightful comment. To better understand the immediate signal transduction events following capillary-like constriction, we performed phosphoproteomic profiling of melanoma cells post deformation using a multiplexed immunolabeling approach that allows simultaneous assessment of multiple phospho-proteins at the single-cell level analysis ^{7, 8}. The results revealed activation of several key signalling pathways downstream of PIEZO1-mediated calcium influx, including nuclear translocation of NF-κB, YAP and p-Erk, nuclear activation of mTOR and membrane recruitment of PKC. These pathways are well-established downstream effectors of intracellular calcium signalling and have recognized roles in regulating chromatin state and transcriptional programs. In particular, NF-κB and YAP have both been implicated in the transcriptional activation of plasticity- and stemness-associated genes^{9, 10}, while PKC and mTOR can regulate epigenetic modifiers, including histone methyltransferases such as those affecting H3K9me3 levels^{11, 12}. Together, these findings support our model in which capillary-induced mechanical deformation triggers a coordinated calcium-dependent signalling network that promotes chromatin remodelling and gene expression changes contributing to tumour plasticity. The relevant data have now been included in the revised manuscript (see new Fig. S9) and Result section (p.24, lines 10-14).

“To evaluate signal transduction downstream of PIEZO1, we performed phosphoproteomic profiling at early time points post-constriction using multiplexed immunolabeling ^{7, 8}. This analysis revealed elevated nuclear localisation of phosphorylated mechanotransduction elements downstream of PIEZO1 including NF-κB, PKC, YAP1, mTOR, and ERK (**Fig. S9**).”

and Discussion section (p.34, lines 20-32).

“In line with PIEZO1-mediated calcium influx, we observed distinct activation patterns across several mechanosensitive signalling pathways following microcapillary constriction (**Fig. S9**). Specifically, NF-κB and YAP exhibited increased nuclear translocation, consistent with their established roles in transcriptional activation of plasticity and survival-associated gene programs¹³⁻¹⁷. Nuclear enrichment of mTOR was also observed, supporting its involvement in epigenetic regulation following mechanical stress ^{18, 19}. In contrast, pERK showed a brief increase in nuclear localization, which then decreased, suggesting a rapid transcriptional response that is tightly controlled to prevent prolonged ERK activation, which could otherwise disrupt cellular homeostasis ²⁰. Finally, PKC activity exhibited a rapid cytoplasmic spike immediately after constriction, likely driven by calcium influx, followed by downregulation over time as homeostasis was re-established. Collectively, these early phosphorylation events downstream of PIEZO1 likely converge to promote transcriptional plasticity and support the emergence of a melanoma stem cell-like state”

5. It would be interesting to see the outcome of introducing the squeezed melanoma cells in a cardiac injection model of metastasis rather than tail vein, to see if this model recreates any of the organ-specific metastasis sites of melanoma. If it doesn't, what would that imply for the proposed importance of capillary squeezing?

We appreciate this comment regarding organ-specific metastasis and the relevance of capillary squeezing *in vivo*. In response, we performed new intracardiac injection experiments using A375-luciferase melanoma cells to better model systemic dissemination and metastatic colonization.

These experiments, conducted across two independent cohorts (n = 8 for control, n = 10 for squeezed cells), demonstrated that microcapillary-like constriction significantly increases metastatic burden and reduces survival. Specifically, Kaplan–Meier analysis revealed decreased survival in the squeezed group beginning at week 4 (New Figure 3M). Whole-body bioluminescence imaging showed increased total metastatic signal in the squeezed group (New Fig. 3G). Importantly, organ-specific analysis revealed increased colonization in the lungs, bone, and brain, which are key target organs in clinical melanoma metastasis (New Fig. 3H, L).

These findings strengthen the biological relevance of our model and support the hypothesis that mechanical deformation through capillary-like constrictions prime tumour cells for enhanced metastatic potential, including organotypic dissemination. The new data have been included in the revised manuscript and are now discussed in detail in the Results sections (p.22, lines 1223):

“Building upon this preliminary finding, we next expanded and validated our observations using a more sensitive and systemic intracardiac injection model to assess global metastatic dissemination and organotropic colonisation. Following injection of 1×10^5 A375-Luc2 melanoma cells into the left ventricle, mice receiving squeezed cells exhibited a striking increase in total metastatic burden and distribution. Longitudinal bioluminescence imaging demonstrated elevated whole-body photon flux in the SQZD group by week 4 (**Fig. 3G**), including increased median signals in the lung (**Fig. 3H**) and significantly higher signals in bone and brain (**Fig. 3I, L**), indicating elevated organ-specific homing. Importantly, survival analysis indicated that animals injected with squeezed cells had reduced overall survival, with multiple animals reaching ethical endpoints before day 50 (**Fig. 3M**). In fact, 9/10 animals in the SQZD group reached ethical endpoint by day 52, compared to 1/8 animals in the CTRL group. Collectively, these results demonstrate that micro constriction enhances the *in vivo* metastatic potency and lethality of melanoma cells in a systemic dissemination model.”

Fig. 3 Microcapillary-like constrictions enhance melanoma cell tumorigenicity. (A) Images of control (CTRL) and squeezed (SQZD) melanoma cells forming tumorsphere in serum-free media after 7 days of culture. (B) Graph representative of the mean tumorsphere area increasing over time when cultured in serum-free media, for CTRL (blue) and SQZD (red) cells. (C) Bar graph displaying the area of tumorsphere for CTRL and SQZD cells at day 7. The result is expressed as the mean from four independent experiments. (D) IVIS Imaging of lung metastases that developed after 30 days from injecting CTRL and SQZD luciferase melanoma cells into the lateral tail vein of 7-week-old female Balb/c Nude mice. [Note the change in scale between the groups]. (E, F) Representative H&E-stained lung sections from mice injected with CTRL cells (E) and SQZD cells (F) harvested 30 days post tail vein injection. (G) Whole body luciferase signal of mice injected intra-cardially at week 4, the final timepoint at which all mice in both groups were accounted for. (H-L) Photon flux from individual metastatic sites revealed increased median signals in the (H) lung and significantly higher signals in (I) bone and (L) brain in the SQZ group compared to controls. Each dot represents an individual animal. (M) Metastatic burden and Kaplan–Meier survival curves following IC injection with control (black) and squeezed (red) cells. Data represent pooled results from each group (CTR, n = 8; SQZ, n = 10).

6. I did not see much discussion of a major limitation of the *in vitro* capillary squeezing system, namely that the microfluidic chip is not deformable like an *in vivo* capillary. The implication is that this study might overestimate the importance of this type of stimulus in the overall cancer development.

We agree that a major limitation of our *in vitro* capillary squeezing system is that the microfluidic chip is not deformable like *in vivo* capillaries. This discrepancy may lead to an overestimation of the role of capillary constriction in cancer progression, since the compliant nature of real capillaries is not fully replicated. Nevertheless, these findings provide a guide to further work where the mechanics of channels may be modified to evaluate signalling response. We have added the following sentence to the manuscript to clarify this point (p.32, lines 3-7):

“Although microfluidic technology does not fully replicate the mechanical properties of *in vivo* capillaries, particularly their deformability, which may lead to an overestimation of the role of capillary constriction in cancer progression, it has nonetheless proven invaluable for modelling flow in confined spaces, such as microcapillary beds, offering broad design versatility and precise control over dimensions and dynamic experimental conditions 21-26”

We believe that this addition clarifies the limitations of our model and provides important context for interpreting our findings.

7. Has anyone ever documented calcium influx in response to cancer cells caught or passing through mouse capillaries, via intravital microscopy? It seems that if this was a predominant, first-order effect, that it would have been found by others examining the passage of CTCs.

To the best of our knowledge, no study has specifically documented calcium influx in response to cancer cells caught or passing through mouse capillaries using intravital microscopy. While there has been significant research on circulating tumour cells (CTCs) and their interactions with the vasculature, the real-time measurement of calcium signalling in CTCs as they pass through capillaries *in vivo* remains largely unexplored.

Our study offers a new perspective on the role of calcium signalling in CTC behaviour within the vasculature, which could reveal new insights into the metastatic process and colonisation mechanisms. We are currently in discussions with collaborators to explore the analysis of cells through an intravital imaging window. However, this method will take considerable time to develop and is better suited as a follow up study.

8. Might have missed this, but if cells can be directly observed in the squeezing chip, it should be possible to observe cells "mid-squeeze" instead of just analysing the whole population afterwards. In this manner, the degree of cell polarization (more calcium etc. in the leading edge vs. the trailing edge, or vice versa). It would be quite interesting to determine to what degree such polarization occurs, and how important polarization might be for the downstream processes. Also, interesting would be to try and modulate polarization by changing the direction of flow in the chip...

We fully agree that investigating potential polarization phenomena, such as asymmetric calcium distribution across the leading and trailing edges during constriction, would provide valuable mechanistic insights.

In our current experiments, we performed real-time calcium imaging during squeezing at millisecond resolution, focusing on the short time window while cells were passing through the constriction. We analysed calcium influx by normalizing the peak calcium signal to the baseline level recorded just before the cell entered the constriction chamber. However, the fast flow rates used to replicate physiological mechanical forces (260 $\mu\text{L}/\text{min}$) limit our spatial resolution. Although we could track calcium dynamics over time, the current imaging setup does not allow us to resolve detailed spatial differences within the cell while it is being squeezed. Slowing the flow to achieve better spatial resolution would compromise the mechanical conditions that are key to inducing the deformation we study.

We agree that exploring polarization effects with higher spatial resolution or with flow-modulation strategies would be an excellent direction for future work, and we thank the reviewer for highlighting this possibility.

9. The quantification of nuclear deformation is interesting... others have shown that nuclear structural proteins such as Lamin A/B are important for conferring mechanical survival to circulating breast cancer cells when subjected to pulses of high shear stress.

We appreciate the reviewer's comment regarding the role of nuclear structural proteins such as Lamin A/B in conferring mechanical survival to cancer cells. In our study, we examined Lamin A

as part of our analysis and did not observe any significant alterations in its expression or distribution within the 4-hour post-injection period. These results suggest that, under our experimental conditions, the nuclear architecture appears to remain largely intact despite significant deformation during passage through the microfluidic channel, which aligns with the concept of mechanical survival during circulation.

The new data have been included in the revised manuscript (New Fig. S1A) and are now discussed in the Results sections (p. 17, lines 22-24):

“Immunolabelling of the nuclear membrane protein Lamin A demonstrated no significant change in squeezed cells compared to control cells (Fig. S1A)”

Fig. S1 No significant differences in nuclear structure and DNA damage were observed between control and squeezed conditions. (A) Representative staining of nuclear protein Lamin A in CTRL and SQZD melanoma cells. **(B)** Representative staining of γ H2A.X in CTRL and SQZD melanoma cells. **(C)** Bar graph displaying the γ H2A.x levels in CTRL and SQZD melanoma cells.

10. The effect of squeezing force on endothelial cell interactions is also interesting. To what degree is this dependent on E-selectin? Others have shown a distinct E-selectin dependence of melanoma adhesion to stimulated endothelium. Speaking of, I don't see that the endothelium used in these studies have been pre-stimulated into an inflammatory state (believed to be important for blood borne metastasis) via TNF-alpha or other signal.

We thank the reviewer for the valuable comment. The role of E-selectin in melanoma adhesion is well established, particularly in mediating the initial rolling and tethering of cancer cells to the endothelium. However, these early adhesion events are generally insufficient to drive full extravasation. Our study focused on the later stages of this process, where firm adhesion and integrin-mediated signalling, such as through VCAM-1, play a more decisive role in promoting extravasation and pro-invasive behaviour.

Regarding the inflammatory state of the endothelium, we chose not to pre-stimulate the endothelial monolayer with exogenous cytokines like TNF- α . Instead, we aimed to model a scenario where tumour cells themselves initiate endothelial activation. Consistent with this, we observed increased VCAM-1 expression and remodelling of VE-cadherin junctions specifically in response to squeezed melanoma cells. These findings suggest that the mechanical and molecular properties of the tumour cells are sufficient to induce an inflammatory-like response, supporting a more physiologically relevant interaction model.

We have added clarification on this point in the revised Discussion (p. 35, lines 27-33 and p. 36, lines 1-13):

“While E-selectin primarily mediates initial tethering and rolling of circulating melanoma cells on the endothelium, especially under inflammatory conditions ²⁷, later stages of extravasation depend on firm adhesion and integrin-dependent signalling involving molecules such as VCAM-1 ²⁸. Adhesion to the endothelium through CD44 has been shown to involve VCAM-1, a transmembrane glycoprotein implicated in cancer progression and metastatic dissemination in several malignancies, including breast cancer ^{29, 30}. Based on the enhanced metastatic potential observed in our experimental metastasis model, we hypothesized that mechanical constriction might promote tumour-endothelial interactions by enhancing this adhesion pathway. Using custom microfluidic assays, we observed that squeezed melanoma cells exhibited increased adhesion to endothelial monolayers. This was accompanied by upregulation of VCAM-1 expression and disruption of junctional VE-cadherin in endothelial cells, indicating endothelial activation and loss of barrier integrity. Deletion of PIEZO1 abolished these effects, supporting a role for PIEZO1-dependent mechanotransduction in driving both the tumour cell pro-adhesive phenotype and the endothelial response. Importantly, to specifically assess tumour-intrinsic mechanisms, we deliberately avoided pre-stimulation of endothelial cells with TNF- α or other cytokines, which are often used to mimic inflammatory conditions in extravasation models. Remarkably, squeezed melanoma cells alone were sufficient to initiate this endothelial activation, even in the absence of exogenous inflammatory cues. This finding suggests a tumour-intrinsic capacity to modulate the endothelial environment post-constriction, enhancing metastatic potential in a physiologically relevant manner.”

11. The stem cell marker emergence is notable and is consistent with research showing that stem cell markers emerge in breast cancer cells in conjunction with up regulation of cell-cell adhesion proteins and glycoproteins such as E-selectin ligand-1 and E-cadherin.

We thank the reviewer for highlighting this important link in breast cancer. While previous studies have implicated adhesion proteins such as E-selectin ligand-1 and E-cadherin in promoting stem-like phenotypes, our focus was specifically on VCAM-1 signalling, which appears particularly relevant in the melanoma context^{28, 31, 32}. Although we did not directly assess a mechanistic link between CD44 and VCAM, our data suggest a potential association in the context of inflammatory extravasation. This aligns with existing literature implicating CD44 in adhesion and migration processes^{33, 34}. Together, our findings in conjunction with previous studies support a broader model in which mechanical stress promotes stem-like, pro-metastatic traits via modulation of cell adhesion pathways. We appreciate this valuable perspective and note it as a direction for further investigation

Reviewer #3 (Remarks to the Author)

The manuscript by Silvani et al studies the effect of mechanical constriction on the transition of cells into a cancer-like state. The study mainly utilized a microfluidic-based squeezing assay, accompanied with a series of cell/molecular biology experiments. The authors identified Piezo1 as a key molecular factor in this cancer-like transition. Overall, I found this topic interesting, potentially to a broad readership. The manuscript is, for the most part, well written and easy to follow. Most of the data presented in the manuscript are convincing. However, the reviewer found key experimental conditions that are missing. Additionally, several errors/typos throughout the main text and figure captions significantly reduced the readability of the manuscript.

We thank the reviewer for the overall positive assessment of our work.

My main concerns are:

1) There is very little information about the time scale of the experiments. Since key conclusions of the paper are based on measurements of protein expression. The durations of applied stimulation/waiting time are critical.

We thank the reviewer for raising this important point. We agree that the time scale of stimulation and downstream analysis is critical for interpreting protein expression changes. In our setup, mechanical deformation occurs as cells transit the capillary-like constrictions under physiological flow conditions, with transit times ranging from milliseconds up to a few seconds depending on cell size and deformability. To better understand the time scales associated with our findings, for the revised manuscript we analysed cells at different time points post-squeezing, ranging from early to later time points (15 minutes - 72 hours), to capture both immediate and sustained effects of mechanical stress. Notably, we observed modest transcript changes at early timepoints, followed by activation of key phosphosignalling cascades within one hour, followed by stabilisation of a stemness phenotype (e.g. CD44 and ABCB5) 3–4 hours post-deformation, with peak expression at 24 hours and a decline by 72 hours. This temporal pattern suggests a delayed but robust phenotypic switch following transient mechanical input.

We have added new figure in supplementary material (New Fig. S6) and clarified these time points and findings in the revised manuscript (p.21, lines 15-22):

“To clarify the timing of these phenotypic changes, we assessed marker expression at multiple time points ranging from 15 minutes to 72 hours post-constriction. Differences in protein expression for stemness-associated markers like CD44 and ABCB5 began to emerge around 3–4 hours, peaked at 24 hours, and diminished by 72 hours (**Fig. S6**). Although the passage through the microfluidic constriction occurs within milliseconds under physiologically relevant flow conditions, the resulting changes in protein expression emerged over a longer timescale, indicating that the initial mechanical stimulus triggers sustained downstream signalling activity.”

Fig. S6 Temporal dynamics of phenotypic marker expression following transient mechanical constriction. (A) Bar graph displaying the CD44 levels in CTRL and SQZD cells at multiple time points (15 min to 3 hrs). **(B)** Bar graphs displaying the CD44 levels in CTRL and SQZD cells at multiple time points (4 hrs to 72 hrs). **(C)** Bar graph displaying the ABCB5 levels in CTRL and SQZD cells at multiple time points (15 min to 3 hrs). **(D)** Bar graphs displaying the ABCB5 levels in CTRL and SQZD cells at multiple time points (4 hrs to 72 hrs).

2) Figure 1B, the change in DI (deformation index) seems to depend on cell-cell (or potentially cell-substrate) adhesion (cells are narrow constrictions appear to cluster/adhere to each other, making them less spherical). Additionally, the inverse correlation between DI and channel diameter should be clearly plotted.

We thank the reviewer for this observation. While the image in Figure 1B may suggest potential cell-substrate interactions, we would like to clarify that the microfluidic device was pre-treated with 1% BSA to minimize cell-substrate adhesion which was verified during imaging. Furthermore, given that the cell density and injection flow rate were consistent across all channel widths, any substantial influence of cell-cell adhesion would be expected to affect the deformation index (DI) similarly across both narrow and wider channels.

The consistent inverse correlation observed between DI and channel diameter supports our interpretation that the changes are primarily driven by mechanical deformation within constrictions. As requested, we have now included a plot of DI versus channel diameter in the revised figure (**Fig. 1E**) and expanded on these points in the Results section.

Figure 1E. Inverse relationship between microchannel diameter (black, left Y-axis) and corresponding median deformation (% Median Deformation, red, right Y-axis) for melanoma cells transiting sequential microfluidic constrictions.

3) Please clearly state the biological meaning of the observed changes in H3K9.

We appreciate the reviewer's request for clarification. The observed changes in H3K9 modifications reflect a rapid chromatin remodelling in response to mechanical deformation. Specifically, we found that squeezing induced a significant decrease in H3K9 acetylation (H3K9ac) and a concurrent increase in H3K9 trimethylation (H3K9me3). These histone marks are well-established indicators of chromatin state: H3K9ac is associated with open, transcriptionally active euchromatin, while H3K9me3 marks condensed, transcriptionally repressed heterochromatin. The shift from H3K9ac to H3K9me3 following constriction suggests that mechanical stress induces chromatin condensation and transcriptional silencing in specific genomic regions. This epigenetic reprogramming may serve as a rapid adaptive mechanism, priming the cells for downstream phenotypic changes such as stemness acquisition or enhanced survival. We have included additional figure (New Fig. S3) and text in the revised results to clarify our interpretations (p19, lines 7-13):

“These histone modifications occurred rapidly, within 15 minutes of mechanical deformation, and reflect a shift from transcriptionally active to repressive chromatin states. To assess the stability of these changes, we performed additional time-course experiments, which revealed that H3K9ac reduction was transient and largely restored by 24 hours, while H3K9me3 remained stably elevated (**Fig. S3**). This suggests that mechanical stress induces chromatin condensation and transcriptional silencing in specific genomic regions, potentially priming cells for downstream transcriptional reprogramming and adaptive phenotypic changes”

Fig. S3. Temporal dynamics of histone modifications following mechanical deformation. Representative immunofluorescence images of H3K9ac (A) and H3K9me3 (C) staining at different times post-constriction in control (CTRL) and squeezed (SQZD) A375-MA2 cells. Bar graphs displaying the quantification of ABCB5 H3K9ac (B) and H3K9me3 (D) mean fluorescence intensity levels in CTRL and SQZD cells at multiple time points (4 hrs to 72 hrs).

4) Figure 2A, what is 'X' and wouldn't the large variation between replicates significantly reduce the strength of the conclusion?

We thank the reviewer for the observation. In Figure 2A, 'X' is used as a generic label to denote individual biological replicates across both CTRL and SQZD groups. As noted, one replicate within the SQZD group shows partial separation from the others. However, despite this variation, all SQZD samples, including this one, exhibit statistically significant differential gene expression compared to the CTRL group, as shown in Figure 2C. We interpret this variability as reflective of the inherent heterogeneity in biological responses to mechanical stress. In fact, the persistence of key differentially expressed genes across replicates, even in the presence of some variance, reinforces the robustness of the observed transcriptional shift.

We have clarified this point in the revised figure legend and Results section (p 19, Lines 24-26):

“The distribution of squeezed replicates is broader than control which we attribute to heterogeneity in biological response. Nevertheless, all squeezed samples exhibit statistically significant differential gene expression compared to the control group.”

5) Figure 2D and 2E appear mis-labelled. Why are there many more genes presented in 2E than in 2D?

We thank the reviewer for pointing this out; there was a mislabelling in the figure legend. Figure 2D is the heatmap of the differentially expressed genes, while Figure 2E is a bar graph summarizing the number of genes associated with selected GO biological processes and KEGG pathways, including metabolic reprogramming (orange), tumorigenicity and metastasis (red), cell-matrix interactions (green), and cell cycle (blue). To better display subtle changes in transcript expression, in the revised manuscript we broadened our threshold from $\log_2FC > 1.2$ to $\log_2FC > 0.6$ and presented the differentially expressed genes as both a volcano plot (Fig. 2C) and heatmap (Fig. 2D). We have also included more description on the analysis performed which we hope improves the results presentation.

6) Consider the large amount of cell death after constriction (Figure 1C), how do the authors control the total number of cells when carrying out experiments in Figure 3? More importantly, would signals released during cell death potentially affect the interpretation of the results. Notably, an inflammatory response was observed in figure 6.

We thank the reviewer for raising this important point. In all downstream experiments (e.g., tumorsphere assays, animal studies, and co-culture systems), we took care to isolate viable cells after microfluidic squeezing. Immediately following collection, cells were centrifuged at low speed to selectively pellet intact, viable cells while leaving dead or lysed cells in suspension. This method has been validated in our lab and by others as an effective approach to enrich for live cells. Viability was further confirmed by observing that, after 4 hours in culture, all cells adhered to the substrate, with no significant population of floating or dying cells remaining. Regarding the potential influence of signals released from dying cells, we acknowledge this as a relevant consideration. However, in experiments such as the endothelial co-culture (Figure 6), the observed inflammatory response, including VCAM-1 upregulation and VE-cadherin remodelling, appeared to be driven by direct interaction between live squeezed melanoma cells and the endothelium. Since non-viable cells were removed before seeding, we interpret the endothelial activation as a response to mechanobiological altered, viable tumour cells rather than to byproducts of cell death.

We have clarified this point in the revised result section (p. 27, lines 23-25):

“Once maturation was confirmed, melanoma cell suspensions (WT and KO, squeezed and untreated) were centrifuged and only the viable cells were introduced into the PDMS platform and co-cultured overnight under mild flow conditions.”

7) Figure 3B 3E, error bars are missing. 3F, statistical test missing. 3C, how do the authors know that the increase in area is not due to flattening of the cell cluster?

We thank the reviewer for these observations. In response to feedback of other reviewers and to better assess the metastatic potential of our experimental model, we also have performed an intracardiac injection model, which allows for a more comprehensive assessment of systemic dissemination. The new figure (updated Fig. 3) includes error bars for all quantitative data and are indicated directly in the revised figure legend.

Regarding the reviewer's question on Figure 3C: the tumorsphere assay is conducted in ultra-low attachment plates using serum-free media, conditions that promote 3D spheroid formation rather than adherent, flattened growth. As described in the Methods section, quantification of tumorsphere area is based on brightfield imaging and image analysis using ImageJ. While some variation in spheroid morphology can occur, all conditions were treated equally, and imaging was performed under identical settings. Therefore, any observed increase in area reflects consistent differences between control and squeezed groups. Even in the unlikely case of partial flattening, the relative increase in size remains a robust and comparable metric across conditions.

Fig. 3 Microcapillary-like constrictions enhance melanoma cell tumorigenicity. (A) Images of control (CTRL) and squeezed (SQZD) melanoma cells forming tumorsphere in serum-free media after 7 days of culture. (B) Graph representative of the mean tumorsphere area increasing over time when cultured in serum-free media, for CTRL (blue) and SQZD (red) cells. (C) Bar graph displaying the area of tumorsphere for CTRL and SQZD cells at day 7. The result is expressed as the mean from four independent experiments. (D) IVIS Imaging of lung metastases that developed after 30 days from injecting CTRL and SQZD luciferase melanoma cells into the lateral tail vein of 7-week-old female Balb/c Nude mice. [Note the change in scale between the groups]. (E, F) Representative H&E-stained lung sections from mice injected with CTRL cells (E) and SQZD cells (F) harvested 30 days post tail vein injection. (G) Whole body luciferase signal of mice injected intra-cardially at week 4, the final timepoint at which all mice in both groups were accounted for. (H-L) Photon flux from individual metastatic sites revealed increased median signals in the (H) lung and significantly higher signals in (I) bone and (L) brain in the SQZ group compared to controls. Each dot represents an individual animal. (M) Metastatic burden and Kaplan–Meier survival curves following IC injection with control (black) and squeezed (red) cells. Data represent pooled results from each group (CTR, n = 8; SQZ, n = 10).

8) Figure 4, the reviewer didn't find data that support the increase in 'Piezo1 translocation to the membrane'. Also, please provide the dynamic traces of Ca²⁺. Please provide scale bars in 4C.

We thank the reviewer for this helpful feedback. We agree that the term “translocation” may be misleading in this context. Our intent was to describe the observed increase in PIEZO1 signal intensity at the cell plasma membrane, as shown in Figure 4A. We have revised the text to clarify that this refers to enhanced PIEZO1 localization at the membrane following squeezing, rather than active translocation per se (p.24, lines 1-6):

“To test this, we investigated PIEZO1 expression and localization at the plasma membrane using immunofluorescence, noting an increase in PIEZO1 signal intensity in squeezed melanoma cells when compared to un-squeezed controls (Fig. 4A, B). This suggests that mechanical deformation promotes membrane reorganization and clustering of PIEZO1, positioning the channel for enhanced mechanosensitive activation under capillary-like stress and facilitating downstream calcium influx.”

Regarding the request for dynamic calcium traces (Fig. 4C), we would like to clarify that due to the fast transit of cells through the microcapillary constrictions under physiological flow conditions, which occurs within milliseconds to a few seconds depending on cell deformability, continuous calcium imaging during deformation was technically unfeasible. Instead, we selected snapshot images of cells located within the constriction and quantified their intracellular calcium levels at that timepoint. These values were normalized to each cell's baseline fluorescence, measured just prior to entering the constriction in the upstream relaxation chamber. This approach allowed us to assess deformation-associated calcium influx while preserving physiologically relevant flow and mechanical conditions. Slowing the flow to permit longer recordings would compromise the mechanical stress applied, thereby altering the biological response we aimed to study.

Finally, as requested, we have added scale bars to Figure 4C.

9) Caption for figure 5I is missing. What is the meaning of each data point in 4D? The legend

for Figure 5I is present and reads:

“(I) Representative fluorescence images of WT and KO melanoma cancer cells, seeded on glass 4 hours after being squeezed into the microfluidic device, for biomarkers related to stemness characteristics, e.g., CD44, CD271, ABCB5, PRDM14.”

Regarding the question about Figure 4D, we kindly note that the comment may be referring to a different panel or figure, as Figure 4D consists of representative fluorescence images and does not contain individual data points. If the reviewer intended to refer to another figure, we would be happy to provide further clarification.

10) The effect of Piezo1 on the recovery of DI after constriction is somewhat frustrating. What are the potential mechanisms where a membrane protein could change the deformability of the entire cell?

We appreciate the reviewer's question regarding how a membrane-localized ion channel such as PIEZO1 could influence whole-cell deformability. While PIEZO1 is primarily known for its role in mechanosensitive calcium influx, increasing evidence suggests that its activity has broader implications for cell mechanics, even under resting conditions.

To directly assess this, we performed real-time deformability cytometry (RTDC) on wild-type (WT) and PIEZO1 knockout (KO) melanoma cells under control (unsqueezed) conditions, thereby isolating effects on intrinsic mechanical properties from those induced by external challenge. We found that PIEZO1 KO cells exhibit reduced deformability and an increased apparent Young's modulus compared to WT cells (see Rebuttal Figure 1 below). This suggests that PIEZO1 plays a role in maintaining a more compliant baseline mechanical phenotype.

Mechanistically, PIEZO1-mediated calcium signalling is known to modulate actomyosin contractility, membrane-cortex tension, and cytoskeletal organization through downstream pathways such as RhoA/ROCK and YAP/TAZ³⁵⁻³⁸. These pathways are critical regulators of cell stiffness and elasticity, even in the absence of acute mechanical stimulation. The difference observed in our KO cells may therefore reflect a disruption in these homeostatic processes.

[REDACTED]

Rebuttal Figure 1. PIEZO1 contributes to the baseline mechanical phenotype of melanoma cells. (A) Contour plots of cell deformation versus projected area obtained by real-time deformability cytometry (RT-DC) for wild-type (WT_CTRL, dark blue) and PIEZO1 knockout (KO_CTRL, light blue) melanoma cells under control (unsqueezed) conditions. PIEZO1-KO cells display lower deformation values at comparable cell sizes, indicating reduced cellular deformability compared to WT cells. (B) Corresponding contour plots of apparent Young's modulus versus cell area. KO cells exhibit higher apparent stiffness compared to WT cells, supporting a role for PIEZO1 in modulating cellular elasticity at baseline.

11) Figure 6D, the integrity of the endothelium seems to be damaged in all cases with cancer cells. 6J, what are the meaning of control and vessel?

We agree that the presence of melanoma cells induces visible changes in endothelial integrity across all conditions as seen in Figure 6D. This is consistent with an inflammatory response triggered by tumour–endothelium interactions. However, as detailed in the manuscript, the degree of endothelial disruption is cell-type dependent. Specifically, squeezed WT melanoma cells induce more pronounced junctional remodelling, as evidenced by VE-cadherin disorganization, while squeezed PIEZO1 knockout (KO) cells have a markedly reduced effect. This distinction supports our conclusion that PIEZO1 contributes to the capacity of tumour cells to disrupt endothelial barriers during extravasation. We have added asterisks in Figure 6D to better highlight the differences in endothelial integrity between conditions.

Regarding Figure 6J, “CTRL” refers to a chip without any cells, serving as a baseline for passive dye diffusion. “Vessel” refers to a chip containing an endothelial monolayer only (no cancer cells) and demonstrates reduced dye permeability due to the presence of an intact biological barrier. This comparison validates the functionality of the endothelial layer and allows us to attribute increased permeability in experimental groups to cancer cell-induced disruption.

We have clarified these points in the revised figure legend:

“(I) Bar graph showing permeability coefficient comparison for CTRL (cell-free device), Vessel, WT-CTRL (Vessel + WT melanoma cells non treated), WT-SQZD (Vessel + WT squeezed melanoma cells), KO-SQZD (Vessel + KO squeezed melanoma cells).”

Figure 6D. Representative fluorescence images of vascular endothelial proteins, e.g. Vascular endothelial cadherin junction (VE-Cadherin) and Vascular Cell Adhesion Protein (VCAM), when the endothelial monolayer is co-culture is WT and KO cells under control (CTRL) and squeezed (SQZD) experimental conditions. **Asterisks are reported in monolayer's area, where VE-Cadherin remodel and leave gap in between cells.**

12) The author should comment on whether the identified Piezo1 effects are purely Ca²⁺-based or may have to do with more unique molecular features of Piezo1.

We appreciate the importance of making this distinction. While PIEZO1 is best known as a mechanosensitive ion channel mediating Ca²⁺ influx, there are studies suggest its effects may extend beyond calcium signalling, potentially involving membrane-cytoskeleton interactions and transcriptional regulation. To investigate whether the effects observed in our study are primarily Ca²⁺-dependent, we conducted additional experiments in which melanoma cells were subjected to squeezing in calcium-free media. We specifically tracked the expression of key stemness markers, including CD44 and ABCB5. Under calcium-depleted conditions, we observed no upregulation of these markers, suggesting that PIEZO1's role in promoting stem-like features is indeed dependent on extracellular calcium influx.

These findings support the interpretation that the phenotypic changes observed are primarily driven by PIEZO1-mediated Ca²⁺ signalling. However, we do not exclude the possibility that additional, calcium-independent functions of PIEZO1 could contribute under different physiological contexts.

The new data have been included in the revised manuscript (New Fig. S11) and are now discussed in the Method section (p.9, lines 6-9):

“For calcium-deprivation experiments, cells were suspended in calcium-free DMEM (Gibco, Cat. Number 21068-028) throughout the constriction procedure to assess the role of extracellular calcium influx during mechanical stimulation. After constriction in calcium-free media, cells were similarly seeded for downstream analysis.”

Results sections (p. 25, lines 8-14):

“Finally, to directly assess the requirement of extracellular calcium influx, we performed squeezing experiments under calcium-depleted conditions. In calcium-free media, mechanical squeezing failed to induce upregulation of stemness markers CD44 and ABCB5 (Fig. S11), indicating that PIEZO1-mediated calcium entry is necessary to initiate the stem-like phenotype. Together, these results support a model in which mechanical stress activates PIEZO1-dependent calcium signalling to drive melanoma stem cell-like reprogramming”.

And Discussion (p.33, lines 32-33: p. 34, lines 1-3):

“The increased calcium signalling observed in our microcapillary constriction model suggests a relationship between calcium dynamics and the emergence of a stem cell-like state. Consistent with this, when cells were squeezed in calcium-free media, we observed no upregulation of stemness markers, indicating that calcium signalling is required to mediate the phenotypic response to mechanical stress.”

Fig. S11 PIEZO1-induced stemness marker expression is dependent on extracellular calcium influx. Representative immunofluorescence images showing CD44 (A) and ABCB5 (C) expression in melanoma cells under different conditions: control (CTRL) and squeezed cells incubated in calcium-free media during and after squeezing (SQZD Ca²⁺-free). Quantification of fluorescence intensity for CD44 (B) and ABCB5 (D) across experimental groups.

13) The use of Ruthenium Red as a Piezo1 blocker is not well justified. How specific is the blocker? How does it compare to potentially more broadly used blockers such as GxMT4?

We appreciate the reviewer’s point regarding the specificity of mechanosensitive channel blockers. Ruthenium Red (RR) is a broad-spectrum inhibitor and not selective for PIEZO1 specifically. However, it is one of the most widely used tools to probe ion channel activity due to its consistent performance and ability to inhibit multiple calcium-permeable channels, including

mechanosensitive ion channels, transient receptor potential (TRP) channels, ryanodine receptors, and mitochondrial calcium uniporters³⁹⁻⁴².

Nevertheless, we considered the reviewers point and performed additional experiments using the suggested GsMTx4. While GsMTx4 is also not exclusively specific to PIEZO1⁴³, unlike RR which blocks a broad spectrum of Ca²⁺ pathways, it primarily targets stretch-activated ion channels. This added specificity toward mechanosensitive channels helps narrow down the role of membrane tension in driving the observed stem-like phenotype. Treatment with GsMTx4 led to reduced expression of stemness markers and a significant decrease in tumorsphere area, consistent with our results using (RR) and PIEZO1 knockout cells. These findings further support the interpretation that PIEZO1-mediated mechanosensation, and the associated calcium influx, are critical for inducing the stem-like phenotype following mechanical deformation.

We have included these new data in the revised manuscript (New Fig. S10) and discussed in the Method (p.9, lines 31-34: p. 10, lines 1-5):

“For functional assessment of PIEZO1 activation under mechanical stress, blockers were also applied during the constriction experiments. Specifically, for microcapillary constriction assays, cells were prepared and suspended in DMEM according to the protocol described in the "Microcapillary-like constriction experiment" section. Immediately prior to injection into the microfluidic device, either Ruthenium Red (30 μM) or GsMTx4 (10 μM) was added directly to the cell suspension, allowing continuous exposure to the inhibitor during the constriction procedure. Following transit through the microcapillary device, cells were immediately collected and seeded into 96-well plates under normal growth conditions without additional inhibitor treatment.”

Results sections (p.24, lines 3-8):

“To further examine the specificity of mechanosensitive channel involvement, we repeated the experiments using GsMTx4, a more selective blocker of stretch-activated ion channels. Consistent with the results obtained using Ruthenium Red, treatment with GsMTx4 suppressed the upregulation of CD44 and ABCB5 following mechanical squeezing and significantly reduced tumorsphere formation capacity (**Fig. S10**). These findings strengthen the evidence for a mechanosensitive ion channel-dependent mechanism underlying the observed phenotypic changes”

And Discussion (p. 35, lines 2-4):

“Treatment with GsMTx4, a more selective inhibitor of stretch-activated ion channels, produced similar effects, further supporting the role of mechanically activated calcium influx in driving this phenotype.”

Fig. S10 Inhibition of mechanosensitive channels with GsMTx4 reduces stemness marker expression and tumorsphere formation following mechanical squeezing. (A) Representative immunofluorescence images of CD44 and ABCB5 expression in control (CTRL), squeezed (SQZD), and both cells group treated with GsMTx4. **(B, C)** Quantification of fluorescence intensity for CD44 and ABCB5 across experimental groups. **(D)** Graph representative of the mean tumorsphere area over time when cultured in serum-free media, for CTRL (blue), SQZD (red), and both cells group treated with GsMTx4 (light blue, orange).

Minor suggests:

1)It would be helpful to more clearly discuss the novelty of the current findings with respect to previous studies of mechanically induced metastatic transition and of Piezo-mediated cell mechanics.

As suggested, we have now added a dedicated section in the revised Discussion that more clearly outlines the novelty of our findings in the context of previous studies on mechanically induced metastatic transitions and PIEZO1-mediated cell mechanics (p.37, lines 8-14):

“Previous studies have demonstrated that mechanical deformation during confined migration can promote nuclear remodelling, chromatin changes, and invasive behaviour; however, these models have largely focused on interstitial matrix confinement rather than the mechanical challenges encountered by circulating tumour cells during microcapillary transit. Our study addresses this gap by modelling the transient, extreme deformations experienced in capillary-scale constrictions, revealing that such deformation alone is sufficient to induce a stem cell-like reprogramming in melanoma cells. ”

2)The (i) to (iv) label in figure 1A are wrong. And several labels of figure axis are too small to read.

We thank the reviewer for pointing this out. The (i) to (iv) labels in Figure 1A were inadvertently mismatched between the main text and the figure panel. We have corrected this labelling error to ensure consistency. Additionally, we have enlarged the axis labels across the figure panels to improve readability and overall clarity.

3)The concentration of Yoda1 (20 µM) is somewhat high, have the authors tested the effect of lower Yoda1 concentration?

The concentration of Yoda1 (20 µM) used in our experiments was based on prior studies in the literature where similar concentrations have been shown to effectively activate PIEZO1 in cancer and other cell types ⁴⁴. While we acknowledge that this is on the higher end of the effective range, our goal was to ensure robust activation for the purpose of mimicking mechanical stimulation.

We did not test lower concentrations in this study, as our focus was not on dose-response profiling but rather on evaluating the downstream effects of PIEZO1 activation. We agree that investigating concentration-dependent effects of Yoda1 will be an important follow-on study in future work.

4)Please be more specific on what CAAX motif was used for membrane label.

The protocol for engineering and labelling these cells has been previously published and is referenced in the Methods section of the manuscript ⁴⁵. For completeness, the relevant section of the Methods is reproduced below.

Briefly, to visualise A375-MA2 cells with live-cell microscopy, a construct was designed to label the plasma membrane via the expression of mScarlet with the CAAX motif (Figure 2.1; Table A 1. Adapted from reference 45). This construct was sequenced using primers outlined in Table 2.3 (Ramaciotti Centre for Genomics, UNSW). The mScarlet-CAAX plasmid DNA construct was linearised with BamHI-HF (NEB) for 2 h at 37°C and the restriction digest was run on a 1% agarose gel. Bands of linearised plasmid DNA were excised and purified using a QIAquick gel extraction kit (Qiagen) according to manufacturer’s instructions. A transfection mixture was administered to A375-MA2 CRISPR/Cas9-edited clones at a 1:25 volume before a 24-h incubation at 37°C in 5% CO₂. Growth media was then changed with warm complete media. A mixed population of transfected cells was cultured in complete culture media for 1 week, after which cells were sorted into sub-populations based on fluorescence intensity.

Figure 2.1 Visual schematic outlining key features mScarlet-CAAX plasmid construct. Features include the CMV enhancer (61 bp – 364 bp; grey), CMV promoter (365 bp – 568 bp; grey), mScarlet (660 bp – 1355 bp; red) and CAAX motif (1356 bp – 1433 bp; yellow). The plasmid was linearised and stably transfected into all CRISPR/Cas9-edited clones (from both WM266-4 and A375 cell lines). Schematic drawn to scale. Refer Table A 1 for construct details. ***adapted from reference 45***

Table 2.3 Sequencing primers for mScarlet-CAAX construct.

Sequencing primer	Sequence (5' → 3')
Forward primer CMV-forward	CGCAAATGGGCGGTAGGCGTG
Reverse primer safeharbour_amp_r	TTTGTAAATAACCGCGGAATACG

Table A 1 Features of the mScarlet-CAAX construct.

Features of the construct	Sequence (5' → 3')
CMV enhancer	CGT TAC ATA ACT TAC GGT AAA TGG CCC GCC TGG CTG ACC GCC CAA CGA CCC CCG CCC ATT GAC GTC AAT AAT GAC GTA TGT TCC CAT AGT AAC GCC AAT AGG GAC TTT CCA TTG ACG TCA ATG GGT GGA GTA TTT ACG GTA AAC TGC CCA CTT GGC AGT ACA TCA AGT GTA TCA TAT GCC AAG TAC GCC CCC TAT TGA CGT CAA TGA CGG TAA ATG GCC CGC CTG GCA TTA TGC CCA GTA CAT GAC CTT ATG GGA CTT TCC TAC TTG GCA GTA CAT CTA CGT ATT AGT CAT CGC TAT TAC CAT G
CMV promoter	GTG ATG CGG TTT TGG CAG TAC ATC AAT GGG CGT GGA TAG CGG TTT GAC TCA CGG GGA TTT CCA AGT CTC CAC CCC ATT GAC GTC AAT GGG AGT TTG TTT TGG CAC CAA AAT CAA CGG GAC TTT CCA AAA TGT CGT AAC AAC TCC GCC CCA TTG ACG CAA ATG GGC GGT AGG CGT GTA CGG TGG GAG GTC TAT ATA AGC AGA GCT
mScarlet	ATG GTG AGC AAG GGC GAG GCA GTG ATC AAG GAG TTC ATG CGG TTC AAG GTG CAC ATG GAG GGC TCC ATG AAC GGC CAC GAG TTC GAG ATC GAG GGC GAG GGC GAG GGC CGC CCC TAC GAG GGC ACC CAG ACC GCC AAG CTG AAG GTG ACC AAG GGT GGC CCC CTG CCC TTC TCC TGG GAC ATC CTG TCC CCT CAG TTC ATG TAC GGC TCC AGG GCC TTC ACC AAG CAC CCC GCC GAC ATC CCC GAC TAC TAT AAG CAG TCC TTC CCC GAG GGC TTC AAG TGG GAG CGC GTG ATG AAC TTC GAG GAC GGC GGC GCC GTG ACC GTG ACC CAG GAC ACC TCC CTG GAG GAC GGC ACC CTG ATC TAC AAG GTG AAG CTC CGC GGC ACC AAC TTC CCT CCT GAC GGC CCG TAA TGC AGA AGA AGA CAA TGG GCT GGG AAG CGT CCA CCG AGC GGT TGT ACC CCG AGG ACG GCG TGC TGA AGG GCG ACA TTA AGA TGG CCC TGC GCC TGA AGG ACG GCG GCC GCT ACC TGG CGG ACT TCA AGA CCA CCT ACA AGG CCA AGA AGC CCG TGC AGA TGC CCG GCG CCT ACA ACG TCG ACC GCA AGT TGG ACA TCA CCT CCC ACA ACG AGG ACT ACA CCG TGG TGG AAC AGT ACG AAC GCT CCG AGG GCC GCC ACT CCA CCG GCG GCA TGG ACG AGC TGT ACA AG
CAAX motif	CCG AAT TCC CGG GTC AAG ATG AGC AAA GAT GGT AAA AAG AAG AAA AAG AAG TCA AAG ACA AAG TGT GTA ATT ATG TAA

5)'u' was used instead of 'μ' in several units.

We apologize for this error and have replaced all instances of 'u' with the correct 'μ' symbol in units throughout the manuscript.

6)Please be more specific about how total cell number (and dead cells) were determined in viability assay? Did the author consider the possibility of trypan blue staining live cells?

In the original manuscript, cell viability was reported through the trypan blue exclusion assay. Total and dead cell numbers were determined manually using a hemocytometer immediately after staining. Cells that excluded the dye were counted as viable, while trypan blue–positive cells were classified as non-viable.

This information has been clarified in the revised Methods section (p 4, Lines 31-33, p.5, lines 16).

“Cell Viability. Cell viability was assessed using the trypan blue exclusion method. Following treatment or mechanical constriction, cells were collected, resuspended in PBS, and mixed 1:1 with 0.4% trypan blue solution (Thermo Fisher Scientific). The cell suspension was incubated at room temperature for 2–3 minutes and then immediately loaded into a hemocytometer. Viable (unstained) and non-viable (blue-stained) cells were manually counted under a light microscope. Total cell number and percentage viability were calculated from at least three independent fields per sample. All samples were processed in triplicate and analysed promptly to avoid staining artifacts.”

Reviewer #2 (Remarks to the Author):

Silvani and colleagues show that capillary-like constriction triggers phenotypic changes in melanoma cells toward a tumorigenic stem cell-like state through PIEZO1 mechanosensation. Authors show that, upon constriction melanoma cells display membrane deformation, which is partially recovered, chromatin condensation, and changes in chromatin marks, with reduced H3K9Ac and increased HeK9me3. Moreover, after crossing the microcapillaries, cells show increased expression of molecular markers of stemness, enhanced interactions with endothelial cells, and increased tumour sphere formation capacity. Using pharmacological agents and gene editing approaches, they conclude that PIEZO1 is responsible for these phenotypic changes.

Findings are intriguing and in line with recent studies showing mechanical compression as a driver of phenotypic changes in cancer cells including fostering metastatic potential. However, most of the conclusions are only backed by experiments in a single cell line, which is only representing a subtype of melanoma, and the *in vivo* studies are weak and not sufficiently powered to make any valid conclusions. Also, PIEZO modifying cancer phenotype is not fully novel.

We thank the reviewer for the positive assessment of the work and for pointing out areas in need of further study. Below we describe how we have addressed these issues in our revised manuscript.

Major issues:

1. Unclear if findings are generalizable across cancer types. Authors make general statements, most of the study was conducted on a single cell line (A375-M2). There is insufficient justification for using melanoma as a model and how much the findings apply to other cancer types. Therefore, authors should conduct studies in other melanoma cell lines, and in other cancer types if they want to make general claims beyond melanoma. Otherwise, the conclusions should be tempered, and the word ‘melanoma’ added to the title.

We appreciate the reviewer’s thoughtful comment regarding the generalizability of our findings beyond the A375-M2 melanoma model. To address this important point, we have extended our analysis to include additional tumour types: (i) the parental A375 melanoma line, (ii) triple-negative breast cancer (MDA-MB-231 cells), and (iii) osteosarcoma (HT-1080 cells).

For the *in vivo* metastasis assays, we specifically utilized the parental A375 melanoma cells, which had been previously engineered to express a luciferase reporter construct, thus allowing us to track metastatic progression via bioluminescence imaging. The use of this parental line provides additional evidence that the mechanically induced acquisition of metastatic potential is not limited to the A375-M2 subline but can also be observed in the parental melanoma context.

In our additional *in vitro* models, both MDA-MB-231 and HT-1080 cells exhibited a similar response to mechanical constriction, with upregulation of stemness-associated markers (CD44, CD133) and enhanced tumorsphere formation. However, we did observe some differences in the temporal dynamics between tumour types. In breast cancer and melanoma, the acquisition of stemness markers occurred within a shorter timeframe (4–24 hours), enabling clear detection of tumorsphere formation within a standard 7-day assay. In contrast, in the osteosarcoma model, we observed a delayed response, with significant increases in stemness markers emerging after 48 hours post-constriction. This delay limited the applicability of the tumorsphere assay in osteosarcoma, as the full phenotypic shift occurs beyond the optimal tumorsphere formation window used for the other models. Nonetheless, across all tumour types evaluated, the trend of mechanical induction of stem-like traits following microcapillary constriction was consistently observed. The differences in temporal regulation could explain why this finding has not been made in previous studies of constriction.

Taken together, these additional experiments support the broader relevance of our findings, while also highlighting that certain tumour types may exhibit distinct kinetic profiles in their response to mechanical deformation. We have incorporated these new data into the supplementary material (New Fig. S14) and discussed their implications in the revised Discussion sections (p.37, lines 14-18).

“Furthermore, preliminary experiments in breast cancer and osteosarcoma cell lines showed similar upregulation of stem cell-like features following microcapillary constriction (**Fig. S14**), albeit with variation in the timing and magnitude of response. These findings suggest that mechanosensitive reprogramming may be a conserved phenomenon across multiple solid tumours, though its kinetics may differ depending on cellular context.”

Fig. S14 Mechanical constriction induces stemness markers and tumorsphere formation across additional cancer cell types. Triple-negative breast cancer cells (MDA-MB-231) and osteosarcoma cells (HT-1080) were subjected to microcapillary constriction (SQZD) or maintained as controls (CTRL). Representative immunofluorescence images show expression of stemness markers CD44 (green) and CD133 (red) following mechanical constriction for MDA-MB-231 cells (A, C) and HT-1080 cells (G, I). Quantification of fluorescence intensity demonstrates upregulation of CD44 and CD133 expression after constriction in both cancer types; MDA-MB-231 cells (B, D) and HT-1080 cells (H, L). (E) Representative brightfield images of tumorsphere formation for MDA-MB-231 cells. (F) Graph representative of the mean tumorsphere area over time when cultured in serum-free media, for CTRL (red), SQZD (red) MDA-MB-231 cells.

2. The metastasis experiments presented are inadequate. Tail vein injection is not a true model of metastasis; cancer cells are trapped in the lung capillaries and therefore this assay is more indicative of tumour cell seeding. Intracardiac injection or intradermal injection followed by survival surgery would truly recapitulate the metastatic process. Also, the number of mice (n= 4 per group) is clearly insufficiently powered for this type of assays. Differences in Fig 3F are not statistically significant. A true quantitation of the number of metastatic foci and the size of those mets (tumour burden) is missing.

We thank the reviewer for this critical and constructive comment regarding the limitations of the tail vein injection model. In response, and to strengthen the biological relevance of our findings, we have now performed intra-cardiac injection experiments using A375-luciferase melanoma cells to model systemic metastasis more accurately. We used the lowly metastatic parental line as a means to demonstrate how constriction elevates stemness and metastatic potency. These experiments were conducted across two independent cohorts (n = 8 for CTR, n = 10 for SQZ), and demonstrated that microcapillary-like constriction increases metastatic burden and significantly reduces survival. Specifically:

- Kaplan–Meier analysis showed a significant decrease in survival in the squeezed (SQZ) group starting from week 4.
- Whole-body bioluminescence imaging at week 4 showed a significantly higher median photon flux in SQZ-injected mice.
- Quantification of photon flux from individual organs revealed increased metastatic colonization in the lungs, and significantly elevated signals in bone and brain in the SQZ group compared to controls.

These data have been included as a new figure in the main manuscript (New Fig. 3) and expanded upon in the Results and Discussion sections. We appreciate the reviewer's comments, which prompted us to strengthen the *in vivo* component of our study with a more physiologically relevant and statistically powered metastasis model. We are excited about these findings which open up numerous avenues to further explore how mechanotransduction through constriction influences metastasis.

The new data have been included in the revised manuscript and are now discussed in detail in the Results sections (p.22, lines 13-23):

“Building upon this preliminary finding, we next expanded and validated our observations using a more sensitive and systemic intracardiac injection model to assess global metastatic dissemination and organotropic colonisation. Following injection of 1×10^5 A375-Luc2 melanoma cells into the left ventricle, mice receiving squeezed cells exhibited a striking increase in total metastatic burden and distribution. Longitudinal bioluminescence imaging demonstrated elevated whole-body photon flux in the SQZD group by week 4 (**Fig. 3G**), including increased median signals in the lung (**Fig. 3H**) and significantly higher signals in bone and brain (**Fig. 3I, L**), indicating elevated organ-specific homing. Importantly, survival analysis indicated that animals injected with squeezed cells had reduced overall survival, with multiple animals reaching ethical endpoints before day 50 (**Fig. 3M**). In fact, 9/10 animals in the SQZD group reached ethical endpoint by day 52, compared to 1/8 animals in the CTRL group. Collectively, these results demonstrate that micro constriction enhances the *in vivo* metastatic potency and lethality of melanoma cells in a systemic dissemination model.”

Fig. 3 Microcapillary-like constrictions enhance melanoma cell tumorigenicity. (A) Images of control (CTRL) and squeezed (SQZD) melanoma cells forming tumorsphere in serum-free media after 7 days of culture. (B) Graph representative of the mean tumorsphere area increasing over time when cultured in serum-free media, for CTRL (blue) and SQZD (red) cells. (C) Bar graph displaying the area of tumorsphere for CTRL and SQZD cells at day 7. The result is expressed as the mean from four independent experiments. (D) IVIS Imaging of lung metastases that developed after 30 days from injecting CTRL and SQZD luciferase melanoma cells into the lateral tail vein of 7-week-old female Balb/c Nude mice. [Note the change in scale between the groups]. (E, F) Representative H&E-stained lung sections from mice injected with CTRL cells (E) and SQZD cells (F) harvested 30 days post tail vein injection. (G) Whole body luciferase signal of mice injected intra-cardially at week 4, the final timepoint at which all mice in both groups were accounted for. (H-L) Photon flux from individual metastatic sites revealed increased median signals in the (H) lung and significantly higher signals in (I) bone and (L) brain in the SQZ group compared to controls. Each dot represents an individual animal. (M) Metastatic burden and Kaplan–Meier survival curves following IC injection with control (black) and squeezed (red) cells. Data represent pooled results from each group (CTRL, n = 8; SQZ, n = 10).

3. Mechanistically, the study lacks important aspects: How PIEZO protein levels or subcellular localization change in response to microcapillary 'squeezing'?

We appreciate the importance of tracking PIEZO1 localization, as this surely relates to function. As shown in Figure 4A and B, we have performed immunofluorescence microscopy to examine PIEZO1 expression following microcapillary-like squeezing, which demonstrates an increased PIEZO1 signal at the plasma membrane in squeezed cells compared to controls. This suggests that mechanical deformation enhances PIEZO1 expression and/or its recruitment to the membrane, potentially facilitating mechanosensitive signalling in response to capillary-like stress. Furthermore, in the revised manuscript we evaluated important downstream phosphosignals of Ca^{2+} /PIEZO1 signal transduction, noting nuclear translocation of these

proteins within 15 minutes of constriction (New Fig. S9). We have discussed these findings further in the revised manuscript (p. 24, lines 1- 14):

“To test this, we investigated PIEZO1 expression and localization at the plasma membrane using immunofluorescence, noting an increase in PIEZO1 signal intensity in squeezed melanoma cells when compared to un-squeezed controls (**Fig. 4A, B**). This suggests that mechanical deformation promotes membrane reorganization and clustering of PIEZO1, positioning the channel for enhanced mechanosensitive activation under capillary-like stress and facilitating downstream calcium influx.” ... “To evaluate signal transduction downstream of PIEZO1, we performed phosphoproteomic profiling at early time points post-constriction using multiplexed immunolabeling^{7, 8}. The analysis revealed dynamic changes in key mechanosensitive pathways including NF- κ B, PKC, YAP1, mTOR, and ERK, consistent with their established roles downstream of PIEZO1 activation (**Fig. S9**).”

Fig. S9. Phosphoprotein imaging reveals time-resolved activation of mechanosensitive signalling pathways. (A) Heatmap representing log₁₀ fold change in the activation of selected phospho-markers (NF- κ B, PKC, YAP1, mTOR, and p-ERK) at 7-, 15-, and 50-minutes post-constriction. Line plot showing quantification of N/C ratios for YAP1 (B), p-ERK (C), and NF- κ B (D) over time. Box plot representing mTOR nuclear intensity (E) and PKC cytoplasmic intensity (F) over time. Error bars indicate the Standard Error of the Mean (SEM).

Is the mechanism Ca²⁺-dependent or PIEZO1 conformation dependent.

To determine whether the downstream effects are specifically Ca²⁺-dependent, we repeated the squeezing assay in calcium-free media. Specifically, mechanical constriction failed to induce upregulation of stemness markers or promote tumorsphere formation (**New Fig.S11**), indicating that extracellular calcium is required for phenotypic reprogramming. Consistently, pharmacological inhibition of mechanosensitive ion channels with both Ruthenium Red (RR) (**Fig**

4) and GsMTx4 (**New Fig. S10**) prevented the emergence of the stem-like state following deformation. Furthermore, activation of PIEZO1 using the agonist Yoda1 was sufficient to promote stem-like features even in the absence of mechanical stimulation, consistent with the role of calcium influx as the critical downstream mediator. Together, these results strongly support that PIEZO1-dependent calcium entry is required to drive the phenotypic changes observed following mechanical deformation.

Calcium-free media

The new data have been included in the revised manuscript (New Fig. S11) and are now discussed in the Method section (p.10, lines 6-10):

“For calcium-deprivation experiments, cells were suspended in calcium-free DMEM (Gibco, Cat. Number 21068-028) throughout the constriction procedure to assess the role of extracellular calcium influx during mechanical stimulation. After constriction in calcium-free media, cells were similarly seeded for downstream analysis.”

Results sections (p. 25, lines 8-14):

“Finally, to directly assess the requirement of extracellular calcium influx, we performed squeezing experiments under calcium-depleted conditions. In calcium-free media, mechanical squeezing failed to induce upregulation of stemness markers CD44 and ABCB5 (**Fig. S11**), indicating that PIEZO1-mediated calcium entry is necessary to initiate the stem-like phenotype. Together, these results support a model in which mechanical stress activates PIEZO1 -dependent calcium signalling to drive melanoma stem cell-like reprogramming.”

And Discussion (p.33, lines 32-33: p. 34 lines 1-3):

“The increased calcium signalling observed in our microcapillary constriction model suggests a relationship between calcium dynamics and the emergence of a stem cell-like state. Consistent with this, when cells were squeezed in calcium-free media, we observed no upregulation of stemness markers, indicating that calcium signalling is required to mediate the phenotypic response to mechanical stress.”

Fig. S11 PIEZO1-induced stemness marker expression is dependent on extracellular calcium influx. Representative immunofluorescence images showing CD44 (A) and ABCB5 (C) expression in melanoma cells under different conditions: control (CTRL) and squeezed cells incubated in calcium-free media during and after squeezing (SQZD Ca²⁺-free). Quantification of fluorescence intensity for CD44 (B) and ABCB5 (D) across experimental groups.

GsMTx4 inhibition

We have included these new data in the revised manuscript (New Fig. S10) and discussed in the Method (p.9, lines 31-34: p. 10, lines 1-5):

“For functional assessment of PIEZO1 activation under mechanical stress, blockers were also applied during the constriction experiments. Specifically, for microcapillary constriction assays, cells were prepared and suspended in DMEM according to the protocol described in the "Microcapillary-like constriction experiment" section. Immediately prior to injection into the microfluidic device, either Ruthenium Red (30 μ M) or GsMTx4 (10 μ M) was added directly to the cell suspension, allowing continuous exposure to the inhibitor during the constriction procedure. Following transit through the microcapillary device, cells were immediately collected and seeded into 96-well plates under normal growth conditions without additional inhibitor treatment.”

Results sections (p.25, lines 3-8):

“To further examine the specificity of mechanosensitive channel involvement, we repeated the experiments using GsMTx4, a more selective blocker of stretch-activated ion channels. Consistent with the results obtained using Ruthenium Red, treatment with GsMTx4 suppressed the upregulation of CD44 and ABCB5 following mechanical squeezing and significantly reduced tumorsphere formation capacity (**Fig. S10**). These findings strengthen the evidence for a mechanosensitive ion channel-dependent mechanism underlying the observed phenotypic changes”

And Discussion (p. 35, lines 2-4):

“Treatment with GsMTx4, a more selective inhibitor of stretch-activated ion channels, produced similar effects, further supporting the role of mechanically activated calcium influx in driving this phenotype.”

Fig. S10 Inhibition of mechanosensitive channels with GsMTx4 reduces stemness marker expression and tumorsphere formation following mechanical squeezing. (A) Representative immunofluorescence images of CD44 and ABCB5 expression in control (CTRL), squeezed (SQZD), and both cells group treated with GsMTx4. (B, C) Quantification of fluorescence intensity for CD44 and ABCB5 across experimental groups. (D) Graph representative of the mean tumorsphere area over time when cultured in serum-free media, for CTRL (blue), SQZD (red), and both cells group treated with GsMTx4 (light blue, orange).

How does PIEZO1 change results in H3K9me3 increase and expression of stem-like genes?

We thank the reviewer for raising this important mechanistic question. To explore how PIEZO1 activation leads to increased H3K9me3 and the expression of stem-like genes, we first performed immediate transcriptomics profiling (within 15 minutes of constriction), which suggests increased expression of genes related to invasive processes with some positive regulation of several transcripts associated with de-differentiation (e.g. THY1, ZNF385B, TGF β pathway components, and UCHL1). However, we acknowledge that transcriptomics does not paint a clear picture of “stemness” regulation.

To better understand the immediate signal transduction post constriction that could underlie these changes, we performed phosphoproteomic profiling of melanoma cells after mechanical deformation through microcapillary-like constrictions, using a novel multiplexed immunolabelling approach that allows simultaneous assessment of biological markers in each cell^{7, 8}. Our analysis revealed activation of several key signalling pathways downstream of PIEZO1-mediated calcium influx, including increased nuclear translocation of NF- κ B and YAP, as well as nuclear activation of mTOR, membrane recruitment of PKC, and rapid activation of p-ERK. These pathways are known to be responsive to intracellular calcium and have established roles in modulating chromatin state and transcriptional regulation. Specifically, NF- κ B and YAP have both been implicated in the transcriptional activation of stemness-associated genes¹³⁻¹⁷, while pathways such as PKC and mTOR can regulate epigenetic modifiers, including those involved in histone methylation¹⁸⁻²⁰. Therefore, we propose that mechanical activation of PIEZO1 leads to these calcium-dependent signalling cascades that promote chromatin remodelling and regulation of transcriptional programs linked to stemness and invasive characteristics. These new data have been included as a new figure in the revised manuscript (see new Fig. S9), with expanded results in the main text (p.24, lines 10-14).

“To evaluate signal transduction downstream of PIEZO1, we performed phosphoproteomic profiling at early time points post-constriction using multiplexed immunolabeling^{7, 8}. The analysis revealed dynamic changes in key mechanosensitive pathways including NF- κ B, PKC, YAP1, mTOR, and ERK, consistent with their established roles downstream of PIEZO1 activation (**Fig. S9**).”

And discussion section to highlight their relevance to the proposed mechanism (p.35, lines 2032).

“In line with PIEZO1-mediated calcium influx, we observed distinct activation patterns across several mechanosensitive signalling pathways following microcapillary constriction. Specifically, the phosphorylated state of NF- κ B and YAP exhibited increased nuclear localisation, consistent with their established roles in transcriptional regulation of plasticity and survival-associated gene programs¹³⁻¹⁷. Nuclear enrichment of mTOR was also observed, supporting its involvement in epigenetic regulation following mechanical stress^{18, 19}. In contrast, pERK showed a brief increase in nuclear localization, which then decreased, suggesting a rapid transcriptional response that is tightly controlled to prevent prolonged ERK activation, which could otherwise disrupt cellular homeostasis²⁰. Finally, PKC activity exhibited a rapid cytoplasmic spike immediately after constriction, likely driven by calcium influx, followed by downregulation over time as homeostasis was re-established. Collectively, these early phosphorylation events downstream of PIEZO1 likely converge to promote transcriptional plasticity and support the emergence of a melanoma stem cell-like state.”

4. Limited evidence of clinical relevance. Authors should provide patient-derived data supporting their findings.

We acknowledge the reviewer's point regarding clinical relevance and the use of patient-derived cells. While this is indeed an important direction for future validation, we have found that the high heterogeneity in size and mechanical properties of patient-derived tumour cells presents a practical limitation for our microfluidic squeezing platform. The device is calibrated for precise channel dimensions and adapting it to accommodate variable cell sizes would require extensive redesign and optimization for each individual patient sample, an effort that is technically demanding, time-consuming, and costly. Nevertheless, we were able to deform a subpopulation of cells from a BRAF mutant patient-derived cell line (gift from colleagues at the Mayo Clinic) using our system and observed a clear increase in CD44 expression after microconstriction (see Rebuttal Figure 2 below). However, considerable work remains to make this simple microfluidic platform amenable to a broad spectrum of patient cells, which we believe lies outside of the scope of the current study.

[REDACTED]

Rebuttal Figure 2. Mechanical constriction induces CD44 expression in patient-derived melanoma cells. (A) Representative immunofluorescence images of CD44 staining in a BRAF-mutant patient-derived melanoma cell line under control conditions (CTRL) and following microcapillary-like constriction (SQZD). (B) Quantification of CD44 fluorescence intensity confirms increased CD44 expression following microconstriction.

We fully agree that validating our findings in patient-derived models will be a critical next step, and we plan to explore this in future studies by incorporating size-adaptive microfluidic technologies and broader clinical sampling.

Other specific points to be addressed:

Figure 1

1.1. It is not clear why authors decided to use A375-MA2 melanoma cells which are an increased metastatic version of the parental A375. If the scope of the study is to prove that compression increases metastatic potential, authors should have used the parental A375 which are less metastatic.

We initially chose the A375-MA2 melanoma cells because they represent a subpopulation of A375 cells that have already undergone *in vivo* selection for metastatic potential. As such, we believe they are more physiologically relevant for modelling circulating tumour cells that are likely to encounter physical constraints such as capillary-like constrictions. Our aim was to isolate and study the effect of mechanical stress during capillary transit, rather than earlier steps in the metastatic cascade.

That said, in the revised manuscript we have performed several new experiments using the parental A375 line (luciferase-modified) as suggested by the reviewer and found a comparable response to microconstriction with respect to molecular markers of stemness and *in vitro* tumorigenicity (Fig. S8). We also decided to use the parental A375 cells to perform the requested *in vivo* intracardiac injection experiments, as this model is relatively lowly metastatic compared to the MA2 and would better demonstrate how constriction activates stemness and pro-metastatic behaviour. Indeed, this experiment clearly demonstrates significant differences in survival and metastatic burden after microconstriction (new Fig. 3) as discussed in response to point 2 above.

Fig. S8 Verification of elevated tumorigenicity and stemness marker expression in A375-luciferase melanoma cells prior to *in vivo* experiments. (A) Representative staining of CD44 in CTRL and SQZD melanoma cells. Bar graph displaying the CD44 levels in CTRL and SQZD cells. (B) Representative staining of CD271 in CTRL and SQZD melanoma cells. Bar graph displaying the CD271 levels in CTRL and SQZD cells.

Also, in line with my comment above, confirming some key findings in additional melanoma cell lines and at least other 2 cancer models (i.e., breast, lung, ovarian) it is necessary to make general conclusions.

As mentioned in our previous response, we have extended our analysis beyond melanoma to include two additional cancer types: breast cancer (MDA-MB-231) and osteosarcoma (HT-1080) (See Fig. S14). Size constraints of the constriction device limited our ability to perform these experiments using the suggested lung and ovarian model. In both models, we observed a similar trend following mechanical constriction, including upregulation of stemness markers (e.g., CD44, CD133) and increased tumorsphere formation. These results support the idea that mechanical deformation through capillary-like constrictions can induce stem-like traits in a manner not limited to melanoma cells. However, we acknowledge that there are differences in the magnitude and duration of the response across cell types that will require additional follow up experiments. Nevertheless, we believe these experiments demonstrate that microconstriction regulated tumorigenicity could be a widely employed mechanoadaptation during metastasis from solid tumours which we hope will catalyse further research beyond melanoma. We appreciate the reviewer's suggestion, which helped strengthen the generalizability of our conclusions.

1.2. No justification is provided for choosing gradient 30-5µm channels; are these sizes based on physiological observations? How long time is required for the cells to pass through the whole device?

We thank the reviewer for this valuable observation. The gradient of 30 µm to 5 µm in our microfluidic channels was intentionally designed to mimic physiologically relevant dimensions encountered by circulating tumour cells (CTCs) as they transition from larger vessels into narrow capillary beds. Capillaries in human and murine tissues often range from 5 to 10 µm in diameter, and constrictions as narrow as 5 µm have been observed *in vivo* using intravital imaging studies^{46, 47}. The larger upstream channel dimensions allow for smooth cell entry and flow alignment before encountering the physiologically relevant constrictions.

We have clarified these design choices and added relevant references to the revised Result section (p.17, lines 3-6).

“These dimensions were chosen to reflect the range of vessel sizes encountered during microvascular transit, with the final 5 µm constrictions approximating capillary diameters observed *in vivo*⁴⁶⁻⁴⁸.”

Regarding transit time, high-speed live imaging performed under our experimental flow conditions (260 µL/min) allowed us to estimate that cells typically pass through each individual constriction within milliseconds. The total residence time for a cell to traverse the entire microfluidic device, from inlet to outlet, is on the order of a few seconds. This design allows us to replicate both the magnitude and rapidity of mechanical stress that circulating tumour cells experience during dissemination.

1.3. The viability was measured exclusively by Trypan blue (according to material and methods) which can create bias in immunofluorescence staining.

We appreciate the reviewer raising this point. We would like to clarify that Trypan Blue staining was used exclusively for assessing cell viability and was not used in conjunction with immunofluorescence imaging. Therefore, we believe trypan blue was a suitable choice for assessment of cell viability. We have made sure to clarify this point in the revised manuscript.

In addition, what is the control? Are cells subjected to the same flow rate but in a channel-free device or cells at the inlet? This can create a strong biological bias between control and squeezed cells. Please provide a clear description of controls.

In our initial experiments we tested a “no-channel” version of the microfluidic device, which allows cells to pass through the chip under the same flow rate but without encountering any constrictive elements.

We compared this condition to seeding cells directly on glass (no flow) and observed no significant changes in key readouts (e.g., CD44/ABCB5 expression and tumorsphere formation) between these control conditions, thereby validating the use of this control. Importantly, while shear stress has been shown to affect cancer cell behaviour, those studies generally involve continuous shear exposure for minutes to hours⁴⁹⁻⁵². In contrast, our microfluidic setup exposes cells to flow for only milliseconds, a duration unlikely to activate canonical shear-responsive mechano-transduction pathways.

These results support the conclusion that the phenotypic changes we observe are not attributable to fluid shear, but rather to the mechanical confinement and deformation experienced during transit through capillary-like constrictions. We have included this result in

supplementary material (New Fig. S7) and clarified in the revised Discussion sections (p.21, lines 22-32):

“To confirm that deformation, rather than flow alone, drives the observed effects, we conducted control experiments using a non-constrictive device where cells experienced identical flow rates without confinement. No changes in stemness markers were observed under these conditions, supporting that mechanical constriction is the critical factor driving phenotypic reprogramming (Fig. S7 A-D).” ...” Notably, no tumorspheres formation were observed in control conditions lacking mechanical deformation, underscoring the role of constriction-induced reprogramming (Fig. S7 E, F).”

Fig. S7 Mechanical deformation, not flow-induced shear stress, drives acquisition of stem-like traits. Immunofluorescence analysis of CD44 (green, A) and ABCB5 (red, C) expression in control (static culture) and flow-only (no constriction) conditions. (B, D) Quantification of CD44 and ABCB5 fluorescence intensity. (E) Representative brightfield images of tumorsphere formed from control and flow-only groups. (F) Tumorsphere growth curves over time, showing no significant differences between control and flow-only conditions.

1.4. Authors should perform γ H2AX to evaluate the activation of DNA damage, total H2AX is insufficient.

We thank the reviewer for this helpful comment. We agree that γ H2AX, the phosphorylated form of H2AX (Ser139), is the appropriate marker for assessing DNA damage response. We would like to clarify that we did use γ H2AX in our experiments; however, it was incorrectly referred to in the manuscript as total H2AX. Specifically, we used the antibody Affinity Biosciences, Cat# AF3187, which targets phospho-Histone H2AX (Ser139) and is indeed the correct marker referred to as γ H2AX. We have corrected this throughout the revised manuscript, including the text, figure legends, and Methods section. We appreciate the reviewer’s attention to this important detail.

1.5. How long after deformation histone modifications are maintained?

We thank the reviewer for this important question. To assess the persistence of histone modifications following mechanical deformation, we performed time-course analyses for both H3K9ac and H3K9me3. We observed that the increase in H3K9me3 levels remained significantly elevated compared to control up to 24 hours post-deformation, suggesting a relatively stable shift toward a more repressive chromatin state. In contrast, H3K9ac levels showed a significant decrease at 4 hours, but this reduction was no longer detectable at 24 hours, indicating a more transient deacetylation response. These findings suggest a two-phase epigenetic response to mechanical stress: an early and reversible loss of acetylation, potentially linked to immediate transcriptional repression or chromatin remodelling, followed by a longer-lasting accumulation of H3K9me3, which may reflect a more stable silencing mechanism associated with chromatin compaction and de-differentiation. This biphasic regulation is consistent with stress-induced epigenetic remodelling described in other systems, where transient histone deacetylation is followed by the accumulation of repressive methylation marks, contributing to longer-term transcriptional silencing and cellular reprogramming⁵³⁻⁵⁵.

We have included this result in supplementary material (New Fig. S3) and clarified in the revised Result sections (p.19, lines 7-13):

“These histone modifications occurred rapidly, within 15 minutes of mechanical deformation, and reflect a shift from transcriptionally active to repressive chromatin states. To assess the stability of these changes, we performed additional time-course experiments, which revealed that H3K9ac reduction was transient and largely restored by 24 hours, while H3K9me3 remained stably elevated (Fig. S3). This suggests that mechanical stress induces chromatin condensation and transcriptional silencing in specific genomic regions, priming cells for downstream transcriptional reprogramming and adaptive phenotypic changes.”

Fig. S3. Temporal dynamics of histone modifications following mechanical deformation. Representative immunofluorescence images of H3K9ac (A) and H3K9me3 (C) staining at different times post-constriction in control (CTRL) and squeezed (SQZD) A375-MA2 cells. Bar graphs displaying the quantification of ABCB5 H3K9ac (B) and H3K9me3 (D) mean fluorescence intensity levels in CTRL and SQZD cells at multiple time points (4 hrs to 72 hrs).

1.6. In line with my previous concern, immunofluorescence should be repeated in the

presence of phospho-Caspase3 to quantify the increase in Hoechst and H3K9me3 within alive cells.

We understand the concern regarding distinguishing histone modifications in viable versus apoptotic cells. However, we note that the increase in H3K9me3 and Hoechst intensity is observed hours after deformation and remains detectable up to 24 hours (New Fig. S3), a timeframe during which apoptotic cells would typically be cleared or fragmented. This sustained difference over time strongly suggests that the modifications are present in cells that remain viable following constriction. While we did not co-stain with phospho-Caspase 3 in this assay, the persistence of the signal and the fact that these cells remain adherent and morphologically intact supports the conclusion that the epigenetic changes occur primarily in surviving, non-apoptotic cells.

1.7. Control and SQZD cells should be assessed by H3K9me3 ChIP-seq or CUT&Tag/Run to determine if/how H3K9me distribution across the genome changes upon microcapillary induced constriction. These analyses would reveal if the observed changes in histone marks are (at least partially) responsible for the observed differentially expressed genes.

We agree that genome-wide mapping of H3K9me3 distribution using techniques such as ChIP-seq or CUT&Tag would provide valuable information about the epigenetic landscape and its relationship to transcriptional changes. However, these proposed experiments would be very demanding of time and resources, which we believe are better suited as follow up work. Here we demonstrate functional consequences of mechanical deformation with evidence for changes in chromatin state, transcription, phosphosignalling, tumorigenicity and metastatic potential. We hope that these findings will spur a range of studies aimed at elucidating additional molecular detail, which are the logical next steps in exploring these phenomena.

Figs. 2 and 3

2.1. Related to the previous question on the amount of time required for the cells to pass through the device, the time point in which the RNAseq was performed is not clear. If the time window is too narrow, the true effects in gene expression may be lost due to the high stability of many RNAs. Also, author should consider again the concern in viability which may bias the overall differential expression analysis. Interestingly, PIEZO1 is linked to a mechano-dependent apoptosis pathway (Hope et al., Cell Death Dis 2019; Song et al, iScience 2022) which the authors should mention.

For the RNA-seq analysis, cells were lysed immediately after deformation in order to capture early transcriptional changes induced by mechanical stress. Regarding cell viability, only intact, viable cells were collected for RNA extraction. Following deformation, cells were pelleted by low-speed centrifugation to remove lysed cells and debris, and RNA was extracted directly from this enriched population. While some level of stress-induced apoptosis is expected, we focused our analysis on the surviving fraction, which displayed consistent transcriptomic and phenotypic changes across replicates.

We also appreciate the reviewer's reference to prior studies linking PIEZO1 to mechanosensitive apoptosis pathways^{5, 6}. These studies highlight PIEZO1's dual role in stress adaptation and apoptotic sensitivity. While our study centres on the adaptive responses in cells that survive mechanical constriction, we have added some discussion on the potential interplay between PIEZO1 signalling and apoptosis in the revised Discussion section (p.36, lines 22-32):

“While we primarily focused on the adaptive, tumorigenic phenotype that emerges from cells surviving mechanical deformation, it is important to recognize that PIEZO1 activation may also sensitize circulating tumour cells to immune-mediated apoptosis, as highlighted in recent studies involving TRAIL exposure under fluid shear stress conditions ^{5, 6}. Indeed, in our system, we observed a measurable loss in cell viability immediately after constriction (Fig. 1C), suggesting that mechanical stress may trigger cell death in a subset of cells. However, the long-term transcriptional and phenotypic reprogramming observed in the surviving population underscores the capacity of certain cells to resist apoptosis and adapt, acquiring stem-like and pro-invasive features. Thus, PIEZO1 may represent a double-edged sword in the metastatic cascade, promoting adaptation and metastasis in some cells, while increasing apoptotic vulnerability in others when exposed to immune effectors such as TRAIL-expressing NK cells, macrophages, or dendritic cells.”

2.2. Authors performed sphere assays which is a good proxy for clonogenicity in 3D justified by the increase in stemness-like genes. However, I find essential to complement this study with an *in vivo* tumour formation by intradermal/subQ injection followed by survival surgery, a true model of metastasis. As mentioned above, the number of animals presented in the tail vein model was insufficient (n=4). Power calculations should be conducted and provided, and the corresponding n of mice should be used.

We agree that intradermal or subcutaneous tumour formation assays are valuable for assessing local tumour initiation and growth. However, our study aimed to specifically model the dissemination and colonization phase of metastasis following mechanical stress, rather than primary tumour formation. To this end, we evaluated a model that better represents post-constriction environments (intracardiac injection assay, new Fig.3), which enables systemic distribution of tumour cells and allows for assessment of metastatic potential in distal organs such as bone, brain, and lungs. In our opinion, intradermal/subcutaneous growth could tell us about the population's tumorigenicity but does not relate to our aim of evaluating extravasation and metastatic outgrowth.

2.3. It is counterintuitive that there are almost no metastases by IVIS in the control conditions since A375-MA2 are highly metastatic. Is it possible that cells lost expression of the luciferase reporter? Quantitation of tumour burden by histological analyses could be more reliable. Indeed, in FIG3g, one can see micro-mets in the lungs H&E control although the bioluminescence shows no signal.

We appreciate the reviewer's careful observation and agree that the lack of bioluminescence signal in the control group may appear counterintuitive given the metastatic potential of A375-MA2 cells. However, we would like to clarify a few key points:

First, we do not believe that the absence of signal is due to loss of luciferase expression, as luciferase activity over time was confirmed in control cells *in vitro* prior to injection. Rather, this discrepancy is likely due to the sensitivity limitations of *in vivo* bioluminescence imaging, particularly in detecting small or early-stage metastatic foci, which may fall below the detection threshold of IVIS. This is especially relevant when signal is diffuse or located in deeper tissues, where absorption and scattering reduce photon output.

In light of these limitations, and to more robustly assess differences in metastatic potential between control and squeezed cells, we adopted an intracardiac injection model as suggested by the reviewer. This approach allows for a more systemic dissemination of cells, particularly to bone and brain, and provides improved sensitivity for capturing early metastatic events. Using this refined model, we observed clear and significant differences in metastatic burden, with

squeezed cells showing accelerated dissemination, higher tumour burden across multiple tissues, and reduced overall survival compared to controls, as detailed in the revised Results section (New Fig. 3).

Figure 4

3.1 PIEZO1 comes out of nowhere. Is this gene among the differentially expressed between control and squeezed cells? The choice of this gene should be justified.

We appreciate the reviewer's comment and the opportunity to clarify our rationale for focusing on PIEZO1. Our early selection of PIEZO1 as a key marker was driven by substantial existing literature linking PIEZO1 to mechanosensation, calcium influx, and downstream signalling pathways relevant to cancer cell invasion⁵⁶. Mechanosensitive ion channels, including PIEZO1, have been shown to mediate cellular adaptation to mechanical stress in multiple biological contexts, supporting the plausibility of its role in our model. While PIEZO1 expression was not differentially expressed at the transcript level immediately after constriction, immunofluorescence analysis revealed a notable increase in PIEZO1 protein expression and localization at the plasma membrane. Furthermore, we observed changes in calcium mediated signalling transcripts which we verified through live cell calcium imaging during microcapillary transit. Together, this evidence led us to believe that PIEZO1 was a strong candidate for mechanotransduction during constriction. Our subsequent experiments using pharmacological modulators and CRISPR/Cas9 knock out confirmed the role of PIEZO1 in mediating constriction induced stemness and invasiveness.

This rationale is now clearly stated in the revised Results sections (p23, lines 5-9; p. 24, lines 117):

Cells interpret mechanical stimuli through specialized mechanosensors, including mechanically gated ion channels. Among these, PIEZO1 has emerged as a critical regulator of mechanotransduction in cancer, previously implicated in responses to shear stress, membrane stretch, and compression⁵⁶. Given that our microcapillary model imposes abrupt mechanical deformation and elicits downstream changes associated with calcium handling, we hypothesized that PIEZO1 could play a key role in sensing and transducing these mechanical cues. **To** test this, we investigated PIEZO1 expression and localization at the plasma membrane using immunofluorescence, noting an increase in PIEZO1 signal intensity in squeezed melanoma cells when compared to un-squeezed controls (**Fig. 4A, B**). This suggests that mechanical deformation promotes membrane reorganization and clustering of PIEZO1, positioning the channel for enhanced mechanosensitive activation under capillary-like stress and facilitating downstream calcium influx. To verify this result we conducted calcium imaging of cells transiting the microcapillary. Using the calcium sensing dye Calbryte 520 AM, we observed rapid influx of calcium ions coinciding with cell deformation during live cell imaging (**Fig. 4C**). Notably, this calcium influx was primarily observed during transit through the narrower constrictions (10 μm and 5 μm), where significant deformation was induced. To evaluate signal transduction downstream of PIEZO1, we performed phosphoproteomic profiling at early time points post-constriction using multiplexed immunolabeling^{7, 8}. This analysis revealed elevated nuclear localisation of phosphorylated mechanotransduction elements downstream of PIEZO1 including NF- κB , PKC, YAP1, mTOR, and ERK (**Fig. S8**). Together, these findings provide molecular evidence that PIEZO1 is activated in response to capillary-scale mechanical compression and likely plays a key role in initiating the downstream signalling cascades observed in this model.

3.2 PIEZO1 translocation to the cytoplasmic membrane is not well supported. A PIEZO1 Western blot of subcellular protein fractionations should be provided.

We acknowledge that our original wording may have been misleading. The term ¹'translocation'¹ could suggest an active trafficking of PIEZO1 from the cytoplasm to the plasma membrane, which

was not directly measured in our study. Instead, our intent was to describe the observed increase in PIEZO1 membrane-associated signal following mechanical deformation, as visualized by immunofluorescence (**Fig. 4A, B**). This likely reflects a redistribution or clustering of PIEZO1 within membrane subdomains in response to membrane tension, positioning the channel for enhanced mechanosensitive activation.

In support of functional activation, real-time calcium imaging demonstrated immediate calcium influx upon squeezing, consistent with PIEZO1 channel opening. Moreover, pharmacological activation of PIEZO1 with Yoda1 induced stem-like features even in the absence of mechanical stress, while inhibition with Ruthenium Red and GsMTx4 prevented the acquisition of stem-like traits following constriction. These functional assays collectively confirm that membrane-localized PIEZO1 activation mediates the observed phenotypic effects.

We believe that the combined evidence from our imaging, functional calcium assays, pharmacological modulation, and genetic knockout together provide robust and physiologically meaningful support for PIEZO1 activity at the plasma membrane following mechanical deformation. We have clarified these points in the revised manuscript (p. 23, lines 8-9; p. 24 lines 1-6).

“Given that our microcapillary model imposes abrupt mechanical deformation and elicits downstream changes associated with calcium handling, we hypothesized that PIEZO1 could play a key role in sensing and transducing these mechanical cues. To test this, we investigated PIEZO1 expression and localization at the plasma membrane using immunofluorescence, noting an increase in PIEZO1 signal intensity in squeezed melanoma cells when compared to un-squeezed controls (**Fig. 4A, B**). This suggests that mechanical deformation promotes membrane reorganization and clustering of PIEZO1, positioning the channel for enhanced mechanosensitive activation under capillary-like stress and facilitating downstream calcium influx.”

3.3 To further support the specificity of their findings, authors could include a cell line which does not express PIEZO1. If not available, they could try overexpressing PIEZO1 and test if cells become stem-like without compression.

We thank the reviewer for this constructive suggestion. To our knowledge, there are currently no commonly used cancer cell lines that completely lack PIEZO1 expression. Instead of overexpression, we chose to employ Yoda1, a highly specific chemical agonist of PIEZO1, to directly test whether PIEZO1 activation alone is sufficient to induce a stem-like phenotype. Treatment with Yoda1 led to a significant increase in stemness marker expression and enhanced tumorsphere formation, even in the absence of mechanical deformation, supporting the idea that PIEZO1 activation is sufficient to drive this phenotypic switch. These results, combined with our PIEZO1 knockout and inhibitor data, strongly reinforce the specificity of the observed effects and further support the central role of PIEZO1 in mediating the response to mechanical constriction.

3.4 The observation of PIEZO1 pathway in modulating stem-like genes in a Yoda1-dependent manner is interesting; the use of Yoda1 to evaluate the epistasis makes sense, however ruthenium red (RR) inhibits all Ca²⁺ channels. It'd be more direct to inhibit PIEZO1, by transducing cells siRNA or sgRNA.

To address the issue of specificity more directly, we complemented our approach with both genetic and targeted pharmacological tools. Specifically, we performed CRISPR/Cas9-mediated knockout of PIEZO1, which provided a definitive means to assess the role of this channel. In these knockout cells, the acquisition of stem-like features following mechanical deformation was abolished, underscoring the necessity of PIEZO1 for the observed phenotype. To support the

Yoda1 (specific PIEZO1 agonist) and RR (broad channel antagonist) results, we conducted a new study using GsMTx4, a peptide inhibitor known to target mechanosensitive ion channels including PIEZO1 (New Fig. S10). Treatment with GsMTx4 significantly reduced stemness marker expression and tumorsphere formation. Taken together, the consistency across pharmacological activation, broad and targeted inhibition, and genetic knockout provides robust support for a PIEZO1-dependent mechanism. That said, we agree with the reviewer that siRNA approaches could offer temporal control of PIEZO1 suppression and may be valuable in future studies to dissect more dynamic aspects of its regulation.

In addition, the authors should address whether is the actual influx of Ca²⁺ and/or the conformational change of PIEZO1 to modulate the pathway using Ca²⁺ chelators or reproduce the experiment in Ca²⁺-free media.

We thank the reviewer for this excellent suggestion. To directly assess whether the observed effects are dependent on Ca²⁺ influx rather than PIEZO1 conformational change alone, we conducted additional experiments in Ca²⁺-free media. Under these conditions, melanoma cells subjected to mechanical constriction did not exhibit upregulation of stemness markers (e.g., CD44, ABCB5), indicating that extracellular calcium entry is required to drive the phenotypic switch (New Fig. S11). In combination with our pharmacological inhibition experiments (using Ruthenium Red and GsMTx4), PIEZO1 knockout data, and Yoda1 activation studies, these results collectively confirm that PIEZO1-mediated calcium influx specifically modulates the downstream phenotypic reprogramming following mechanical deformation.

The new data have been included in the revised manuscript (New Fig. S11) and are now discussed in the Method section (p.10, lines 6-9):

“For calcium-deprivation experiments, cells were suspended in calcium-free DMEM (Gibco, Cat. Number 21068-028) throughout the constriction procedure to assess the role of extracellular calcium influx during mechanical stimulation. After constriction in calcium-free media, cells were similarly seeded for downstream analysis.”

Results sections (p.25, lines 8-14):

“Finally, to directly assess the requirement of extracellular calcium influx, we performed squeezing experiments under calcium-depleted conditions. In calcium-free media, mechanical squeezing failed to induce upregulation of stemness markers CD44 and ABCB5 (**Fig. S11**), indicating that PIEZO1-mediated calcium entry is necessary to initiate the stem-like phenotype. Together, these results support a model in which mechanical stress activates PIEZO1-dependent calcium signalling to drive melanoma stem cell-like reprogramming.”

And Discussion (p. 33, lines 32-33; p. 34 line 1-3):

“The increased calcium signalling observed in our microcapillary constriction model suggests a relationship between calcium dynamics and the emergence of a stem cell-like state. Consistent with this, when cells were squeezed in calcium-free media, we observed no upregulation of stemness markers, indicating that calcium signalling is required to mediate the phenotypic response to mechanical stress.”

3.5 There is no western blot nor qPCR showing the actual efficacy of PIEZO1 KO, please provide it.

We thank the reviewer for this comment. The efficacy of PIEZO1 knockout was previously validated and documented in the doctoral thesis of Amrutha Patkunarajah (Reference 26:

Patkunarajah, A., *Mechanoelectrical signalling via ELKIN1 and PIEZO1 in cell lines derived from metastatic melanoma*, UNSW Sydney, 2022). As these validation data have been peer-reviewed and made publicly available via UNSWorks, we believe that referencing this work is sufficient to confirm the efficacy of PIEZO1 KO. However, should the editor or reviewers require, we are happy to provide the original data as supplementary material.

Is this a clone KO or a pool? In Material & Methods for this, the wrong reference (ref #24) seems to be cited. Specify, if it's a single clone and not pooled cells, authors should repeat the experiments in additional 2 clones.

We thank the reviewer for raising this important point. The PIEZO1 knockout line used in our study is based on a single CRISPR-generated clone originally described in Patkunarajah et al., *Mechanoelectrical signalling via ELKIN1 and PIEZO1 in cell lines derived from metastatic melanoma*. 2022, UNSW Sydney. (corrected reference).

While we acknowledge that the use of a single clone may not fully capture potential clonal variability, we believe that several lines of converging evidence strongly support the functional role of PIEZO1 in our system:

- Pharmacological activation of PIEZO1 using Yoda1 was sufficient to induce stemness features in the absence of mechanical stress.
- Pharmacological inhibition of PIEZO1 using both Ruthenium Red and GsMTx4 abrogated the acquisition of stem-like traits after mechanical deformation.
- Real-time calcium imaging demonstrated calcium influx upon deformation, absent in the PIEZO1 knockout cells.
- The KO cells failed to exhibit deformation-induced changes in histone modifications, stemness marker expression, or tumorsphere formation, phenocopying the effects of pharmacological inhibition.

These functional data, combined with the genetic knockout, provide robust and consistent evidence that PIEZO1 mediates the observed mechanotransduction responses. We agree that additional independent KO clones could further strengthen the conclusions, and this remains a priority for future work

Also, is the viability of KO cells affected?

The viability of PIEZO1 KO cells is reduced by mechanical constriction, but to no greater extent to what was observed in the WT cells, indicating that PIEZO1 is not required for immediate survival under deformation. However, a key distinction emerges in the resilient, surviving cell population: while WT cells show a clear adaptive switch characterized by stemness marker upregulation and functional changes (e.g., tumorsphere formation), KO cells fail to exhibit this phenotypic shift. This suggests that PIEZO1 is not involved in acute survival but is essential for triggering the mechanosensitive transcriptional reprogramming that underlies post-constriction adaptation. We have included viability data in Figure 5B and clarified this in the revised Result (p.25, lines 2022):

“Viability analysis showed that KO cells experienced a similar degree of cell loss upon deformation as WT cells, indicating that PIEZO1 deletion does not impact acute survival after constriction (Fig. 5B).”

And Discussion section (p.35, lines15-20):

“Viability analysis showed that KO cells experienced a similar degree of cell death upon deformation as WT cells, indicating that PIEZO1 is not essential for acute survival following

mechanical stress. However, in the surviving population, only WT cells exhibited the adaptive acquisition of stemness features and increased tumorsphere formation, while PIEZO1 KO cells failed to upregulate these markers or form tumorsphere, behaving similarly to RR-treated cells.”

To really prove the specificity of PIEZO1 KO effects, two independent sgRNAs or an orthogonal approach (e.g. shRNA) should be used. Alternatively, to prove the specificity of the KO experiments with Yoda1, RR, or Ca²⁺ depletion should be repeated in KO cells (or cells lacking PIEZO1 levels).

In the current study, we did not use multiple independent sgRNAs or an orthogonal knockdown approach such as shRNA. However, we confirmed the efficacy of the PIEZO1 knockout at both the genomic and protein levels, ensuring the reliability of the KO model used in our experiments. We believe this is a valuable suggestion. However, in the interest of communicating our finding to the broader community in a timely fashion, we believe our evidence is sufficient to garner interest and we hope these approaches will follow our work in subsequent studies.

3.6 Fig5N-P cell viability of all conditions is missing- 2D proliferation assays should be provided, to demonstrate if the effects are specific for tumour sphere formation or simply affect cell viability.

We thank the reviewer for this suggestion. We did not perform a 2D proliferation assay in the context of the tumorsphere experiments. However, all conditions were seeded at equal densities and cultured under identical conditions, and cell viability was assessed prior to tumorsphere seeding to ensure that differences in tumorsphere formation were not due to baseline differences in cell health (New Fig. 5B). While we acknowledge that a direct 2D proliferation comparison could further support the specificity of the tumorsphere phenotype, we believe the current experimental design, using controlled seeding, consistent viability, and phenotypic readouts, adequately isolates the effect of mechanical deformation and PIEZO1 signalling on stem-like behaviour.

Figure 5B. Quantification of viable KO melanoma cells in control group (CTRL) compared to squeezed (SQZD) group. Cells were stained with Trypan Blue and counted using a hemocytometer. The results are expressed as the mean from three independent experiments.

Why is KO-SQZD-RR missing?

We appreciate the reviewer’s observation. The KO-SQZD-RR condition was not included in our experimental design because it represents a double negative scenario: both genetic ablation of PIEZO1 and pharmacological inhibition of mechanosensitive calcium influx. Based on our findings, PIEZO1 knockout alone (KO-SQZD) significantly reduces tumorsphere formation, and pharmacological inhibition in WT cells (WT-SQZD-RR) yields a similar reduction. As such, we considered the combined condition to be redundant, as it would not be expected to reveal additional insight beyond the effects already observed in PIEZO1-deficient cells. To focus the

study and maintain clarity in the experimental narrative, we prioritized conditions that directly test the role of PIEZO1 activation (e.g., Yoda1) and its necessity (KO and inhibitor), which together support the central conclusion.

Figs 5,

4.1. Authors conclude that squeezing increases the ability of cancer cells to disrupt blood vessel integrity to enhance extravasation and metastatic spread. However, the *in vivo* assay that they use is not conducive to test this hypothesis because it doesn't require extravasation. In the tail injection assay cancer cells are just adhered to the lung microcapillaries and start growing there. A true metastasis assay from a flank tumor or intracardiac injection requires extravasation and would represent a better model to assess effects of compression/squeezing on extravasation. This part of the study should be complemented by proper *in vivo* metastasis studies.

We thank again the reviewer for this important point. We fully acknowledge that the initial tail vein injection model is limited in its ability to directly assess extravasation, as it primarily measures lung trapping rather than true metastatic dissemination. In recognition of this, and as part of addressing the reviewer's prior major concern (point 2), we have already performed additional experiments using the intracardiac injection model, which enables systemic arterial dissemination and requires tumour cells to extravasate into distal tissues. As described in detail in our response to major concern #2 and included in the revised manuscript (New Fig.3), these experiments demonstrated that squeezed cells exhibited significantly increased metastatic burden across multiple organs (bone, brain, lungs) and reduced survival compared to controls. Together with our *in vitro* endothelial assays, these data support the conclusion that mechanical deformation primes cells for enhanced metastatic competence, including extravasation.

4.2. Optimally, the authors should evaluate the relevance of their findings to patient tissues. Authors should analyze FFPE of primary invasive melanomas (in vertical growth phase) for PIEZO1, looking for correlations with stem cells markers (CD44, ABCB5, PRDM14, CD271), or histone markers (H3K9me3, H3K9ac), to characterize whether their observations are likely to happen in human samples. Based on their findings, they would expect PIEZO1 cytoplasmic membrane localization in cells close to the epidermal to dermal boundary, and to endothelial/pericyte cells of the vessels. PIEZO1 positive cells should also express more stem-cell markers. Do PIEZO1 levels correlate with patients' outcomes?

We wholeheartedly agree that evaluating the relevance of our findings in patient tissue is an important endeavour. We anticipate that publication of our study will prompt exactly this type of investigation, and we look forward to seeing how clinical data aligns with our studies results. However, this undertaking will take considerable time which would preclude the timely sharing of our findings. We believe these aspects are better suited for follow on studies which we hope to explore after securing appropriate funding to do so. In the revised manuscript we have added a sentence about the importance of verifying these models and mechanisms in patient tissue (p.37, lines 24-26).

“An important next step will be to examine whether similar mechanisms operate in patient-derived melanoma tissues by assessing PIEZO1 expression and its potential correlation with stemness markers and epigenetic changes.”

- (1) Luo, M.; Cai, G.; Ho, K. K.; Wen, K.; Tong, Z.; Deng, L.; Liu, A. P. Compression enhances invasive phenotype and matrix degradation of breast Cancer cells via Piezo1 activation. *BMC molecular and cell biology* **2022**, *23*, 1-17.
- (2) Hung, W.-C.; Yang, J. R.; Yankaskas, C. L.; Wong, B. S.; Wu, P.-H.; Pardo-Pastor, C.; Serra, S. A.; Chiang, M.-J.; Gu, Z.; Wirtz, D. Confinement sensing and signal optimization via Piezo1/PKA and myosin II pathways. *Cell reports* **2016**, *15* (7), 1430-1441.
- (3) Fernandez-Gonzalez, R.; de Matos Simoes, S.; Röper, J.-C.; Eaton, S.; Zallen, J. A. Myosin II dynamics are regulated by tension in intercalating cells. *Developmental cell* **2009**, *17* (5), 736-743.
- (4) Ren, Y.; Effler, J. C.; Norstrom, M.; Luo, T.; Firtel, R. A.; Iglesias, P. A.; Rock, R. S.; Robinson, D. N. Mechanosensing through cooperative interactions between myosin II and the actin crosslinker cortexillin I. *Current Biology* **2009**, *19* (17), 1421-1428.
- (5) Hope, J. M.; Lopez-Cavestany, M.; Wang, W.; Reinhart-King, C. A.; King, M. R. Activation of Piezo1 sensitizes cells to TRAIL-mediated apoptosis through mitochondrial outer membrane permeability. *Cell death & disease* **2019**, *10* (11), 837.
- (6) Song, Y.; Chen, J.; Zhang, C.; Xin, L.; Li, Q.; Liu, Y.; Zhang, C.; Li, S.; Huang, P. Mechanosensitive channel Piezo1 induces cell apoptosis in pancreatic cancer by ultrasound with microbubbles. *Iscience* **2022**, *25* (2).
- (7) Gunawan, I.; Kohane, F. V.; Dey, M.; Nguyen, K.; Zheng, Y.; Neumann, D. P.; Vafaei, F.; Meijering, E.; Lock, J. G. Extensible Immunofluorescence (ExIF) accessibly generates high-plexity datasets by integrating standard 4-plex imaging data. *Nature Communications* **2025**, *16* (1), 1-15.
- (8) Lock, J. G.; Mann, T. J.; Zheng, Y.; Khan, T.; Neumann, D.; James, A.; Becker, T.; Roberts, T. Harnessing the potential of circulating tumour cells for precision diagnostics via multiplexed imaging of the levels and subcellular localizations of over 50 cell identity, state and signaling markers per cell. In *CLINICAL CANCER RESEARCH, 2024; AMER ASSOC CANCER RESEARCH 615 CHESTNUT ST, 17TH FLOOR, PHILADELPHIA, PA ...*: Vol. 30.
- (9) Espinoza-Sánchez, N. A.; Enciso, J.; Pelayo, R.; Fuentes-Pananá, E. M. An NFκB-dependent mechanism of tumor cell plasticity and lateral transmission of aggressive features. *Oncotarget* **2018**, *9* (42), 26679.
- (10) Ma, D.; Liang, R.; Luo, Q.; Song, G. Pressure loading regulates the stemness of liver cancer stem cells via YAP/BMF signaling axis. *Journal of Cellular Physiology* **2025**, *240* (1), e31451.
- (11) Sun, J.; He, N.; Wang, W.; Dai, Y.; Hou, C.; Du, F. PKC inhibitors regulate stem cell self-renewal by regulating H3K27me3 and H3K9me3. *American Journal of Translational Research* **2022**, *14* (6), 4295.
- (12) Kim, H. *Investigating the Role of mTOR in Histone Methylation*; McGill University (Canada), 2023.
- (13) Ueda, Y.; Richmond, A. NF-κB activation in melanoma. *Pigment cell research* **2006**, *19* (2), 112-124.
- (14) Madonna, G.; Ullman, C. D.; Gentilcore, G.; Palmieri, G.; Ascierto, P. A. NF-κB as potential target in the treatment of melanoma. *Journal of translational medicine* **2012**, *10*, 1-8.
- (15) Dhawan, P.; Richmond, A. A novel NF-κB-inducing kinase-MAPK signaling pathway up-regulates NF-κB activity in melanoma cells. *Journal of Biological Chemistry* **2002**, *277* (10), 79207928.
- (16) Diazzi, S.; Tartare-Deckert, S.; Deckert, M. The mechanical phenotypic plasticity of melanoma cell: an emerging driver of therapy cross-resistance. *Oncogenesis* **2023**, *12* (1), 7.
- (17) Huang, F.; Santinon, F.; Flores González, R. E.; Del Rincón, S. V. Melanoma plasticity: promoter of metastasis and resistance to therapy. *Frontiers in oncology* **2021**, *11*, 756001.
- (18) Zhang, S.; Cao, S.; Gong, M.; Zhang, W.; Zhang, W.; Zhu, Z.; Wu, S.; Yue, Y.; Qian, W.; Ma, Q. Mechanically activated ion channel Piezo1 contributes to melanoma malignant progression through AKT/mTOR signaling. *Cancer Biology & Therapy* **2022**, *23* (1), 336-347.
- (19) Han, Y.; Liu, C.; Zhang, D.; Men, H.; Huo, L.; Geng, Q.; Wang, S.; Gao, Y.; Zhang, W.; Zhang, Y. Mechanosensitive ion channel Piezo1 promotes prostate cancer development through the activation of the Akt/mTOR pathway and acceleration of cell cycle. *International journal of oncology* **2019**, *55* (3), 629-644.

- (20) Lake, D.; Corrêa, S. A.; Müller, J. Negative feedback regulation of the ERK1/2 MAPK pathway. *Cellular and Molecular Life Sciences* **2016**, *73* (23), 4397-4413.
- (21) Davidson, P. M.; Sliz, J.; Isermann, P.; Denais, C.; Lammerding, J. Design of a microfluidic device to quantify dynamic intra-nuclear deformation during cell migration through confining environments. *Integrative Biology* **2015**, *7* (12), 1534-1546.
- (22) Nath, B.; Bidkar, A. P.; Kumar, V.; Dalal, A.; Jolly, M. K.; Ghosh, S. S.; Biswas, G. Deciphering hydrodynamic and drug-resistant behaviors of metastatic EMT breast cancer cells moving in a constricted microcapillary. *Journal of clinical medicine* **2019**, *8* (8), 1194.
- (23) Cognart, H. A.; Viovy, J.-L.; Villard, C. Fluid shear stress coupled with narrow constrictions induce cell type-dependent morphological and molecular changes in circulating tumor cells. *bioRxiv* **2019**, 722306.
- (24) Malboubi, M.; Jayo, A.; Parsons, M.; Charras, G. An open access microfluidic device for the study of the physical limits of cancer cell deformation during migration in confined environments. *Microelectronic engineering* **2015**, *144*, 42-45.
- (25) Li, P.; Liu, X.; Kojima, M.; Huang, Q.; Arai, T. Automated cell mechanical characterization by on-chip sequential squeezing: from static to dynamic. *Langmuir* **2021**, *37* (27), 8083-8094.
- (26) Kamyabi, N.; Khan, Z. S.; Vanapalli, S. A. Flow-induced transport of tumor cells in a microfluidic capillary network: role of friction and repeated deformation. *Cellular and Molecular Bioengineering* **2017**, *10*, 563-576.
- (27) Woodward, J. Crossing the endothelium: E-selectin regulates tumor cell migration under flow conditions. *Cell adhesion & migration* **2008**, *2* (3), 151-152.
- (28) Schlesinger, M.; Bendas, G. Vascular cell adhesion molecule-1 (VCAM-1)—an increasing insight into its role in tumorigenicity and metastasis. *International journal of cancer* **2015**, *136* (11), 2504-2514.
- (29) Wang, P.-C.; Weng, C.-C.; Hou, Y.-S.; Jian, S.-F.; Fang, K.-T.; Hou, M.-F.; Cheng, K.-H. Activation of VCAM-1 and its associated molecule CD44 leads to increased malignant potential of breast cancer cells. *International journal of molecular sciences* **2014**, *15* (3), 3560-3579.
- (30) Okada, T.; Hawley, R. G.; Kodaka, M.; Okuno, H. Significance of VLA-4–VCAM-1 interaction and CD44 for transendothelial invasion in a bone marrow metastatic myeloma model. *Clinical & Experimental Metastasis* **1999**, *17*, 623-629.
- (31) Klemke, M.; Weschenfelder, T.; Konstandin, M. H.; Samstag, Y. High affinity interaction of integrin $\alpha 4\beta 1$ (VLA-4) and vascular cell adhesion molecule 1 (VCAM-1) enhances migration of human melanoma cells across activated endothelial cell layers. *Journal of cellular physiology* **2007**, *212* (2), 368-374.
- (32) Strozyk, E. A.; Desch, A.; Poeppelmann, B.; Magnolo, N.; Wegener, J.; Huck, V.; Schneider, S. W. Melanoma-derived IL-1 converts vascular endothelium to a proinflammatory and procoagulatory phenotype via NF κ B activation. *Experimental dermatology* **2014**, *23* (9), 670-676.
- (33) Zhang, P.; Fu, C.; Bai, H.; Song, E.; Song, Y. CD44 variant, but not standard CD44 isoforms, mediate disassembly of endothelial VE-cadherin junction on metastatic melanoma cells. *FEBS letters* **2014**, *588* (24), 4573-4582.
- (34) Price, E. A.; Coombe, D. R.; Murray, J. C. Endothelial CD44H mediates adhesion of a melanoma cell line to quiescent human endothelial cells in vitro. *International journal of cancer* **1996**, *65* (4), 513-518.
- (35) Cox, C. D.; Gottlieb, P. A. Amphipathic molecules modulate PIEZO1 activity. *Biochemical Society Transactions* **2019**, *47* (6), 1833-1842.
- (36) Cox, C. D.; Bae, C.; Ziegler, L.; Hartley, S.; Nikolova-Krstevski, V.; Rohde, P. R.; Ng, C.-A.; Sachs, F.; Gottlieb, P. A.; Martinac, B. Removal of the mechanoprotective influence of the cytoskeleton reveals PIEZO1 is gated by bilayer tension. *Nature communications* **2016**, *7* (1), 10366.
- (37) Yang, S.; Miao, X.; Arnold, S.; Li, B.; Ly, A. T.; Wang, H.; Wang, M.; Guo, X.; Pathak, M. M.; Zhao, W. Membrane curvature governs the distribution of Piezo1 in live cells. *Nature communications* **2022**, *13* (1), 7467.
- (38) Nourse, J. L.; Leung, V. M.; Abuwarda, H.; Evans, E. L.; Izquierdo-Ortiz, E.; Ly, A. T.; Truong, N.; Smith, S.; Bhavsar, H.; Bertaccini, G. Piezo1 regulates cholesterol biosynthesis to influence neural stem cell fate during brain development. *Journal of General Physiology* **2022**, *154* (10).

- (39) Griffiths, E. J. Use of ruthenium red as an inhibitor of mitochondrial Ca²⁺ uptake in single rat cardiomyocytes. *FEBS letters* **2000**, 486 (3), 257-260.
- (40) Amann, R.; Maggi, C. A. Ruthenium red as a capsaicin antagonist. *Life sciences* **1991**, 49 (12), 849-856.
- (41) Tapia, R.; Velasco, I. Ruthenium red as a tool to study calcium channels, neuronal death and the function of neural pathways. *Neurochemistry international* **1997**, 30 (2), 137-147.
- (42) Hamilton, M. G.; Lundy, P. M. Effect of ruthenium red on voltage-sensitive Ca⁺⁺ channels. *The Journal of pharmacology and experimental therapeutics* **1995**, 273 (2), 940-947.
- (43) Gnanasambandam, R.; Ghatak, C.; Yasmann, A.; Nishizawa, K.; Sachs, F.; Ladokhin, A. S.; Sukharev, S. I.; Suchyna, T. M. GsMTx4: mechanism of inhibiting mechanosensitive ion channels. *Biophysical journal* **2017**, 112 (1), 31-45.
- (44) Botello-Smith, W. M.; Jiang, W.; Zhang, H.; Ozkan, A. D.; Lin, Y.-C.; Pham, C. N.; Lacroix, J. J.; Luo, Y. A mechanism for the activation of the mechanosensitive Piezo1 channel by the small molecule Yoda1. *Nature communications* **2019**, 10 (1), 4503.
- (45) Patkunarajah, A. Mechanoelectrical signalling via ELKIN1 and PIEZO1 in cell lines derived from metastatic melanoma. UNSW Sydney, 2022.
- (46) Au, S. H.; Storey, B. D.; Moore, J. C.; Tang, Q.; Chen, Y.-L.; Javaid, S.; Sarioglu, A. F.; Sullivan, R.; Madden, M. W.; O'Keefe, R. Clusters of circulating tumor cells traverse capillary-sized vessels. *Proceedings of the National Academy of Sciences* **2016**, 113 (18), 4947-4952.
- (47) Headley, M. B.; Bins, A.; Nip, A.; Roberts, E. W.; Looney, M. R.; Gerard, A.; Krummel, M. F. Visualization of immediate immune responses to pioneer metastatic cells in the lung. *Nature* **2016**, 531 (7595), 513-517.
- (48) Guyton, A. C.; Hall, J. E. *Guyton and Hall textbook of medical physiology*, Elsevier, 2011.
- (49) Lee, H. J.; Diaz, M. F.; Price, K. M.; Ozuna, J. A.; Zhang, S.; Sevcik-Muraca, E. M.; Hagan, J. P.; Wenzel, P. L. Fluid shear stress activates YAP1 to promote cancer cell motility. *Nature communications* **2017**, 8 (1), 14122.
- (50) Sun, J.; Luo, Q.; Liu, L.; Song, G. Low-level shear stress promotes migration of liver cancer stem cells via the FAK-ERK1/2 signalling pathway. *Cancer letters* **2018**, 427, 1-8.
- (51) Qin, X.; Li, J.; Sun, J.; Liu, L.; Chen, D.; Liu, Y. Low shear stress induces ERK nuclear localization and YAP activation to control the proliferation of breast cancer cells. *Biochemical and biophysical research communications* **2019**, 510 (2), 219-223.
- (52) Triantafillu, U. L.; Park, S.; Klaassen, N. L.; Raddatz, A. D.; Kim, Y. Fluid shear stress induces cancer stem cell-like phenotype in MCF7 breast cancer cell line without inducing epithelial to mesenchymal transition. *International journal of oncology* **2017**, 50 (3), 993-1001.
- (53) Irianto, J.; Xia, Y.; Pfeifer, C. R.; Athirasala, A.; Ji, J.; Alvey, C.; Tewari, M.; Bennett, R. R.; Harding, S. M.; Liu, A. J. DNA damage follows repair factor depletion and portends genome variation in cancer cells after pore migration. *Current biology* **2017**, 27 (2), 210-223.
- (54) Nava, M. M.; Miroshnikova, Y. A.; Biggs, L. C.; Whitefield, D. B.; Metge, F.; Boucas, J.; Vihinen, H.; Jokitalo, E.; Li, X.; Arcos, J. M. G. Heterochromatin-driven nuclear softening protects the genome against mechanical stress-induced damage. *Cell* **2020**, 181 (4), 800-817. e822.
- (55) Hsia, C.-R.; McAllister, J.; Hasan, O.; Judd, J.; Lee, S.; Agrawal, R.; Chang, C.-Y.; Soloway, P.; Lammerding, J. Confined migration induces heterochromatin formation and alters chromatin accessibility. *Iscience* **2022**, 25 (9).
- (56) Karska, J.; Kowalski, S.; Saczko, J.; Moisesescu, M. G.; Kulbacka, J. Mechanosensitive ion channels and their role in cancer cells. *Membranes* **2023**, 13 (2), 167.

Reviewer's Comments:

Reviewer #1 (Remarks to the Author):

The authors have done a good job in responding to my prior review comments. Especially the addition of new data to explore the points raised. The Lamin A experiment probably needs some context for the reader, because in the current manuscript it sort of comes out of nowhere.

We thank the reviewer for this suggestion. In the new revised manuscript, we add some context to the text to rationalise our experiments with Lamin A. See result section (p. 4 lines 13-17):

“Since mechanical stress on the nucleus can weaken the nuclear envelope, we assessed Lamin A, a key nuclear membrane protein responsible for maintaining nuclear shape and stability, to determine whether nuclear structure was compromised. Immunolabelling showed no significant change in Lamin A expression between squeezed and control cells (Fig. S1A).”

Reviewer #2 (Remarks to the Author):

How do PIEZO protein levels or subcellular localization change in response to microcapillary ‘squeezing’? The authors performed phosphoproteomics after squeezing and identified significant increases in pERK, NFkB, YAP1/TEAD, PKC, and mTOR. Using inhibitors against these pathways (e.g., Trametinib, Rapamycin) could provide insights into the mechanisms of PIEZO1 increased expression and activity.

We appreciate the reviewer's guidance and agree that the use of inhibitors would provide additional evidence in support of mechanism. However, the data as presented clearly implicates signal transduction downstream of PIEZO1 which was the intent of the experiment. Future work evaluating intermediate signal transduction has value but is outside of the scope of the [present study.

How does PIEZO1 alteration result in H3K9me3 increase and expression of stem-like genes? A genome-wide characterization of H3K9me3 and H3K27ac deposition would help determine if differences in intensity are accompanied by de novo deposition of histone modifications at specific genes and regulatory elements, which is more compatible with a phenotypic change.

Control and squeezed cells should be assessed by H3K9me3 ChIP-seq or CUT&Tag/Run to determine if/how H3K9me genome distribution changes upon microcapillary-induced constriction. These analyses would reveal if the observed changes in histone marks are (at least partially) responsible for the differentially expressed genes observed. The authors have already performed RNA-seq in control and SQZD conditions. Integrating transcriptional changes with ChIP-seq or Cut&Tag/Run for H3K9me3 and H3K27ac would provide a more complete epigenetic characterization of whether there is a switch to repressive or active chromatin at specific loci. These experiments would also allow for DNA binding motif analysis at the differential peaks to uncover potential transcription factors involved in the direct activation of PIEZO1 and the differentially expressed genes (e.g., NFkB, JUN, TEAD, BHLH, known to be downstream of the pathways identified by phosphoproteomics).

We show how cell and nuclear deformation leads to epigenetic changes via the H3K9 and H3K27 states, corresponding to regulation of gene expression. The data as presented demonstrates how constriction regulates gene expression, which we then follow up by evaluating outside-in signaling through PIEZO1. As suggested by the

reviewer, a more detailed epigenetic analysis would shed light on molecular mechanisms underlying the phenotype shifts. However, these experiments would take a considerable amount of time and we believe it is critical to convey our findings to the community as soon as possible. These suggested experiments are better suited for follow up investigation.

In vivo/clinical relevance: The authors could inspect cancer cells in proximity to blood vessels or in tumor-dense regions where mechanical compression may elicit a similar activation of PIEZO1 and its targets. An in vivo confirmation would provide further support for the physiological relevance of the reported in vitro effects.

This is a very exciting prospect, one of which we are currently in discussion with clinical collaborators as a follow up to our findings. We are hopeful that gauging PIEZO1 related signaling in clinical samples will add weight to our study with scope for helping patients. However, this analysis will require considerable planning and ethical approval, which would delay the timely reporting of our findings. Therefore, we believe this suggestion is better suited as a follow up study.

Nevertheless, we have added the following sentence to the discussion highlighting the potential of these future studies. See Discussion section (p.17 lines. 22-30).

“Tracking CTC dissemination in vivo, particularly in tumour regions subject to mechanical compression or in proximity to blood vessels, could offer further insight into the relevance of PIEZO1-mediated signalling under physiological conditions. Future studies examining PIEZO1 activation and associated signalling in compressed tumour regions or perivascular niches within clinical specimens will be critical to validate the physiological relevance of this mechanism and assess its translational potential in metastatic disease. Furthermore, targeting PIEZO1 or biophysical mechanisms that precede PIEZO1 activation (e.g. capillary-like constrictions) could prove a new therapeutic avenue in combatting metastatic disease.”

Injecting control vs. squeezed cells subcutaneously or intradermally and monitoring tumor growth would be important to assess if the increase in metastasis following compression is truly due to enhanced metastatic capacity and/or proliferation. Faster-growing cells are more likely to cause a higher metastatic burden.

We agree with the reviewer that evaluating subcutaneous or intradermal growth traits would be interesting. However, we believe this is outside of the scope of our study, which aimed to demonstrate how constriction during circulation augmented metastatic potency and tumorigenicity. The results of both tail vein and intracardiac metastasis models in mice are viewed as sufficient evidence to support our findings.

Reviewer #3 (Remarks to the Author):

The authors have satisfactorily addressed my concerns. Just a final suggestion, I noticed the added Method section is written quite carelessly. For example on page

5: what is ISM?, ul?, ship? Also, why does a "viscosity model" give Young's modulus?

We thank the reviewer for their careful reading and helpful feedback. We have revised the Methods section to correct typographical errors (e.g., "ISM" corrected to "mOsm", "ul" to "μL", and "ship" to "chip"). Regarding the calculation of Young's modulus, we clarified in the text that the value was derived using a calibrated viscoelastic model that relates cell deformation and projected area to effective stiffness, as described in reference [26]. We appreciate the reviewer's attention to detail and have addressed these issues to improve clarity and accuracy.